# Significant nutrient consumption in the dark subsurface layer during a diatom bloom: the case study on Funka Bay, Hokkaido, Japan

Sachi Umezawa[1], Manami Tozawa[1], Yuichi Nosaka[2], Daiki Nomura[3,4,1], Hiroji Onishi[1], Hiroto Abe[1], Tetsuya Takatsu[1], Atsushi Ooki[1,4]

[1]Graduate School of Fisheries Sciences/Faculty of Fisheries Sciences, Hokkaido University, 3-1-1 Minato-cho, Hakodate, Hokkaido, 041-8611, Japan

[2]Tokai University, Department of Marine Biology and Sciences, 5-1-1, Minamisawa, Minami-ku, Sapporo, Hokkaido, 005-8601, Japan

[3]Field Science Center for Northern Biosphere, Hokkaido University, 3-1-1 Minato Cho, Hakodate, Hokkaido 041-8611, Japan

[4]Arctic Research Center, Hokkaido University, Kita-21 Nishi-11 Kita-ku, Sapporo, Hokkaido, 001-0021, Japan

*Correspondence to*: Atsushi Ooki (ooki@fish.hokudai.ac.jp)

**Abstract.** We conducted repetitive observations in Funka Bay, Hokkaido, Japan, on 15 February, 4 and 15 March, and 14 April 2019. The diatom spring bloom peaked on 4 March and started declining on 15 March. Funka Bay winter water remained below 30-m depth, which was below the surface mixed-layer and dark-layer depth (0.1% of surface PAR depth) on 4 and 15 March. In the subsurface layer at depths of 30–50 m, concentrations of $NO_3^-$, $PO_4^{3-}$, and $Si(OH)_4$ decreased by half between these dates even in the dark. Incubation experiments using the diatom *Thalassiosira nordenskioeldii* showed that this diatom could consume added nutrients in the dark at substantial rates after the pre-culturing to deplete nutrient. Incubation experiments using natural seawater collected in growing phase of bloom on 8 March 2022 also showed that nutrient-depleted phytoplankton could consume added nutrients in the dark. We excluded possibilities of three physical process, water mixing, diffusive transport, and subduction, as the main reasons for the decrease in nutrients in the subsurface layer. We conclude that the nutrient reduction in the subsurface layer (30–50 m) between 4 and 15 March 2019 could be explained by dark consumption by diatoms in that layer.

## 1 Introduction

The supply of nutrients to the euphotic zone has a potent influence on regulating marine primary production. Numerous studies have examined nutrient utilization by marine biota in relation to the nutrient cycles among the euphotic zone and conterminous zones (e.g., below the euphotic zone, atmosphere, rivers). In the subarctic North Pacific Ocean, the higher Si:N

ratio in the surface of the western gyre (Oyashio region) leads to a diatom-dominant population, and the lower ratio in the eastern gyre leads to a reduced diatom population; both subarctic gyres are known to be high-nitrate, low-chlorophyll (HNLC) regions, where depletion of dissolved iron (D-Fe) limits the primary production (Harrison et al. 2004). Dissolved iron and nitrate ($NO_3^-$) supplied from below the surface to the surface euphotic zone through winter vertical water mixing sustain spring phytoplankton bloom in the Oyashio region (Nishioka et al. 2011). Most previous studies about marine primary production have concerned with the nutrient consumption by phytoplankton in the euphotic zone because most phytoplankton species, except for dinoflagellates (e.g. Cullen and Horrigan 1981), are commonly assumed to be incapable of moving actively between the surface mixed layer and below the surface (subsurface layer). A few studies have focused on the vertical migration of a diatom, *Rhizosolenia*, to uptake nutrients in the subsurface layer and grow in the euphotic zone in the oligotrophic subtropical Pacific (Villareal et al. 1996; Richardson et al. 1998; Villareal et al., 1999; Villareal et al., 2014). As for the subarctic area, a modelling study that simulated chl-a profiles, taking into phytoplankton's migration behaviour, demonstrated that vertically migrating phytoplankton can pump up considerable amount of nutrient to the surface layer from the dark subsurface layer and contributes 7% of net primary production at the subarctic gyre of the western Pacific, Oyashio region (Witz and Lan Smith, 2020). These previous studies have not yet shown observational evidence of nutrient reduction associated with consumption by phytoplankton in the dark subsurface layer, however, nutrient reduction in the dark subsurface layer have been found in the Funka Bay, Hokkaido, Japan (Kudo and Matsunaga, 1999), which faces to the Oyashio-Kuroshio transitional area in the western North Pacific.

Oyashio water reaches the area off the coast of Hokkaido, Japan, where the subtropical water derived from Kuroshio or the Tsugaru warm current water is also found (Rosa et al., 2007). A small portion of Oyashio water enters Funka Bay in early spring. The bay water exchanges twice a year, with cold Oyashio water in early spring and Tsugaru warm water in early fall (Ohtani 1971). The Oyashio water, a cold and low salinity water, flows into the bay along the northern coast of the bay, forming an anticlockwise flow from late of March to middle of April (Nakada et al. 2013). From repetitive observations in the bay, it is possible to collect seawater samples originating from the same water mass in different times when the water remains in the bay during the observation period and then examine the temporal changes of biogeochemical parameters within the same water mass. For example, temporal changes in nutrients (Kudo and Matsunaga 1999; Kudo et al. 2000), dissolved iron (Hioki et al. 2015), volatile organic iodine (Shimizu et al. 2017) and isoprene (Ooki et al. 2019; Ooki et al. 2022) have been examined in relation to primary production in the bay. In the Funka Bay, diatom bloom initiates in late winter, February, before Oyashio water flows into the bay (Kudo and Matsunaga, 1999). A massive spring bloom dominated by diatom species occurs in March every year (Odate 1987; Maita and Odate 1988) when Oyashio water flows into the bay. The bloom lasts until late March or early April when Oyashio water occupies the surface of the bay (Kudo and Matsunaga, 1999). The spring diatom bloom ends because of nitrate depletion, but silicate is further consumed after the nitrate depletion (Kudo et al. 2000). After the bloom, phosphorus depletion in the bay occasionally limits the primary production (Yoshimura and Kudo 2011). D-Fe is not depleted (>3 nmol $L^{-1}$) in the surface waters of the bay in April (post-bloom) (Hioki et al. 2015), so D-Fe would not limit primary production. One-third of annual primary production occurs during the spring bloom (Kudo

and Matsunaga 1999; Kudo et al. 2015). Thus, depletion of macronutrients (N, P, Si) is the dominant limiting factor for production in the bay. After the bloom, there is extensive settling and sedimentation of particulate organic matter on the seafloor (Miyake et al. 1998), and nutrient regeneration rapidly occurs in the bottom water just after the sedimentation (Kudo et al. 2007).

Although most previous studies in Funka Bay have focused on nutrient consumption in the euphotic zone and nutrient regeneration in the bottom water, Kudo and Matsunaga (1999) have pointed out that $NO_3^-$ concentrations in the dark subsurface layer decreased during the spring blooms in 1991 during their observations from 1988 to 1992, and they mentioned that the decrease was due to dilution of water by vertical mixing. In figures of time-depth section of $NO_3^-$ concentration in Kudo and Matsunaga, it seems that decreases in $NO_3^-$ below the surface have occurred during the bloom of all years. This raises the question as to why the nutrient reduction in the dark subsurface layer occurs so frequently, and if it can be attributed to vertical mixing every year, as the surface layer in the bay usually rapidly becomes stratified in spring.

In this paper, we examine the temporal variation of nutrient concentrations in Funka Bay from the early phase of the diatom bloom (February) to post-bloom (April) through repetitive observations in 2019. And we focused on the processes affecting nutrient reduction in the dark subsurface layer during the bloom to show evidence of nutrient consumption by diatoms in the dark.

## 2 Materials and methods

### 2.1 Shipboard observations

Shipboard observations were conducted in Funka Bay, Hokkaido, Japan, on 15 February, 4 and 15 March, and 14 April 2019. We used the training ship (T/S) *Ushio-maru*, operated by the Faculty of Fisheries Sciences, Hokkaido University. Water samples were collected at station 30 (St. 30; 42°16.2′N, 140°36.0′E; bottom depth, 96 m) located in the centre of Funka Bay (Fig. 1, right panel). Seawater samples were collected in 2.5-L Niskin bottles attached to a rosette multi-sampler along with a conductivity-temperature-depth (CTD) probe (SBE 19 plus, Sea-Bird Electronics, Inc.). Surface water was collected with a plastic bucket, and bottom water was collected approximately 1 m above the seafloor using a Van Dorn sampling bottle. The sampling depths were 0, 5, 10, 20, 30, 40, 50, 60, 65, 70, 75, 80, 85, and 95 m (1 m above the seafloor).

### 2.2 Analytical procedures

Chl-*a* concentrations in discrete seawater samples (100 mL) were measured using the fluorometric Welschmeyer method (Welschmeyer 1994) and a Turner Designs fluorometer (model 10-AU-005). Concentrations of nutrients ($NO_3^-$, $NO_2^-$, $NH_4^+$, $Si(OH)_4$, and $PO_4^{3-}$) in discrete seawater samples were measured by colorimetric methods using a QuAAtro system (BL-tec). Analytical precision was 0.12% for $NO_3^-$, 0.21% for $NO_2^-$, 0.19% for $PO_4^{3-}$, 0.11% for $Si(OH)_4$, and 0.34% for $NH_4^+$ as determined by repetitive measurement (n = 7) of reference seawater for nutrient standards (KANSO, standard Lot BZ, Osaka, Japan). Dissolved oxygen was determined by Winkler titration using a 798 MPT Titrino analyzer (Metrohm, Herisau,

Switzerland). Apparent oxygen utilization (AOU) was calculated by subtracting the measured oxygen concentration from the dissolved oxygen concentration at saturation under in situ temperature and salinity (Hansen 1999).

## 2.3 Incubation experiments to test for nutrient consumption by diatoms in the dark

We conducted dark incubation experiments four times using a diatom *Thalassiosira nordenskioeldii*, which predominates in the early phase of the spring bloom in Funka Bay (Ban et al 2000), and twice using natural seawater collected in diatom
bloom 2022.

### 2.3.1 *Thalassiosira nordenskioeldii* incubation experiment

*Thalassiosira nordenskioeldii* was isolated from natural seawater collected in the western subarctic Pacific Ocean in May 2019. In the first and second *Thalassiosira* experiments, an axenic culture of the diatom was grown in modified f/2 medium ($NO_3^-$, 700 µmol $L^{-1}$; $PO_4^{3-}$, 26 µmol $L^{-1}$; $Si(OH)_4$, 75 µmol $L^{-1}$) at 6 °C (pre-culture). We used a 250-mL cell-cultivation
flask with a vent cap (VTC-F75V, Violamo). Incubation procedures were carried out under axenic conditions; however, we did not check for contamination after the incubation. When the chl-*a* concentration in the pre-culture medium reached 1426 µg $L^{-1}$ on day 17 of pre-culturing for the first experiment, the concentrations of $NO_3^-$, $PO_4^{3-}$, and $Si(OH)_4$ in the medium had dropped below 0.05, 0.05 and 1 µmol $L^{-1}$, respectively. We regarded the diatoms in the medium on day 17 as being nutrient-depleted. We added nutrients (stock f/2 medium) into the nutrient-depleted diatom culture, after which concentrations were
as follows: $NO_3^-$, 29.2 µmol $L^{-1}$; $PO_4^{3-}$, 1.13 µmol $L^{-1}$; and $Si(OH)_4$, 4.24 µmol $L^{-1}$. The incubation bottle (n = 1) was put in a dark incubator at 6 °C for 6 days. On days 0, 2, 3, 4, and 6 of dark incubation, 10 mL and 100 µL of incubation medium were filtered to measure nutrient and chl-*a* concentrations, respectively, using the same methods as for the measurements in seawater samples. The setup conditions of experiments were summarized in Table 1.

In the second *Thalassiosira* experiment, another diatom culture *Thalassiosira nordenskioeldii* had grown for 42 days. The
chl-*a* concentration in the pre-cultured medium for the second experiment was only 72.5 µg $L^{-1}$, which was one twentieth of that in the first dark incubation, implying that it was in a decline phase. We set the initial concentrations of nutrients ($NO_3^-$, 744 µmol $L^{-1}$; $PO_4^{3-}$, 27.6 µmol $L^{-1}$; and $Si(OH)_4$, 113 µmol $L^{-1}$) at 25 times those of the first experiment and put it in the dark incubator. On days 0, 1, 2, and 10, 10 mL and 1 mL of incubation medium (n = 1) were filtered to measure nutrient and chl-*a* concentrations, respectively.

In the third and fourth *Thalassiosira* experiments, 40 mL of diatom culture was divided into two pre-culturing incubation bottles with 170 mL of modified f/2 medium ($NO_3^-$, 175 µmol $L^{-1}$; $PO_4^{3-}$, 6.5 µmol $L^{-1}$; $Si(OH)_4$, 18.8 µmol $L^{-1}$). For the third *Thalassiosira* experiments, the pre-culturing was done for 10 days; concentration of chl-a in the pre-cultured medium became 145 µg $L^{-1}$ and concentrations of $NO_3^-$, $PO_4^{3-}$, and $Si(OH)_4$ dropped as follows: $NO_3^-$, 9.27 µmol $L^{-1}$; $PO_4^{3-}$, 0.42 µmol $L^{-1}$; and $Si(OH)_4$, < 1 µmol $L^{-1}$.  Relatively high concentrations of $NO_3^-$ and $PO_4^{3-}$ remained in the medium. For the
fourth *Thalassiosira* experiments, the pre-culturing was done for 11 days; concentration of chl-a in the pre-cultured medium became 198 µg $L^{-1}$ and concentrations of $NO_3^-$, $PO_4^{3-}$, and $Si(OH)_4$ dropped as follows: $NO_3^-$, 0.66 µmol $L^{-1}$; $PO_4^{3-}$, 0.42

μmol L$^{-1}$; and Si(OH)$_4$, < 1 μmol L$^{-1}$. Nutrients had been rapidly consuming at the end of the pre-culturing on day 10 of the third experiment. We added nutrients (stock f/2 medium) into the pre-cultured mediums, after which concentrations (day 0 of dark incubations) were as follows: NO$_3^-$, 187 μmol L$^{-1}$; PO$_4^{3-}$, 7.66 μmol L$^{-1}$; and Si(OH)$_4$, 16.1 μmol L$^{-1}$ for the third experiment; NO$_3^-$, 213 μmol L$^{-1}$; PO$_4^{3-}$, 9.08 μmol L$^{-1}$; and Si(OH)$_4$, 21.2 μmol L$^{-1}$ for the fourth experiment. The 30 mL of each pre-cultured medium was used for chl-a and nutrient measurements. The remaining 160 mL of each pre-cultured medium was divided into 4 cell-cultivation flasks, and they were put in the dark at 6 °C. On days 1, 2, and 3, 8 mL and 1 mL of incubation medium (n = 4) were filtered to measure nutrient and chl-$a$ concentrations, respectively.

### 2.3.2 Natural seawater incubation experiment

For the natural seawater incubation experiment, we collected seawater at 5- and 40-m depths at the station 30 of Funka Bay on 8 March 2022. Fourteen of 200 mL seawater samples were collected in cell-cultivation flasks (11 flasks for 5-m depth water and 3 flasks for 40-m depth water), and they were stored in a refrigerator for a day until treatment of the culture experiment. The concentrations of chl-a at 5- and 40-m depths were 14.3 and 9.09 μg L$^{-1}$, respectively. Three flasks of both depths water were put in a dark incubator at 5 °C for 12 days without nutrient addition (continuous dark (5 m / 40 m) in Table 1) as control. Another 8 flasks of 5-m depth water were precultured under light condition (100 μmol photon s$^{-1}$, light : dark = 12hr : 12 hr) for 5 days at 5 °C to deplete nutrients (nutrient-deplete (5 m) in Table 1); concentration of chl-a in the pre-cultured medium became 25.1 μg L$^{-1}$ and concentrations of NO$_3^-$, PO$_4^{3-}$, and Si(OH)$_4$ dropped as follows: NO$_3^-$, < 0.05 μmol L$^{-1}$; PO$_4^{3-}$, < 0.05 μmol L$^{-1}$; Si(OH)$_4$, < 1 μmol L$^{-1}$. Then nutrients were added into the 4 flasks of the nutrient-deplete precultured 5 m water, after which concentrations (day 0) were as follows: NO$_3^-$, 12.6 μmol L$^{-1}$; PO$_4^{3-}$, 0.38 μmol L$^{-1}$; and Si(OH)$_4$, 17.8 μmol L$^{-1}$. No nutrients were added to the other 4 of the 8 flasks of the nutrient-deplete precultured seawater. These 8 precultured incubation bottles (4 nutrient-addition bottles and 4 no-nutrient-addition bottles) were put in the dark incubator at 5 °C for 7 days. On days 0 and 7 on dark incubations, 8 mL and 10 mL of incubation medium were filtered to measure nutrient and chl-a concentrations, respectively.

### 2.4 Water mass types and mixed-layer, euphotic-zone, and dark-layer depths

There are two main water masses in Funka Bay throughout the year. Tsugaru water that originates from the subtropical North Pacific and Oyashio water that originates from the subarctic North Pacific. The subtropical Tsugaru water has higher salinity (33.6–34.2) and is modified into winter water by winter cooling. The subarctic Oyashio water has lower salinity (32.6–33.0) and is modified into low density summer water (S) by solar radiative heating and freshwater input. These four water masses were first described by Ohtani and Kido (1980). Ooki et al. (2019) added the transitional waters to the water-mass classification: changing from winter water to Oyashio water (WO), from Oyashio water to summer water (OS), from summer water to Tsugaru water (ST), and from Tsugaru water to winter water (TW). The temperature-salinity ranges of water masses are illustrated in supplementary figure (Fig. s1).

The surface mixed layer was defined as the layer in which density differences ($\Delta\sigma$) were within 0.125 kg m$^{-3}$ relative to the density at 5-m depth. The threshold $\Delta\sigma = 0.125$ kg m$^{-3}$ is often used for monthly mean of mixed layer in oceanic climate studies (Spall 1991), while the threshold $\Delta\sigma = 0.01$ kg m$^{-3}$ is used for snap-shot observations (Thomson and Fine 2003). We used the maximum threshold $\Delta\sigma = 0.125$ kg m$^{-3}$ to ensure that the subsurface layer water had not mixed with the surface layer during intervals (11 days to a month) between our observations. The euphotic-zone depth was defined as the depth at which photosynthetically active radiation (PAR) was 1% of the surface PAR, i.e. where photosynthesis equals respiration (Marra, 2014). We defined the dark-layer depth as the depth at which PAR was 0.1% of the surface PAR, i.e. where PAR was 0.1% of the surface PAR, where amount of photosynthesis is approximately a tenth of that at the 1% PAR level, taking into account the light intensity only as a limiting factor of photosynthesis.

## 2.5 Spatial distributions of temperature, salinity, and density at the sea surface

Spatial distributions of temperature, salinity, and density at the sea surface (1 m) were obtained from an ocean reanalysis product provided by Meteorological Research Institute in Japan. This is produced with an operational system for monitoring and forecasting the status of coastal and open-ocean waters around Japan (the JPN system; Hirose et al., 2020). The JPN system includes a double-nested ocean model, the core of which is a Japanese coastal model with a horizontal resolution of 2 km. Three sub-models are interconnected using a nesting technique: a global model (horizontal resolution ~100km), a North Pacific model (horizontal resolution ~10km), and Japanese coastal model (horizontal resolution ~2km). A four-dimensional variational method is applied to the North Pacific model as the assimilation scheme. The process of tides and river runoff are taken into consideration in this JPN system. See the technical report for more detail (Hirose et al. 2020).

## 3 Results

### 3.1 Hydrographic features

Vertical distributions of temperature, salinity, density, and ratio of PAR relative to the surface PAR from 15 February to 15 March are shown in Fig. 2a-d. Overview of temporal variation of water-mass structure is shown in Fig. 3. Temperature, salinity, and density were all vertically uniform on 15 February (Fig. 2a-c), where the water mass was categorized as winter water (Fig. 3). On 15 February, the temperature was the highest (4 °C) during the observation period. We named it as high-temperature winter water. On 4 March, temperature dropped below 3.7 °C in the total water column. We found significant decreases in salinity and density at the surface 0 – 20-m depth. The influence of Oyashio water (WO or O) was evident at the surface (0–7 m) on 4 March and had extended to the deeper layer (0–24 m) by 15 March, and to 0–52 m on 14 April. In deep layer below 70-m depth, temperature was increased to 3.9 °C on 15 March. We considered that the high-temperature winter water, which remained probably in the back of the bay, moved to the deep layer of the observation station on 15 March. And

we considered that the middle layer (subsurface layer, 30 – 50 m) was relatively stable between 4 and 15 March compared with the surface and deep layers where the waters were exchanged obviously.

As for light environment, euphotic-zone depths (or compensation depth), which are defined as the depth where PAR was 1% of the surface PAR, were 40 m on 15 February, 11 m on 4 March, and 17 m on 15 March. Dark-layer depths, which we defined as the depth where PAR was 0.1% of surface PAR, were 60 m on 15 February, 17 m on 4 March, and 30 m on 15 March (Fig. 2d). The daily average of surface PAR during the period between 4 March and 15 March including day and night was 19.3 mol photon $m^{-2}$ $d^{-1}$ (= 224 μmol photon $m^{-2}$ $s^{-1}$), which was estimated from the global solar radiation at the

Muroran meteorological observatory (Fig. s2). The daily average of PAR at the dark-layer depth was estimated to be 0.0193 mol photon $m^{-2}$ $d^{-1}$, which was only 1.8% of global average of compensation irradiance (1.1 mol photon $m^{-2}$ $d^{-1}$) for metabolic balance, photosynthesis = respiration (Regaudie-De-Gioux and Duarte, 2010) Below the dark-layer depths, we assumed that photosynthesis made no difference in biochemical parameters described in latter sections.

### 3.2 Biogeochemical parameters

We prepared time-depth sections of biogeochemical parameters, showing the surface mixed-layer, euphotic-zone, and dark-layer depths (Fig. 4a–f). Vertical profiles of the parameters are also shown in Fig. 5a-f.

#### 3.2.1 Chl-*a*

**15 February** The surface mixed layer reached at least 85-m depth on 15 February, but the chl-*a* profile was not vertically uniform; see supplementary data file. Chl-*a* concentrations between 0 and 70 m were relatively high, ranging from 0.65 to

205 1.4 μg $L^{-1}$ (average, 0.81 μg $L^{-1}$), compared with concentrations in the deeper layer (75–95 m), where they were below the detection limit (<0.05 μg $L^{-1}$). We suggest that on this date, diatom growth had just started as an early phase of the spring bloom.

**4 March** By 4 March the chl-*a* concentrations had substantially increased at all depths (Fig. 4a; Fig. 5e). Notably high concentrations (0–5m, 27–30 μg $L^{-1}$) were found in the surface mixed layer. The depth range of the mixed layer (0–9 m) was

210 almost the same as that of the euphotic zone (0–13 m). In Funka Bay, a chl-*a* maximum of 10–20 μg $L^{-1}$ has been found at the peak of the diatom bloom in March of every year (Odate et al. 1993; Kudo and Matsunaga 1999; Kudo et al. 2007). We believe that we observed the peak of the diatom spring bloom on 4 March. Even below the dark-layer depth (17 m), there were high chl-*a* concentrations (6.0–14.1 μg $L^{-1}$). The high chl-*a* levels in the dark layer suggested that large amounts of diatom aggregates, which had been produced in the surface layer, were sinking from the surface to the deeper layer.

**15 March** Chl-*a* concentrations had decreased at all depths by 15 March; however, high levels were still found within the surface mixed layer (0–10 m, 11.0–16.2 μg $L^{-1}$) and in the deeper dark layer (30–95 m, 2.3–7.8 μg $L^{-1}$). We considered the spring diatom bloom to be in a declining phase on 15 March.

**14 April** On 14 April the chl-*a* concentrations were very low in the euphotic zone (0–40 m, 0.05–1.6 μg $L^{-1}$) and in the deeper layer (50–85 m, 0.85–4.6 μg $L^{-1}$) except for the bottom water just above the sea floor (95 m, 12.2 μg $L^{-1}$). We believe

that the spring diatom bloom had terminated by 14 April. A local chl-*a* maximum concentration of 4.6 µg $L^{-1}$ was found at 50-m depth, which was between the euphotic-zone depth (42 m) and the dark-layer depth (55 m).

### 3.2.2 Nitrate

**15 February** On 15 February, nitrate ($NO_3^-$) concentrations (8.4—9.2 µmol $L^{-1}$) were vertically uniform (Fig. 4c; Fig. 5a).

**4 March** By 4 March, the $NO_3^-$ concentrations had substantially decreased to 0.15–0.60 µmol $L^{-1}$ in the depth range of 0–10 m within the euphotic zone. The decrease in the surface was due to consumption by diatoms, which had rapidly grown during the spring bloom. Below the surface, the $NO_3^-$ concentrations had also decreased, to 1.9–6.0 µmol $L^{-1}$ (20–95 m), which was approximately half of the concentrations in February. Because the dark-layer depth was 60 m on 15 February and 18 m on 4 March, the depth range of 60–95 m was dark on both dates. Thus, there could have been no photosynthesis-related $NO_3^-$ consumption in the dark layer (60–95 m). However, $NO_3^-$ concentrations in the dark layer had decreased, to 4.1 – 5.8 µmol $L^{-1}$, which was 60% of the concentrations in February. Explanations for the decrease in $NO_3^-$ concentrations in the dark layer are discussed in section 4.1.

**15 March** On 15 March, the $NO_3^-$ levels at 0–10-m depth (0.39–0.79 µmol $L^{-1}$) within the surface mixed layer (0–18 m) had not changed since 4 March. The euphotic-zone depth (1% PAR) and dark-layer depth (0.1% PAR) had deepened to 22 m and 30 m, respectively. The $NO_3^-$ concentrations in the dark subsurface layer (30–50 m) had decreased substantially, to 1.6–3.6 µmol $L^{-1}$, approximately half those on 4 March at the same depth range. The range of reduction rate of $NO_3^-$ per unit chl-a in the subsurface layer between 4 March and 15 March was 0.016 – 0.029 µmol (µg chl-*a*)$^{-1}$ d$^{-1}$. The reduction rate per unit chl-a will be compared with the dark incubation results in section 3.3. We hypothesized that the diatoms that had settled from the surface to the subsurface layer consumed $NO_3^-$ in the dark. This possibility will be discussed in section 4.2. The $NO_3^-$ concentrations in the deeper layer (60–95 m) had not changed since 4 March.

**14 April** By 14 April, the euphotic-zone depth had deepened to 47 m. The $NO_3^-$ concentrations had decreased to below the detection limit (<0.05 µmol $L^{-1}$) in the upper euphotic zone (0–30 m) and decreased to 1.4 µmol $L^{-1}$ in the lower euphotic zone (40 m). In the deep water (80–95 m), the $NO_3^-$ concentrations slightly increased from 5.26 µmol $L^{-1}$ on 15 March to 6.60 µmol $L^{-1}$ on 14 April.

### 3.2.3 Phosphate and silicate

Overall, temporal variations in $PO_4^{3-}$ and $Si(OH)_4$ concentrations were very similar to those of $NO_3^-$ (Fig. 4d,e; Fig. 5b,d). We found decreases in these nutrients in the dark subsurface layer of 60–95 m on 4 March, and 30–50 m on 15 March. On 4 March, the concentrations of $PO_4^{3-}$ and $Si(OH)_4$ in the dark subsurface layer (30–50 m) were 0.66 µmol $L^{-1}$ and 12.4 µmol $L^{-1}$, respectively, decreasing to 0.53 µmol $L^{-1}$ and 8.7 µmol $L^{-1}$ on 15 March. We concluded that the reasons for these decreases were the same as for $NO_3^-$ as discussed in sections 4.1 and 4.2. The ranges of reduction rate per unit chl-a in the subsurface layer between 4 and 15 March were 0.001 – 0.002 µmol (µg chl-*a*)$^{-1}$ d$^{-1}$ for $PO_4^{3-}$ and 0.029 – 0.053 µmol (µg chl-*a*)$^{-1}$ d$^{-1}$ for $Si(OH)_4$. The reduction rate per unit chl-a will be compared with the dark incubation results in section 3.3. In

contrast to the subsurface layer, the average concentrations of $PO_4^{3-}$ and $Si(OH)_4$ in the deep layer (80 – 95-m depth) increased with time; 0.78 µmol $L^{-1}$ and 15.3 µmol $L^{-1}$ on 4 March, 0.89 µmol $L^{-1}$ and 22.3 µmol $L^{-1}$ on 15 March, and 1.57 µmol $L^{-1}$ and 25.1 µmol $L^{-1}$ on 14 April, respectively. A clear increase of AOU in the deep water (80 – 95-m depth) was found from 15 March (average 20.9 µmol $L^{-1}$) to 14 April (average 56.0 µmol $L^{-1}$), see Fig. 4b and Fig. 5f. Because the increase of $PO_4^{3-}$ coincided with the rise in AOU, it likely resulted from remineralization following the decomposition of organic matter suspended in the bottom water or settled on the seafloor. The increase of $Si(OH)_4$ in the bottom water is likely resulted from dissolution of biogenic silica settled on the seafloor.

### 3.2.4 Ammonium

Temporal variations in $NH_4^+$ concentrations were similar to those of the other nutrients, except for substantial increases of $NH_4^+$ in the subsurface and bottom layers on 15 March (Fig. 4f; Fig. 5c). We considered these increases to be due to remineralization of organic matter suspended in the water column or settled on the seafloor. Because the $NH_4^+$ concentrations were at their lowest during winter with total column average of 0.25 µmol $L^{-1}$ on 15 February, the signal from remineralization could be clearly detected on 15 March with average of 0.54 µmol $L^{-1}$ at the subsurface water (30 – 50 m). The deep water $NH_4^+$ concentrations obviously increased with time since 4 March: 0.31 µmol $L^{-1}$ on 4 March, 0.95 µmol $L^{-1}$ on 15 March, and 3.05 µmol $L^{-1}$ on 14 April.

### 3.3 Limiting factor of primary production during the bloom

On 15 February before the occurrence of massive diatom bloom, the average concentrations of $NO_3^-$, $PO_4^{3-}$ and $Si(OH)_4$ at the surface 0 – 10 m were 9.1, 0.86 and 19.8 µmol $L^{-1}$, respectively. On 4 March at the peak of the bloom, the average concentrations of $NO_3^-$, $PO_4^{3-}$ and $Si(OH)_4$ at the surface 0 – 10 m were 0.34, 0.43 and 5.6 µmol $L^{-1}$, respectively. Uptake ratios of N:P and Si:N at the surface between 15 February and 4 March were 20.5 {= (9.1 – 0.34) / (0.86 – 0.43)} and 1.62 {= (19.9 – 5.6) / (9.1 – 0.34)}, respectively. Similar uptake ratios during diatom bloom in Funka Bay have been reported to be N:P = 15.6 – 23.6 and Si:N = 1.9 – 2.7 (Kudo and Matsunaga 1999). From the uptake ratio of N:P, $NO_3^-$ in the surface water could have been depleted since 4 March. On 15 March at the decline phase of the bloom, the average concentrations of $NO_3^-$, $PO_4^{3-}$ and $Si(OH)_4$ at the surface 0 – 10 m were 0.54, 0.37 and 4.5 µmol $L^{-1}$, respectively. Since sufficient amount of $Si(OH)_4$ remained in the surface water on 15 March, we considered that the N-depletion in the surface water limited primary production after the peak of the bloom.

### 3.3 Dark incubation experiments

We show temporal changes in nutrients and chl-a in the dark incubation experiments using the diatom *Thalassiosira nordenskioeldii* (Fig. 6 for the first and second experiments and Fig. 7 for the third and fourth experiments) and the natural seawater (Fig. 8).

### 3.3.1 *Thalassiosira nordenskioeldii* incubation experiment

In the first *Thalassiosira* experiment, we added nutrients into the nutrient-depleted culture medium. The added amount of $NO_3^-$ per unit chl-a ($0.022 = 31.1$ µmol $L^{-1}$ / 1426 µg $L^{-1}$) was 6% of the concentration ratio of $NO_3^-$/chl-a ($0.40 = 4.8$ µmol $L^{-1}$ / 12 µg $L^{-1}$) in seawater at 40 m on 4 March. The first incubation experiment results demonstrated that the diatom culture, which had been depleted in nutrients before the start of the dark incubation, rapidly exhausted nutrients in the dark within six days after the nutrient addition (Fig. 6a-c). Since we did not check the bacterial contamination after the experiment, bacterial consumption and/or recycling of nutrients in the culture might influence the results. We assumed that the bacterial activity had a less effect on nutrient changes in the high-density diatom culture. The daily consumption rates per unit chl-a amount calculated from the concentration difference of nutrients between day 0 and day 2 and the initial concentration of chl-a (1426 µg $L^{-1}$) of the dark incubation were 0.0084 µmol (µg chl-*a*)$^{-1}$ d$^{-1}$ for $NO_3^-$, 0.00036 µmol (µg chl-*a*)$^{-1}$ d$^{-1}$ for $PO_4^{3-}$, and 0.0015 µmol (µg chl-*a*)$^{-1}$ d$^{-1}$ for $Si(OH)_4$. From these consumption rates, we estimated the concentration decreases ($\Delta$Nutrients) in the subsurface layer for the 11 days between 4 and 15 March using an observed chl-*a* concentration of 11.54 µg $L^{-1}$ (average on 4 March). The estimated $\Delta NO_3^-$ ($-1.1$ µmol $L^{-1}$), $\Delta PO_4^{3-}$ ($-0.05$ µmol $L^{-1}$), and $\Delta Si(OH)_4$ ($-0.18$ µmol $L^{-1}$) were 1/2–1/20 of the actual decreases in the subsurface layer between 4 and 15 March: $\Delta NO_3^-$, $-2.0$ µmol $L^{-1}$; $\Delta PO_4^{3-}$, $-0.12$ µmol $L^{-1}$; $\Delta Si(OH)_4$, $-3.7$ µmol $L^{-1}$. Silicic acid was almost depleted on day 2 of the dark incubation. If the diatoms exhausted $Si(OH)_4$ earlier than we collected the culture sample on day 2, the daily consumption rates are underestimated.

In the second *Thalassiosira* experiment, we added excess amount of nutrients into the nutrient-depleted medium, in which cultured diatoms were in a decline phase of growth. The added amount of $NO_3^-$ per unit chl-a ($10.3 = 743.5$ µmol $L^{-1}$ / 72.5 µg $L^{-1}$) was 26 times of the seawater concentration ratio at 40 m on 4 March 2019. The results of the second incubation experiment (Fig. 6d-f) demonstrated that the diatom culture consumed nutrients in the dark. The daily consumption rates were 0.11 µmol (µg chl-*a*)$^{-1}$ d$^{-1}$ for $NO_3^-$, 0.053 µmol (µg chl-*a*)$^{-1}$ d$^{-1}$ for $PO_4^{3-}$, and 0.41 µmol (µg chl-*a*)$^{-1}$ d$^{-1}$ for $Si(OH)_4$, 12–173 times those of the first experiment. We deduced that the high consumption rates per unit chl-a in the second experiment were due to high consumption rates per unit cell, because chl-a was low, although we did not count the cell number density. The estimated $\Delta NO_3^-$ ($-13.9$ µmol $L^{-1}$), $\Delta PO_4^{3-}$ ($-6.74$ µmol $L^{-1}$), and $\Delta Si(OH)_4$ ($-51.5$ µmol $L^{-1}$) were 7 – 56 times of the actual decreases in the subsurface layer.

In the third and fourth *Thalassiosira* experiments, the added amount of $NO_3^-$ per unit chl-a ($0.77 – 0.96$) was $1.9 – 2.4$ times of the seawater concentration ratio at 40 m on 4 March. The results of the third and fourth experiments (Fig. 7a-h) demonstrated that the diatom culture consumed nutrients in the dark. The consumption rates, which were calculated from the concentration difference of nutrient between day 0 and day 3 of the dark incubation and the initial chl-a concentrations on day 0, were $0.034 – 0.043$ µmol (µg chl-*a*)$^{-1}$ d$^{-1}$ for $NO_3^-$, $0.0059 – 0.0086$ µmol (µg chl-*a*)$^{-1}$ d$^{-1}$ for $PO_4^{3-}$, and $0.034 – 0.035$ µmol (µg chl-*a*)$^{-1}$ d$^{-1}$ for $Si(OH)_4$. The estimated $\Delta NO_3^-$ from $-4.3$ to $-5.4$ µmol $L^{-1}$, $\Delta PO_4^{3-}$ from $-0.75$ to $-1.1$ µmol $L^{-1}$, and $\Delta Si(OH)_4$ from $-4.3$ to $-4.4$ µmol $L^{-1}$ were close to the actual decreases between the two dates: $\Delta NO_3^-$, $-2.0$ µmol $L^{-1}$;

$\Delta PO_4^{3-}$, $-0.12$ $\mu mol$ $L^{-1}$; $\Delta Si(OH)_4$, $-3.7$ $\mu mol$ $L^{-1}$. The chl-a concentrations were increased in the dark from 145 $\mu g$ $L^{-1}$ (day 0) to 250 $\mu g$ $L^{-1}$ (day 3) for the third experiment and from 198 $\mu g$ $L^{-1}$ (day 0) to 294 $\mu g$ $L^{-1}$ (day 3) for the fourth experiment.

### 3.3.2 Natural seawater incubation experiment

In the natural seawater incubation experiment using seawater samples collected on 8 March 2022, the added amount of $NO_3^-$ per unit chl-a (0.50 = 12.6 $\mu mol$ $L^{-1}$ / 25.1 $\mu g$ $L^{-1}$) into nutrient-depleted seawater was between the concentration ratio of $NO_3^-$/chl-a (0.33 = 4.9 $\mu mol$ $L^{-1}$ / 14.5 $\mu g$ $L^{-1}$) in seawater at 5m and the ratio of 0.81 (= 7.4 $\mu mol$ $L^{-1}$ / 9.1 $\mu g$ $L^{-1}$) at 40 m on 8 March 2022. The results of the natural seawater experiment, in which nutrients were added into the nutrient-depleted seawater after the pre-culturing under light condition, are shown in Fig. 8a-d. We found decreases in nutrient concentrations in the dark period after the nutrient addition. The consumption rates, which were calculated from the concentration difference of nutrient between day 0 and day 7 of dark incubation and the initial chl-a concentration on day 0, were 0.053 $\mu mol$ $(\mu g$ chl-$a)^{-1}$ $d^{-1}$ for $NO_3^-$, 0.0018 $\mu mol$ $(\mu g$ chl-$a)^{-1}$ $d^{-1}$ for $PO_4^{3-}$, and 0.010 $\mu mol$ $(\mu g$ chl-$a)^{-1}$ $d^{-1}$ for $Si(OH)_4$. The consumption rates were close to the rates found at the dark-subsurface layer (30 – 50 m) between 4 and 15 March 2019. The estimated $\Delta NO_3^-$ ($-6.7$ $\mu mol$ $L^{-1}$), $\Delta PO_4^{3-}$ ($-0.23$ $\mu mol$ $L^{-1}$), and $\Delta Si(OH)_4$ ($-12.9$ $\mu mol$ $L^{-1}$) were greater than the nutrient decreases at the dark subsurface layer between 4 and 15 March 2019. However, results of the natural seawater experiment under continuous dark condition without nutrient-addition (Fig. 8e-h) demonstrated that all nutrients were not consumed by phytoplankton in the dark. We considered that phytoplankton collected from natural seawater on 8 March 2022 did not require nutrients under the continuous dark condition, because there was enough $NO_3^-$ (4.6 $\mu mol$ $L^{-1}$) when the water at 5-m depth was collected in the growing phase of bloom.

### 4 Discussions

### 4.1 Nitrate consumption in the dark layer between 15 February and 4 March

We propose two explanations for the decrease in $NO_3^-$ concentrations in the dark layer (60–95 m) between 15 February and 4 March. The first possible explanation is that $NO_3^-$ was consumed by diatoms during their growth in the upper euphotic zone after 15 February, and then this water, in which $NO_3^-$ had been consumed during photosynthesis, mixed vertically with the deeper water (60–95 m) through winter cooling before the water became stratified by 4 March. The possibility of vertical mixing between these two dates is supported by the temporal variation of AOU, which is defined as the difference between the equilibrium saturation concentration and the measured concentration of oxygen in the water; i.e., positive and negative AOUs suggest net consumption (respiration) or net production (photosynthesis) of oxygen in water, respectively. On 15 February, the absolute value of AOU over most of the water column was the lowest during the observation period (average, 6.2 $\mu mol$ $L^{-1}$; range, $-3$ to 23 $\mu mol$ $L^{-1}$), suggesting that there was no significant net $O_2$ production throughout the total water column. By 4 March, AOU values had dropped to $-14$ to $-94$ $\mu mol$ $L^{-1}$ at 0–50 m, and to $-7$ to $-42$ $\mu mol$ $L^{-1}$ at 60–95 m, even in the dark. Note that the decrease of water temperature between the two dates ($\Delta temp = -0.45$ °C) could have caused

an increase in AOU ($\Delta$AOU = +3.6 µmol L$^{-1}$) due to the increase in the solubility of oxygen. The large negative AOU in the euphotic zone on 4 March was apparently due to photosynthetic O$_2$ production. The negative AOU in the deeper dark layer (60–95 m) on 4 March was thought to be due to mixing with surface water, in which AOU had been lowered by photosynthesis, before the water became stratified by 4 March.

The second possible explanation is that the diatoms, which had grown at the surface and then settled to the deeper layer, consumed NO$_3^-$ in the dark without photosynthetic growth. The possibility of nutrient consumption in the dark is discussed in section 4.2.1. We believe that both explanations apply to the decrease in NO$_3^-$ concentrations in the deeper layer between these two dates; however, we could not separate their effects.

## 4.2 Nutrient consumption in the dark subsurface layer between 4 and 15 March

We found decreases in the nutrient concentrations in the dark subsurface layer between 15 February and 4 March, and between 4 March and 15 March. The latter reduction could not have been affected by photosynthetic consumption by diatoms because there was almost no light available for photosynthesis. Here we discuss the possible reasons for the nutrient reductions between 4 and 15 March.

### 4.2.1 Nutrient consumption by diatoms in the dark

First, we considered that a possible explanation for nutrient reduction in the subsurface layer was consumption by diatoms that were sinking from the surface and suspended in the dark subsurface layer during the bloom. To examine this possibility, we conducted dark incubation experiments using the diatom *Thalassiosira nordenskioeldii*, which dominates during the spring bloom in Funka Bay (Ban et al. 2000). From microscopic image analysis, *Thalassiosira nordenskioeldii* occupied 14.2% of number of phytoplankton cells (n = 1209) collected by plankton net (mesh = 100 µm) on 15 March 2019. Other dominant species were *Chaetoceros* spp. and other *Thalassiosira* sp. We confirmed that *Thalassiosira nordenskioeldii* was one of the dominant species in the spring bloom 2019. From the results of dark incubation experiments using *Thalassiosira nordenskioeldii* and natural seawater collected on 8 March 2022, we demonstrated that the amount of dark consumption of nutrients by diatoms had a potential to cause the nutrient reduction at the dark subsurface layer.

Cochlan et al. (1991) carried out onboard incubations with a diatom dominating natural seawater setting dark periods of 2–4 hours after light periods. They have reported dark consumption rates for NO$_3^-$ of 0.09–0.14 µmol (µg chl-*a*)$^{-1}$ d$^{-1}$, which are close to the results from our second Thalassiosira incubation. Many previous studies have focused on the dark consumption within the day–night cycle in the euphotic zone. Onboard simulated in-situ incubations yielded dark:light ratios of NO$_3^-$ consumption rates of 0–0.67 (Nelson and Conway 1979), 0–1.0 (Conway and Whitledge 1979), and 0–0.51 (Cochlan et al. 1991). These previous works have reported wide ranges of dark consumption rates and ratios. Cochlan et al. (1991) reported that the dark:light uptake ratio was greater in N-impoverished waters than in N-replete waters, suggesting that dark uptake is

enhanced by nutrient stress. They also mention the importance of N uptake by heterotrophic bacteria, citing studies where uptake by heterotrophic bacteria ranges from half the uptake by phytoplankton to half of the total N uptake.

In the dark subsurface layer of Funka Bay between 4 and 15 March 2019, N-depleted diatoms sunk from the surface after the peak of bloom could have enhanced $NO_3^-$ consumption in the dark. However, N-repleted diatoms in growing phase of bloom on 8 March 2022 would not have a potential to consume nutrients in the continuous dark condition (see, Fig. 8e-h). Although the dark consumption rates had wide ranges, we concluded that dark consumption by diatoms after the peak of bloom had a potential to reduce nutrients by half in the dark subsurface layer of Funka Bay.

### 4.2.2 Water mixing as a possible explanation for nutrient reduction

Second, we discuss the possibility that vertical mixing between the surface mixed layer, which already had reduced nutrient levels, and the subsurface layer (30–50 m) resulted in the observed decrease in nutrient concentrations in the subsurface layer. Because the density ($\sigma$) gradient between 20-m and 30-m depths, ($\sigma_{30m} - \sigma_{20m}$) / (30 m – 20 m), 0.0033 (kg m$^{-3}$ m$^{-1}$) on 4 March substantially increased to 0.021 (kg m$^{-3}$ m$^{-1}$) on 15 March, the stratification between these layers had strengthened (Fig. 2c). Additionally, there was no bad weather during this period; the wind speeds were relatively low, with daily averages of 3.0–5.9 m s$^{-1}$, although low air temperature lasted, with daily averages of 1.0 – 5.3 °C (data from Muroran Observatory, Meteorological Agency of Japan; https://www.data.jma.go.jp/risk/obsdl/index.php). The effect of low salinity water inflow on the density decrease at the surface layer overcame the effect of cooling on density increase (Fig. 2). From these observations, we excluded the possibility of vertical mixing between the two layers as an explanation for the decrease in nutrients in the subsurface layer.

We also considered the possibility of horizontal mixing of subsurface water with Oyashio water entering from the surface of the bay. Oyashio water is characterized by its low salinity. Because the salinity at 30 m declined from 33.58 on 4 March to 33.47 on 15 March, it is possible that the influence of low-salinity Oyashio water extended to 30 m. Since the salinity of Oyashio water at 10-m depth on 15 March, where the minimum temperature (2.6 °C) was found suggesting an appearance of the main body of Oyashio water, was 33.0, we assumed that the salinity of original Oyashio water was 33.0. A mixing between 20% of Oyashio water and 80% of Funka Bay water at 30 m would change the salinity at 30-m depth from 33.58 (on 4 March) to 33.47 (on 15 March). Even if the concentrations of nutrients in the original Oyashio water were 0 μmol L$^{-1}$, the mixing ratio of 8:2 (Funka-Bay water:Oyashio water) would reduce the nutrient concentrations at 30-m depth by only 20%. In reality, the $NO_3^-$ concentration at 30 m was decreased by half between these two dates (Fig. 5a), and the salinities at 40- and 50-m depths did not change (Fig. 2b). Thus, we excluded the possibility of mixing with Oyashio water as a reason for nutrient reduction in the subsurface layer.

### 4.2.3 Diffusive transport between the surface and the subsurface layers

Third, we discuss an effect of diffusive transport of $NO_3^-$ on concentration decrease at the subsurface layer (30 – 50 m) on 15 March. Diffusive coefficients ($K\rho$) have not been measured in Funka Bay. We referred a range of $K\rho$ ($= 10^{-6} - 10^{-5}$ m$^2$ s$^{-1}$) measured just below the mixed layer ($\sim$ 30 m) at the western subarctic Pacific in summer (Dobashi et al. 2021).

Concentration gradients of $NO_3^-$ were –0.000221 μmol m$^{-4}$ ($= \Delta NO_3^-{}_{20m-30m}$ / 10 m), –0.000141 μmol m$^{-4}$ ($= \Delta NO_3^-{}_{30m-40m}$ / 10 m), –0.000115 μmol m$^{-4}$ ($= \Delta NO_3^-{}_{40m-50m}$ / 10 m), and –0.0000135 μmol m$^{-4}$ ($= \Delta NO_3^-{}_{50m-60m}$ / 10 m). The range of diffusive transport of $NO_3^-$ were calculated to be 0.00022 – 0.0022 μmol m$^{-2}$ s$^{-1}$ between 20 m and 30 m, which could result in concentration change of 0.021–0.21 μmol L$^{-1}$ at 30 m for 11 days. Concentration changes between 30 m and 40 m and between 40 m and 50 m were calculated to be 0.013–0.13 μmol L$^{-1}$ and 0.011 – 0.11 μmol L$^{-1}$, respectively. The sum of

concentration changes at 30 m, which include transports from 20 m layer and 40 m layer, ranges from –0.20 μmol L$^{-1}$ ($= -0.21 + 0.013$) to +0.11 μmol L$^{-1}$ ($= -0.021 + 0.13$). Ranges of the sum of concentration changes at 40 m and 50 m were from –0.12 to +0.096 μmol L$^{-1}$ and from –0.11 to –0.024 μmol L$^{-1}$, respectively. The observed decreases were of 1.6 μmol L$^{-1}$ at 30 m, 2.0 μmol L$^{-1}$ at 40 m, and 2.4 μmol L$^{-1}$ at 50 m between these dates. Thus, we concluded that diffusive transport of $NO_3^-$ had a minor effect on the concentration decreases at the subsurface layer.

### 4.2.4 Subduction of surface water into the subsurface layer

Fourth, we discuss a possibility if subduction of surface water caused the decrease in nutrient concentrations at the subsurface layer (30 – 50 m) of the observation station 30. At the medium depth (40 m) of the subsurface layer, temperature, salinity, and density were 3.5 – 3.6 °C, 33.64, and 26.7σ, respectively, on 4 and 15 March (Fig. 2). Suppose surface water in certain area of the bay subducted and it reached 40-m depth at the observation station on 15 March, the subducted water

should have the same temperature, salinity, and density as it had been at the surface. The average current speed at 40-m depth between these dates was 3.3 cm s$^{-1}$ (unpublish data), which was obtained from acoustic doppler current profiler (ADCP) set on the sea floor at the station. The middle layer water at the station could have reached from anywhere of the bay within 11 days. We obtained the spatial distributions of temperature, salinity, and density at the sea surface (1 m) on 4 March using the ocean reanalysis product provided by Meteorological Research Institute in Japan (Fig. 9a-c).

From these spatial distributions, there was not any area that satisfied required temperature (3.5 – 3.6 °C), salinity (33.64), and density (26.7σ) to form subduction water at 40-m depth of the observation station, see an enlarged map of Fig. 9c. We considered that the subsurface layer water at the station was not associated with subduction. Note that the modelled vertical profiles of temperature, salinity and density were not in good agreement with the observed profiles (Fig. s3). In the model result, the influence of low salinity Oyashio water, which flowed in the surface of Funka Bay, was estimated to be stronger

than the observational result. If the influence of Oyashio water had been as strong as the model result, the subsurface water (30 – 50 m) might have changed between 4 and 15 March 2019. We considered that the subsurface water had stayed between both dates because of weak Oyashio water inflow.

From the above discussions, we concluded that nutrient reductions in the dark subsurface layer between 4 and 15 March were mainly caused by consumption by diatoms in the dark.

## 4.3 The influence of nutrient consumption by diatoms in the dark subsurface layer

Nutrient uptake by diatoms in the dark subsurface layer after the peak of bloom would have impacts on primary production and distribution of phytoplankton in bloom and post-bloom. We propose alternative hypotheses (1 and 2) to deduce the influence of nutrient uptake in the dark subsurface layer.

1) If the diatom population that had consumed half of the nutrients in the dark subsurface layer sank to the deeper layer during the bloom, then the primary production in the subsurface layer after the bloom, at which time it would be part of the euphotic zone, would be reduced by half at maximum compared to the production in the case where there was no nutrient consumption during the dark period.

2) If the diatoms that had consumed nutrients in the dark subsurface layer remained in that layer after the bloom or migrated to the upper layer, they have a potential to rapidly grow under the returning light conditions when the euphotic zone deepened after the bloom.

In the case of Funka Bay, we note that the consumption of nutrients in the dark subsurface layer would have an impact outside the bay, because the subsurface water was exchanged with Oyashio water.

In relation to the second hypothesis, an interesting survival strategy for diatom, *Rhizosolenia*, which forms large aggregations (mats), has been proposed (Villareal et al. 1996; Richardson et al. 1998; Villareal et al. 1999; Villareal et al. 2014). The survival strategy of *Rhizosolenia* is that they consume $NO_3^-$ in the dark subsurface layer, and then migrate to the euphotic zone where they have a growth advantage in oligotrophic subtropical open ocean areas. The coastal marine diatom, *Thalassiosira weissflogii,* was studied to examine changes in buoyancy in relation to ratios of carbohydrate to protein which determine the cell density (Richardson and Cullen, 1995). That study revealed that accumulation of carbohydrate as a result of nitrate depletion leads rises in cellular density and sinking speed and that accumulation of protein as a result of nitrate addition after the nitrate depletion leads a positive buoyancy. A modelling study estimated that vertically migrating phytoplankton contributes 7% of net primary production at the subarctic gyre of the western Pacific (Witz and Lan Smith, 2020).

These previous studies (Villareal et al. 1996; Richardson et al. 1998; Villareal et al. 1999) have not yet found any evidence of decrease in $NO_3^-$ in the dark subsurface layer from observation. If the hypothesis of diatoms' migration strategy proposed

by previous studies is true, the results of our study will provide evidence for the decrease in $NO_3^-$ in the dark subsurface layer associated with the diatoms' strategy. As for the reduction in $Si(OH)_4$ concentrations found in the dark subsurface layer of the Barents Sea, it has been suggested that diatoms settling from the surface consume $Si(OH)_4$ in the dark subsurface layer to form spores (Rey and Skjoldal 1987).

However, there is not yet sufficient observational data for a complete explanation. Further research is needed to examine these possible strategies and their impacts on biogeochemical cycles.

**5 Conclusions**

We conducted repetitive observations in Funka Bay, Hokkaido, Japan, from 15 February to 14 April 2019 during and after

the spring bloom. We found reductions in nutrient concentrations in the dark subsurface layer both before and after the peak of the bloom and concluded that the latter reduction was caused by dark consumption by diatoms that had grown in the euphotic zone and then sank to the dark subsurface layer. We reached this conclusion using the following rationale.

(1) From the dark incubation experiments, we confirmed that the diatom *Thalassiosira nordenskioeldii*, which is one of the

dominant diatom species in the bloom of Funka Bay, could consume added nutrients in the dark at substantial rates after preculturing to deplete nutrients and that phytoplankton in nutrient-depleted natural seawater collected in the bay before the peak of diatom bloom on 8 March 2022 could also consume added nutrients in the dark. Although the consumption rates varied over a wide range, we concluded that dark consumption of nutrients by diatoms in the dark subsurface layer had a potential to reduce nutrient by half in the dark subsurface layer (30–50 m).

(2) We excluded water mixing, diffusive transport, and subduction as possible reasons for nutrient reduction in the subsurface layer between 4 March and 15 March. First, the stratification between the surface and subsurface layers was strengthened after 4 March, and therefore we considered vertical mixing of water between the layers to be limited. The small decline in salinity at 30 m and no change in salinity at 40–50 m means that mixing with low-salinity Oyashio water could not explain the nutrient reduction, even if the Oyashio water had no nutrients. Second, we estimated the diffusive transport of

$NO_3^-$ to have a minor effect on concentration decrease at the subsurface layer. Third, we showed that there was not any area that satisfied required surface temperature, salinity, and density to form subduction water at the subsurface layer (medium depth of 40 m) at the observation station on 15 March 2019. Thus, we excluded the possibilities of subduction as the main reason for the nutrient reduction.

The consumption of nutrients in the dark has been studied in many simulated in-situ incubation experiments, with the goal of understanding dark consumption during a daily cycle within the euphotic zone. We believe that this is the first study to demonstrate observational evidence of consumption of the three main nutrients ($NO_3^-$, $PO_4^{3-}$, and $Si(OH)_4$) by diatoms in the dark subsurface layer during a bloom. This consumption could result in reduced new production in the subsurface layer after

the bloom, when this layer would once again become part of the euphotic zone, if the diatoms sank to deeper layers. Further research is needed examining the survival strategies of diatoms consuming nutrients in the dark subsurface layer.

**Competing interests.**

The authors declare that they have no conflict of interest.

**Author contributions.**

A.O. designed the research and conducted the observations. S.U. analysed the data. A.O. conducted the diatom incubations in the dark. Y.N. prepared the axenic diatom culture for the dark incubations. M.T., H.A. and D.N. supported the data analysis. H.A. provided the spatial map of temperature, salinity, and potential density. H.O. provided the ADCP data. H.O. and T.T. designed the *Ushio-maru* observations. A.O. and S.U. wrote the manuscript with contributions from all co-authors.

**Acknowledgments**

We thank the captains and crews of T/S *Ushio-maru* (Hokkaido University). This research was supported financially by FY2018–FY2019 Research Projects from the Hokusui Society Foundation, Sapporo, Japan, and the Japan Society for the Promotion of Science (JSPS) KAKENHI grant numbers 16H02929 and 16H01586. The first draft of this article was edited and reviewed by professional scientific editors (ELSS, Inc., Tsukuba, Japan).

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

Table 1 Setup conditions of dark incubation experiments. Dark incubations using *Thalassiosira nordenskioeldii* were carried out four times. The diatom cultures were grown under light condition (pre-culture) before it was put in the dark with nutrient addition. Natural seawater incubations were carried out using seawater collected at station 30 in Funka Bay on 8 March 2022.

| Incubation experiments | | | $NO_3^-$ | $PO_4^{3-}$ | $Si(OH)_4$ | Chl-a |
|---|---|---|---|---|---|---|
| | | | ($\mu mol\ L^{-1}$) | | | ($\mu g\ L^{-1}$) |
| *Thalassiosira* incubations | | | | | | |
| First | Pre-culture (17 days) | Initial | 700 | 26 | 75 | |
| | | Final | < 0.05 | < 0.05 | < 1 | |
| | Dark incubation (n = 1) Day 0 (Nutrient addition) | | 29.2 | 1.13 | 4.24 | 1426 |
| Second | Pre-culture (42 days) | Initial | 700 | 26 | 75 | |
| | | Final | < 0.05 | < 0.05 | < 1 | |
| | Dark incubation (n = 1) Day 0 (Nutrient addition) | | 744 | 27.6 | 113 | 72.5 |
| Third | Pre-culture (9 days) | Initial | 175 | 6.5 | 18.8 | |
| | | Final | 9.27 | 0.42 | < 0.1 | |
| | Dark incubation (n = 4) Day 0 (Nutrient addition) | | 187 | 7.66 | 16.1 | 145 |
| Fourth | Pre-culture (10days) | Initial | 175 | 6.5 | 18.8 | |
| | | Final | 0.66 | 0.42 | < 1 | |
| | Dark incubation (n = 4) Day 0 (Nutrient addition) | | 213 | 9.08 | 21.2 | 198 |
| Natural seawater incubations | | | | | | |
| Nutrient-deplete (5 m) with nutrient-deplete treatment Preculture (5days) Initial | | | 4.86 | 0.58 | 14.0 | 14.5 |
| | | Final | < 0.05 | < 0.05 | < 1 | 25.1 |
| | Dark incubation (n = 4) Day 0 (Nutrient addition) | | 12.6 | 0.38 | 17.8 | |
| | Dark incubation (n = 4) Day 0 (No-nutrient addition) | | < 0.05 | < 0.05 | < 1 | |
| Continuous dark (5 m / 40 m) with no nutrient-deplete treatment Dark incubation (n = 3) initial (No-nutrient addition) | | | (4.59 / 7.39) | (0.56 / 0.76) | (14.6 / 18.4) | (14.3 / 9.09) |

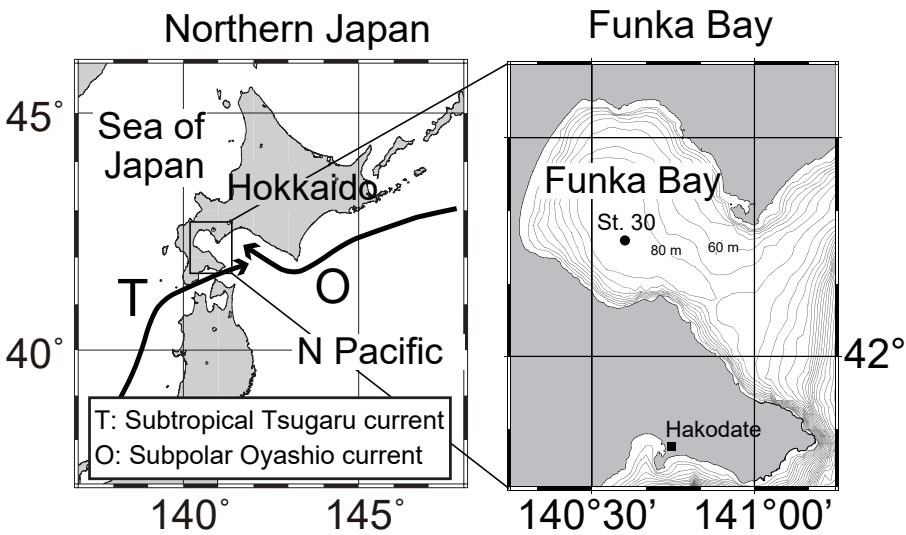

Fig. 1  Sampling sites in Funka Bay, Hokkaido, Japan

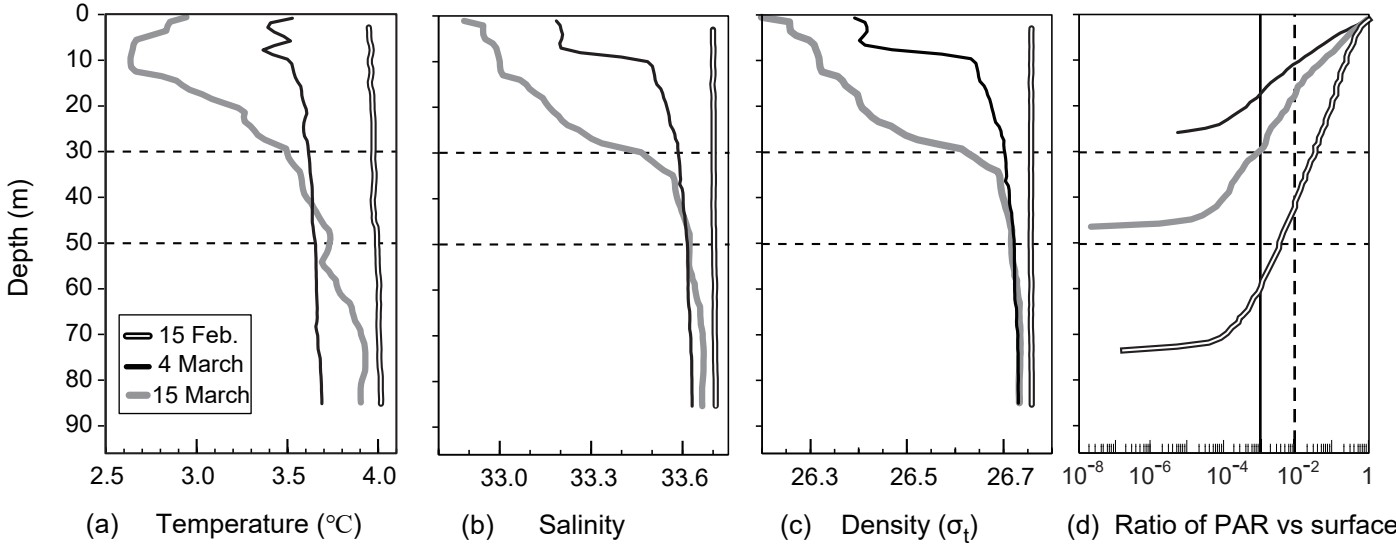

Fig. 2 Vertical profiles of temperature (a), salinity (b), density (c), and ratio of PAR vs surfae PAR (d)  at station 30 in Funka Bay, Japan, on 15 February, 4 March, and 15 March. Broken and solid vertical lines in Fig. 2d indicate the ratio of PAR at euphotic-zone depth (1% PAR vs surface) and  dark-layer depth (0.1% PAR vs surface), respectively.

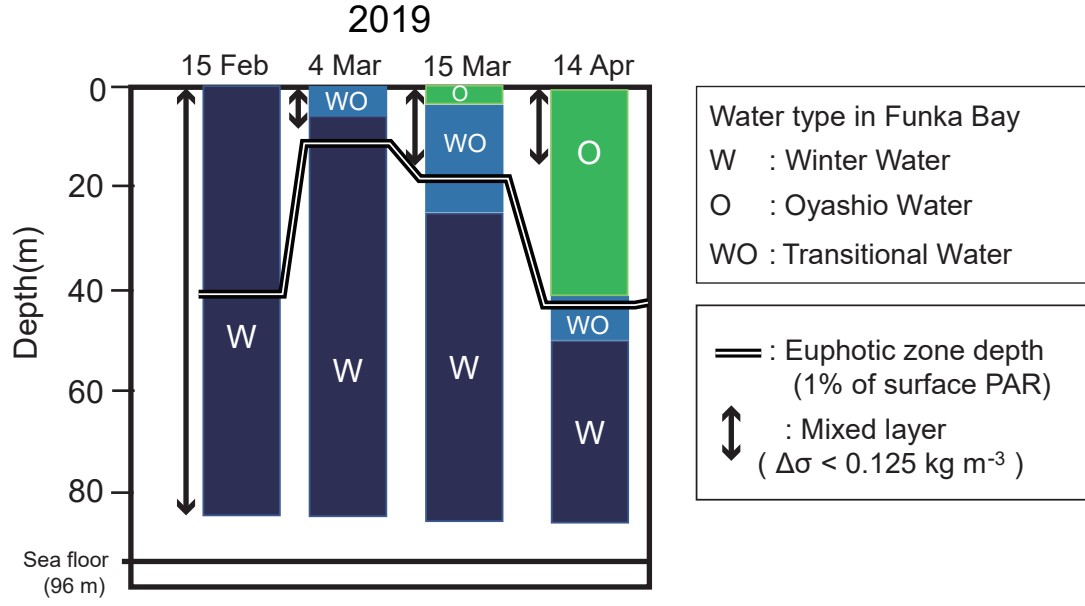

Fig. 3 Temporal change of water-mass structure at station 30 in Funka Bay, Japan. The two main water masses are winter water (W) and Oyashio water (O). Transitional water (WO) is a water changing from winter water to Oyashio water by mixing. Euphotic-zone depth and surface mixed-layer depth (MLD) are also shown. The euphotic-zone depth was defined as the depth at which photosynthetically active radiation (PAR) was 1% of the surface PAR. The MLD was defined as the layer in which density differences ($\Delta\sigma$) were within 0.125 kg m$^{-3}$ relative to the density at 5-m depth.

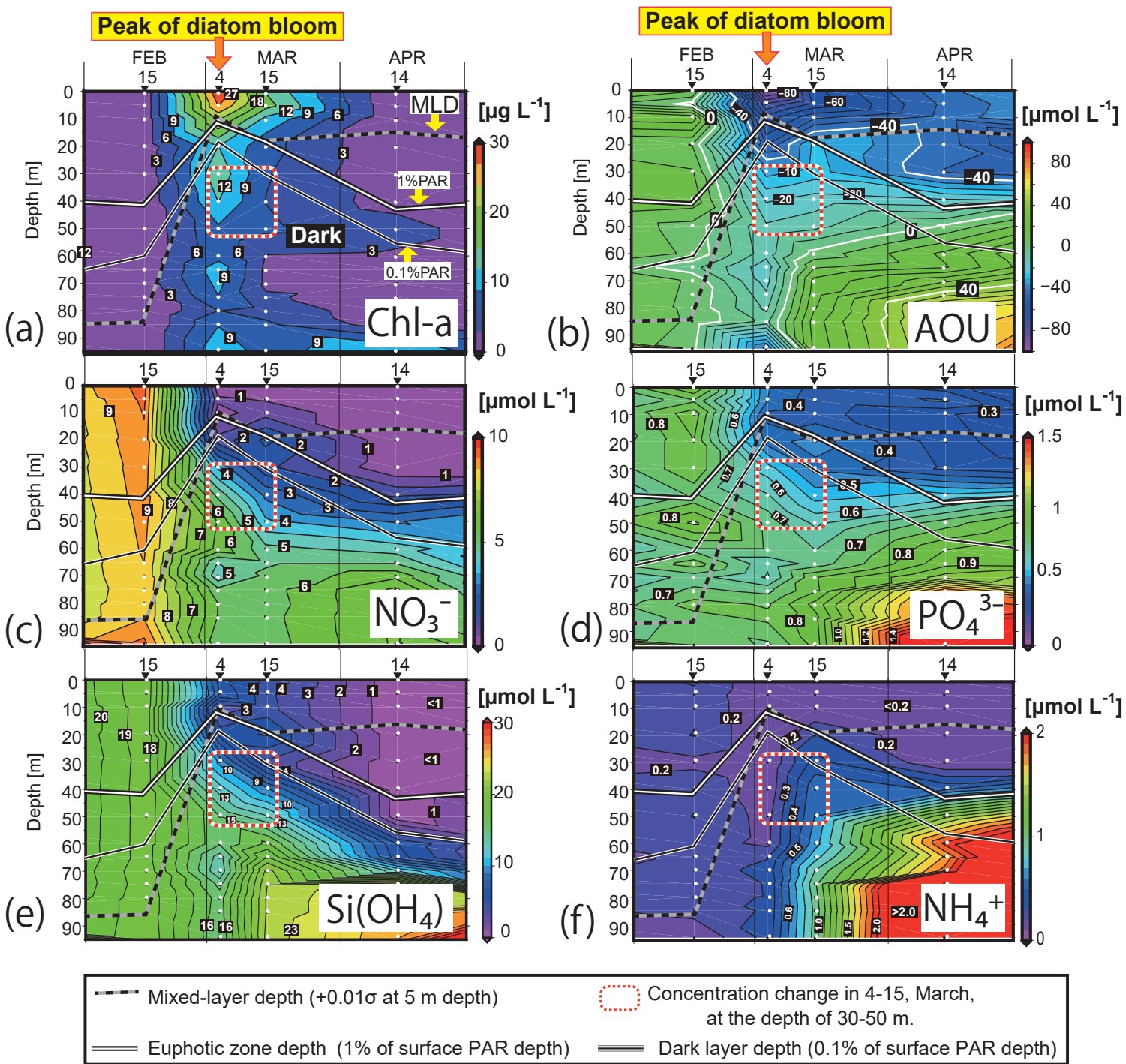

Fig. 4 Time–depth sections of chl-a concentration (a), apparent oxygen utilization (AOU) (b), and concentrations of $NO_3^-$ (c), $PO_4^{3-}$ (d), $Si(OH)_4$ (e), and $NH_4^+$ (f) in the water column in Funka Bay, Japan. Water was collected on 15 February, 4 and 15 March, and 14 April 2019; white circles indicate sampling depths. Solid white lines indicate the euphotic-zone depth (1% PAR). Solid black lines indicate the dark-layer depth (0.1% PAR). Black-and-white dotted lines indicate surface mixed-layer depth. Squares outlined with red-and-white dotted lines indicate the subsurface layer (30–50 m) on 4 and 15 March, where nutrient reductions were observed. PAR, photosynthetically active radiation.

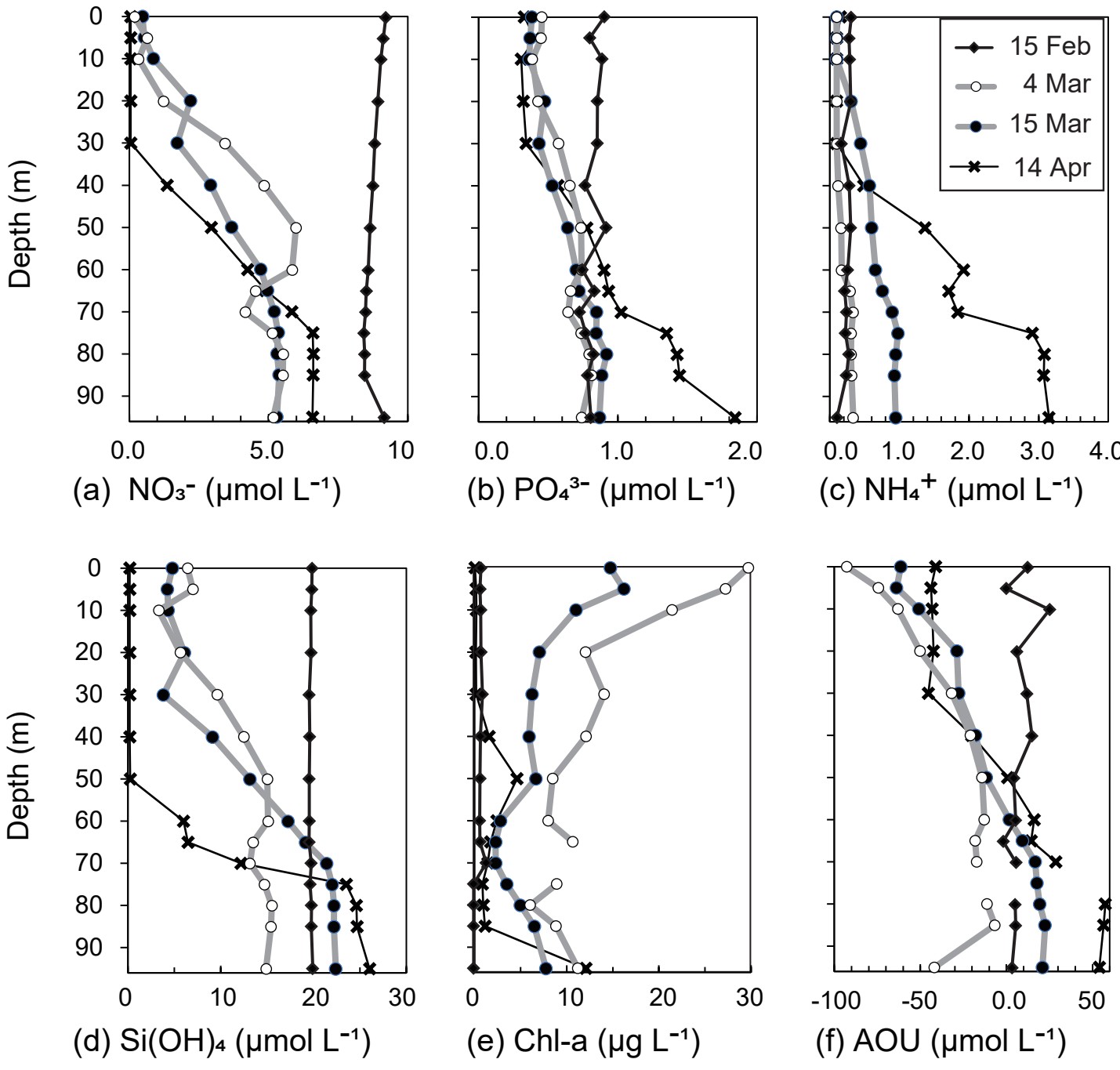

Fig. 5 Vertical profiles of NO₃⁻ (a), PO₄³⁻ (b), NH₄⁺ (c), Si(OH)₄ (d), Chl-a (e), and AOU (f) at station 30 in Funka Bay, Hokkaido, Japan, on 15 February, 4 and 15 March, and 14 April 2019.

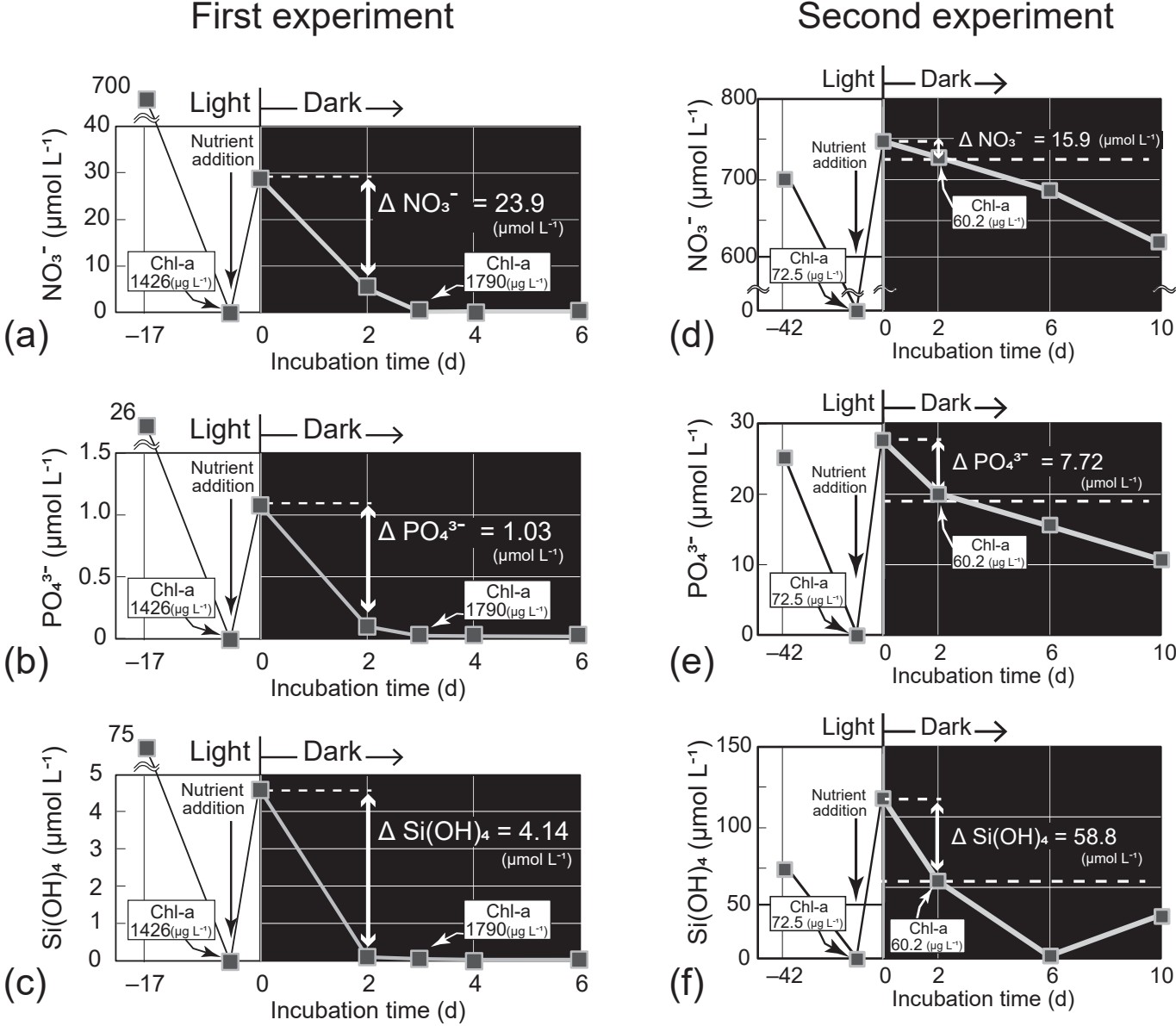

Fig. 6 Temporal change of the nutrient concentrations in the dark incubation experiment using the diatom Thalassiosira nordenskioeldii for $NO_3^-$ (a), $PO_4^{3-}$ (b), and $Si(OH)_4$ (c) of the first experiment and $NO_3^-$ (d), $PO_4^{3-}$ (e), and $Si(OH)_4$ (f) of the second experiment. The diatom was pre-cultured for 17 or 42 days under light conditions before nutrients were added. Each incubation bottle (n=1) with nutrient addition was put in darkness on day 0.

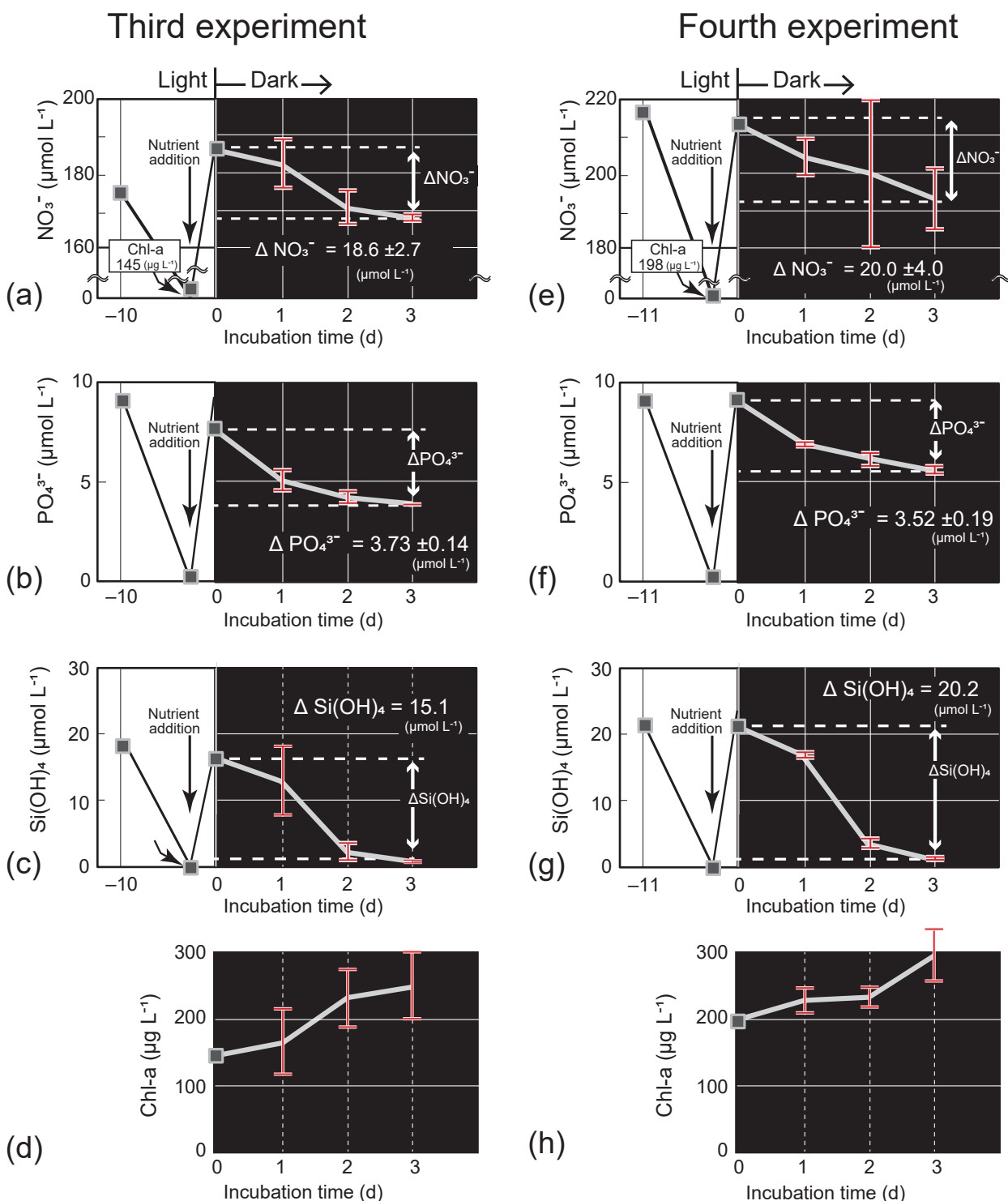

Fig. 7 Temporal change of the nutrient and chl-a concentrations (mean ± 1stdev, n = 4) in the dark incubation experiment using the diatom Thalassiosira nordenskioeldii for $NO_3^-$ (a), $PO_4^{3-}$ (b), $Si(OH)_4$ (c), and chl-a (d) of the third experimet and for $NO_3^-$ (e), $PO_4^{3-}$ (f), $Si(OH)_4$ (g), and chl-a (h) of the fourth experiments. The diatom was cultured for 10 or 11 days under light conditions before nutrients were added. Incubation bottles (n=4) with nutrient addition was put in darkness on day 0.

# Natural seawater (8 March 2022) experiment

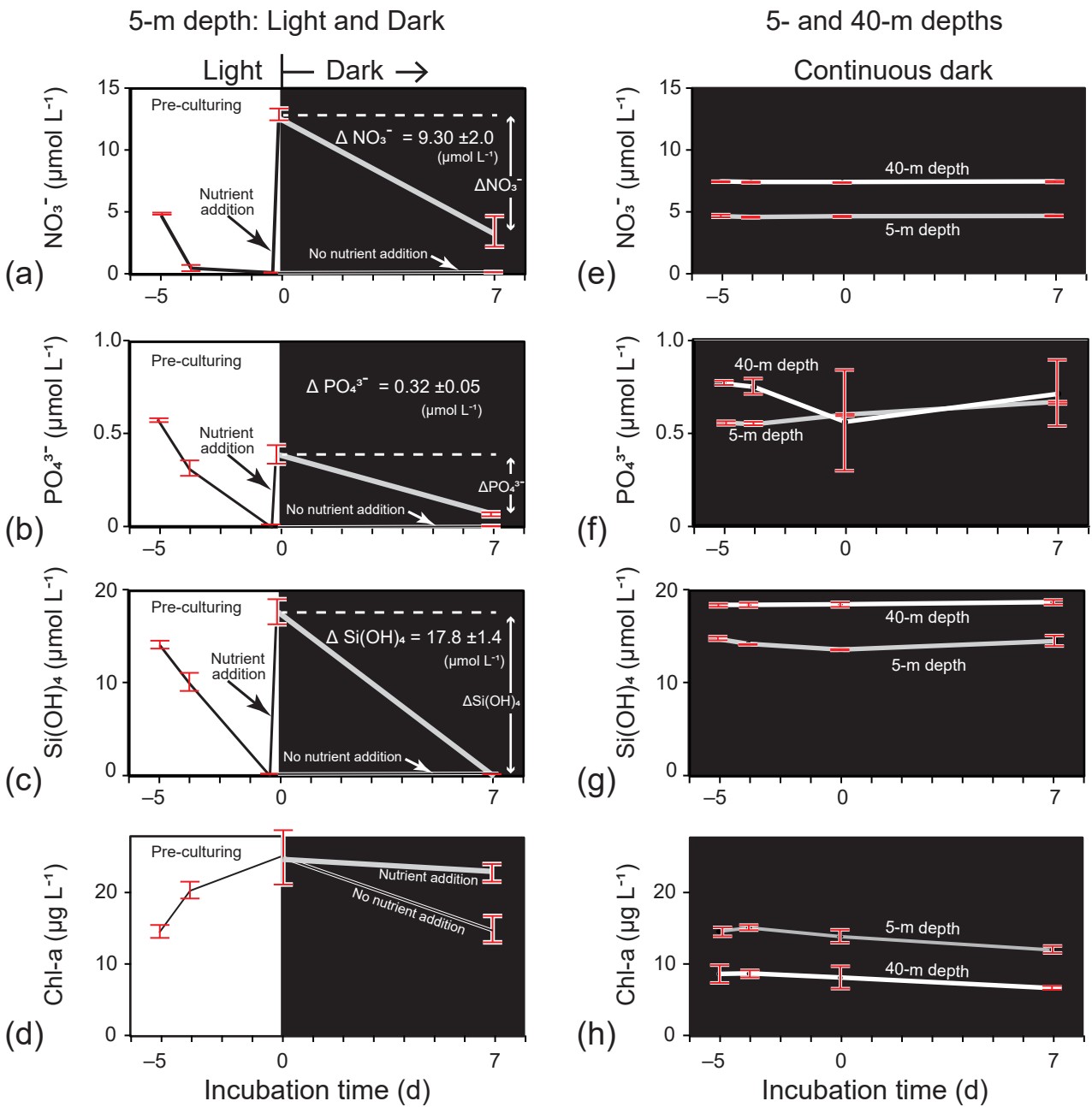

Fig. 8 Temporal change of chl-a and nutrients concentrations (mean ± 1stdev, n = 8 in light period, n = 4 in dark period) in the incubation experiment using natural seawater (5- and 40-m depths) collected from the station 30 of Funka Bay on 8 March 2022. Phytoplanktons in the seawater at 5-m depth were pre-cultured for 5 days under light condition to deplete nutrients before nutrients were added to the seawater on day 0 of dark period (nutrient addition) or not added (no nutrient addition); chl-a (a), $NO_3^-$ (b), $PO_4^{3-}$ (c), and $Si(OH)_4$ (d) for light and dark experiment. Phytoplanktons in seawater at 5- and 40-m depths were cultured under continuous dark condition without nutrient addition after the seawater collection; chl-a (e), $NO_3^-$ (f), $PO_4^{3-}$ (g), and $Si(OH)_4$ (h) for continuous dark experiment.

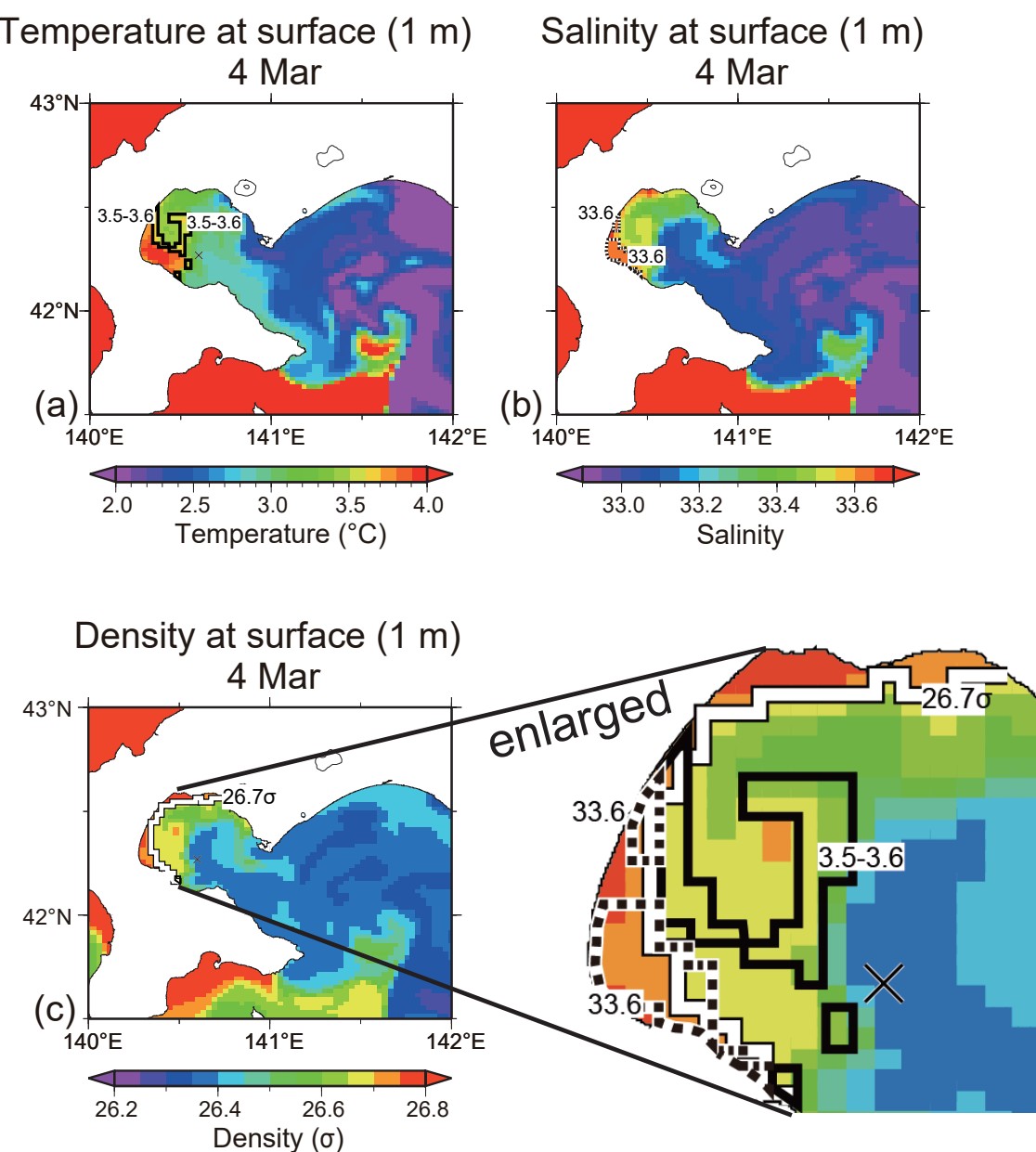

Fig. 9 Horizontal distributions of temperature (a), salinity (b), and density (c) at the surface (1 m) of the Funka-Bay on 4 March 2019. Boudary lines at temperature range 3.5 - 3.6 °C, salinity 33.6, and density 26.7σ were drawn in the respective figures. The all boudary lines were drawn in an enlarged figure of density. The location of observation station was maked with a cross. The ocean reanalysis product using an operational system for monitoring and forecasting the status of coastal and open-ocean waters around Japan (the JPN system) was provided by Meteorological Research Institute in Japan.