# Peer review of "Significant nutrient consumption in the dark subsurface layer during a diatom bloom: the case study on Funka Bay, Hokkaido, Japan"

_Biogeosciences, 2021_

## Referee Comment (RC2)

Umezawa et al. present a study investigating the contribution of diatoms to a nutrient decline in the dark subsurface water of Funka Bay (Japan) during and after the surface water bloom occurrence. Their hypothesis is that diatoms from the genera *Thalassiosira* that dominate the primary production in the surface ocean sink and accumulate below the euphotic zone where they continue to consume nutrient in the absence of what they call "photosynthetic growth". To confirm their hypothesis, authors have designed a laboratory experiment growing *Thalassiosira nordenskioeldii* in the dark and measuring the evolution of nutrient concentration in the medium. The main conclusion of this study reveals that *Thalassiosira nordenskioeldii* grown in culture conditions with replete nutrient concentration is able to consume nitrate in the absence of light. They attempted to demonstrate that this is the only possible explanation for the nutrient drawdown observed in the dark subsurface layer of Funka Bay between March 4th and April 14th.

Generally speaking, this manuscript is poorly organized, not clearly written and in a poor english wording. I would strongly advise the authors to read carefully their manuscript again and have it read by an English-speaking person. I believe that authors didn't use appropriate methods to evaluate the main objective of the study Because the incubation experiment has not been replicated the conclusions from this experiment that are supposed to support their main hypothesis regarding the in-situ nutrient drawdown is only speculative. Moreover, the rates estimated from this experiment cannot fully explain the observations. I have detailed my main concerns below and do not recommend publication of this manuscript in its actual state.

GENERAL COMMENTS
*In general, I feel that the introduction is poorly structured and weak. A lot of background about the hydrology but also about the dynamic of the phytoplankton bloom and productivity in Funka Bay is missing. I invite the authors to develop their bibliography and look closer in the literature for studies that have already been conducted in Funka Bay and develop their introduction a little further. For example (and this is not exhaustive), Nakada et al. (2013) for water mass dynamics, Shinada et al. (1999) and Radiarta and Saitoh (2008) for phytoplankton bloom and productivity dynamics in the bay. Most importantly, the objectives of this work are not clearly presented here. Why authors decided to focus on the processes affecting nutrient reduction in the dark subsurface layer during the bloom, what is the purpose of this study?
*Two main comments for the Material and Method section:
I do not think that 4 sampling days conducted in a random way can be considered as a time series, perhaps a survey…
When investigating biological processes, incubation experiments such as those conducted in this study MUST be replicated (triplicates are a must), otherwise any interpretation coming out from the experiment results is only speculative and conclusion cannot be ensured. This is very important!
*Section 3.2 of the Discussion is really long compared to other sections and contain a lot of redundancies. I suggest to re-organize this section chronologically instead of per nutrient, for example 3.2.1 pre-bloom, 3.2.2 bloom and 3.2.3 post-bloom situation. This might help reducing the length of this section (especially for chl*a* and nitrate) and remove most of the redundancies.

Moreover, this will allow authors to discuss the evolution of the nutrient ratios which will help confirming or infirming their hypothesis and which has not been explored in the discussion?
*The conclusion is poorly reflecting the main observations and hypothesis raised by the authors to explain the nutrient drawdown observed in the dark subsurface water of Funka Bay. Indeed, Authors cannot reject the influence of vertical mixing as they did L284-288, they haven't discussed the possibility of nutrient diffusion (active or passive), and the consumption rate of diatoms measured during the incubation experiment are not enough to fully explain the observed nutrient drawdown. The conclusion MUST be revised to truly reflect the main results of this study.
*Most of the figures of this manuscript either displayed the wrong information or are incomplete.

ABSTRACT
The abstract should use non-technical terms. For example, while "mixed-layer" is a common term used in oceanography and it is ok to use it in the abstract, Dark Zone Depth needs to be defined. Another option could be use "below the euphotic zone" instead of "dark zone depth". The main conclusion of the abstract needs to be revised since the nutrient consumption by diatom in the dark layer is likely not the only possible explanation to the observed nutrient drawdown in the subsurface waters of Funka Bay.

L14 and later in the manuscript: February 15th to April 14th of 2019. Please be careful on the calendar notation along the manuscript.
L16: "On both date"? During the whole time series from Feb 15th to Apr 14th? or during the bloom from March 4th to March 15th? By the way, winter water was above the euphotic zone (so above the dark-zone depths) on March 4th in fig.2.
L17: Same comment as above, please specify which dates. On fig.4, while $NO_3^-$ concentration decreases by approximately half, it is absolutely not the case for $Si(OH)_4$ and $PO_4^{3-}$ (e.g. at 50m concentrations dropped from 0.7 to 0.6 µmol $L^{-1}$ and from 15 to 13 µmol $L^{-1}$ for $PO_4^{3-}$ and $Si(OH)_4$ respectively)
L20-23: Authors present a list of several publications that have already studied dark nutrient consumption rates and ratios (see L250-259)

INTRODUCTION
L25-26: Surface euphotic zone sounds redundant.
L28: Use either "Si:N ratio" or "$Si(OH)_4$:$NO_3^-$ ratio"
L38: It is actually believed that the intrusion of the Oyashio water in the Bay triggers the diatom bloom by strengthening water column stratification.
L56-58: This is unfortunate since nutrient limitation can change from time to time. This manuscript doesn't have to focus on nutrient limitation in the surface water, however, it would be interesting to have a paragraph (even a short one) about what controls the dynamic of the bloom in the euphotic zone during this study. This is important since nutrient drawdown occurring in the surface layer will likely affect the subsurface biogeochemistry by creating gradient (and potentially diffusive fluxes) of nutrients, or by triggering adaptive response from the biology (such as vertical migration, resting spore formation etc.)

L58-61: VOI are not presented nor discussed in this manuscript. Please remove these sentences or add a figure, present and discuss the results of VOI measurements, and highlight their contribution in answering the objectives of the present study (why VOIs are measured?).

MATERIEL AND METHODS
L72-73: What does this sentence mean? What kind of observations? Are they relevant for this paper? Please remove this sentence or develop.
L77: $SiO_2$ is not a nutrient. This is the formula for particulate silica, silicic acid (the nutrient) is $Si(OH)_4$, please make sure to double-check the notations throughout the manuscript.
L78: Analytical precision: this is almost too good to be true! I am curious to see how this has been calculated?
L83: please use axenic instead of sterile. F/2 medium has approximately 106 $\mu mol.L^{-1}$ $Si(OH)_4$ 882 $\mu mol.L^{-1}$ $NO_3^-$, and 36 $\mu mol.L^{-1}$ $PO_4^{3-}$ (see Guillard and Ryther, 1962; Guillard, 1975 or Andersen et al (2005), Algual Culturing Techniques). It is more likely a modified (not particularly Si-rich) f/2 medium then, although I agree it is still plenty of nutrients for diatoms.
L86: This is really unfortunate since checking for contamination is crutial regarding the objective of the incubation experiment. Indeed, contamination by a population of other micro-algae or bacteria can affect the consumption (or recycling) of $NO_3^-$ and $PO_4^{3-}$ (although it is probably less the case for $Si(OH)_4$). This have to be mentioned and discussed in the discussion.
L93-97: What was the purpose of this second dark incubation? and why adding much more nutrient compare to the first one? Please explain.

RESULTS AND DISCUSSION
3.1. Hydrographic features
Are the water masses defined based only on salinity? Or did the authors use a T-S diagram to define the distribution (depth range) of the different water masses during the study? If so, it would have been useful to have a figure with a T-S diagram that illustrate the position of water mass end-members (such as in Shimizu et al. 2017) in supplement material at least.
L111: "biogeochemical"

3.2 Biogeochemical parameters
My guess is that authors meant "biogeochemical" instead of "biochemical" since biogeochemical refers to processes associated nutrient cycles and nutrient distribution while biochemical is more related to intra-cellular processes.
L115-118: Why authors chose this definition of the mixed layer, please explain AND cite appropriate reference. Same comment for euphotic zone and dark depth zone, appropriate referencing is missing.
L118: in general terms like "much less" should be avoided in papers, please be more specific and give a threshold or a range.
Also, mixed layer, euphotic zone and dark depth zone can be defined in the previous paragraph, since they can be considered as hydrographic features as well.
 L120-124: Authors cannot discuss in detail data that are not available either in another ALREADY published work, in a database, in supplement or in the main text of the manuscript. Authors cannot refer to an unpublished work when it concerns data availability, especially

when these data are used to make some of the figures. This is not acceptable! Moreover, I don't understand this sentence since the chl*a* profile in fig.3 looks pretty low and homogenous on Feb 15 and it seems that the chl*a* data ARE presented in the supplement…

L125-126: 27 to 30 µg $L^{-1}$ of chl*a* sounds very high to me. It looks like it's between 1.5 and 2 times higher than what is usually found in the bay (see L126-128). Is it a particularly exceptional year? Why is chl*a* that high?

L131: Sinking particles and suspended particles are different things. What makes authors conclude that particles can be sinking vs. in suspension? Please develop.

L140: A reference for the annual maximum is missing.

L150-167: Since there is a section specifically discussing the nutrient consumption in the dark subsurface layer, I would suggest to remove those hypotheses here to avoid redundancy with section 3.3. and to add this material to the discussion in section 3.3. This section 3.2 could thus be renamed "nutrient dynamic in the euphotic zone and dark subsurface layer" or something similar.

L168-175: I do not see why the first explanation of vertical mixing suggested for March 4$^{th}$ $NO_3^-$ decrease at depth cannot be still valid in March 15$^{th}$. Indeed, authors based their explanation mostly on the AOU profile. However, AOU profile in the 30-50m layer in March 14$^{th}$ looks pretty much the same as in March 4$^{th}$ (see fig.3).

L180-186: Note that the signal of regeneration is also clearly seen from the strongly positive values of AOU at depth in April 14$^{th}$.

L185: It is important to be more specific here. For example, the sentence can be "The very high $PO_4^{3-}$, $Si(OH)_4$ and $NH_4^+$ concentrations measured at depth (1.9, 26.1, and 3.2 µmol $L^{-1}$, respectively) clearly indicate that nutrient regeneration already occurred at the bottom of the water column in April 14$^{th}$".

3.3 Nutrient consumption in the dark subsurface layer

I suggest to add a subsection between subsection 3.3.1 and 3.3.2 to discuss a third hypothesis for nutrient drawdown in the dark subsurface layer, the diffusive flux of nutrient, that have been neglected so far in the manuscript but I think could be of significant importance.

L211: It is currently impossible to see the density difference between 5 and 30m in fig. 4 since both density profiles are incomplete. Moreover, it is more about the shape or curvature of the density profile rather than the magnitude of the density difference. Indeed, a smooth and progressive gradient of density will not act as a strong barrier against exchange compared to a sharp change in density (even though this latter is smaller).

L212-214: Please show these data in a figure

L215: Although this might be true, I do not agree with authors since we do not have access to the observations that support this conclusion (there is no figure illustrating the wind speed and the density profiles on fig.4 are incomplete).

L218-228: This could be another subsection focused on discussing horizontal advection

L222-224: A mixing ratio of 8:2 between Funka-Bay and Oyashio waters is a curious way to present the mixing between these two water masses. Won't it be clearer to say that to generate the observed decrease in salinity, a mixing with 25% of Oyashio water (or 1:4 Oyashio water and Funka Bay water) is necessary. By the way, by using salinities of 33.0, 33.47 and

33.58 for Osahio water, Funka Bay water, and the mix between the two of them, respectively; my estimate is 20% (1:5) of Oyashio water needed to produce the observed decrease in salinity.

L227-228: This is not because the vertical mixing cannot explain all the nutrient drawdown that this hypothesis should be excluded, it is more likely not the main driver of the nutrient drawdown. It doesn't mean that it is not happening, it is more likely a combination of different processes. I suggest to rephrase this conclusion.

L231-232: Same comment as above. The nutrient reduction in the subsurface layer is more likely driven by multiple processes, consumption of nutrient by diatom in the dark might be one of them (not the only one).

L237 and after: Please explain the µg chl$a^{-1}$ d$^{-1}$ unit used for consumption rates. Chl$a$ concentrations during the incubation experiment are not presented in fig.5 nor elsewhere in the manuscript. Also, in the main text $NO_3^-$, $PO_4^{3-}$ as well as $Si(OH)_4$ consumption are discussed but only $NO_3^-$ is show in fig.5. The evolution of ALL nutrient concentration MUST be presented somewhere if authors are discussing them in the manuscript.

L238-342: Although calculation of consumption rates cannot be verified since nutrient concentration during the incubation are not presented, it seems that the consumption rate of diatoms estimated from the incubation experiment are very low and cannot fully explain the nutrient drawdown observed in the 30-50m layer between march 4$^{th}$ and March 14$^{th}$.

L243-245: Are these consumption rates calculated between day 0 and day 2 such as in the first incubation experiment? I don't think rates from the second incubation experiment can be compared to rates in the first experiment since too many parameters were different between the two experiments (e.g. concentrations were much higher in the second experiment, since diatom uptake usually follows a Michaelis-Menten function, this can change a lot the dynamic of the diatom uptake!). Does this mean that, because the nutrient concentrations in the field were much lower compare to those set up during both experiments, we can assume that the nutrient consumption rates were probably much lower in the water column and thus would probably contribute to a very small amount of the observed nutrient drawdown?

L247-249: I do not understand this sentence. Why would the consumption rate be largely dependent on the chl$a$ content in the diatom cells if, as suggested by authors L165, this nutrient consumption occurs "without photosynthetic growth"? Please explain.

L248-249: I totally agree with this statement!

L251: I do not understand this sentence, please rephrase.

L268-273: All diatoms do not migrate. Those vertical migrations have been observed for mats of Rhizosolenia. Do authors have evidence for such a behavior in Thalassiosira? This needs to be discussed here.

L275: What are those assumptions? They need to be presented and discussed here.

CONCLUSION

L279-280: Once again, I'm not convinced that 4 sampling days can be considered as a time series.

L291-292: I do not agree with this. The nutrient consumption rates estimated from the incubation experiment are too low to explained the nutrient drawdown observed in the 30-50m layer in Funka Bay.

FIGURES

Figure 1: There is only one station/sampling site in this study. Describe what O (Oyashio water) and T (Tsugaru water) are in the figure caption or legend.

Figure 2: Define the euphotic zone and MLD in the captions instead of in the legend. Be consistent in the formatting of the legend (e.g. WO: Transitional water instead of Transitional Water: WO).

Figure 3: MLD is not at the same depth on Apr 14$^{th}$ in fig.2 (40m) and fig.3 (15m), same on March 4$^{th}$ fig.2 (~5m) vs. fig.3 (~10m). Please remove the red rectangle that is supposed to show the concentration change in 4-15 March. It actually doesn't help to read the figure. Most importantly this figure involves a lot of interpolations made from a total of four stations. The time coverage is probably not enough and could generate bias in the figure that could lead to misinterpretation. For example, on panel a. it seems that the bloom started soon after the sampling in Feb15, and peaked on March 4$^{th}$, but we know that diatoms can consume nutrients pretty quickly in the mixed layer and the bloom could have started anytime between Feb 15$^{th}$ and March 4$^{th}$, same for the termination of the bloom. I would suggest to present these data as profiles which will be more accurate instead of interpolated time-sections.

Figure 4: The scale on the x axis in the temperature, salinity and density panels are not appropriate since the profile corresponding to March 15$^{th}$ in all panels is cut at the surface, same for the March 4$^{th}$ density profile. It is hard to discuss incomplete profiles. Right now, we cannot conclude that stratification is stronger on March 15$^{th}$ as stated L211.

The nutrient concentration difference in panel d, e and f are pretty obvious indeed, no need to add extra arrows. Moreover, the left-pointing arrow on panel c doesn't mean anything since density increases in the mixed layer from March 4$^{th}$ to March 15$^{th}$. I would be interesting to add the standard deviations for the nutrient concentration measurements.

Figure 5: Overall, I like this figure. It is clear, although the scale of the x axis on panel b does not make sense (two squares correspond to 4 days (0 to 4) then to two days (4 to 6), then back to 4 days (6 to 10). I also still don't understand how the authors have defined their set-up conditions (e.g. why adding 30µmol. L$^{-1}$ NO$_3^-$ in the first experiment and around 750µmol L$^{-1}$ in the second one? Why this threshold of 1429µg L$^{-1}$ chl$a$ in the first experiment but 72.5µg L$^{-1}$ in the second one? Although I don't remember is the culture experiment has been replicated (Working with biology, I think it is crucial to run that kind of experiments in triplicate, or at least duplicate them) but this figure needs error bars.

---

## Author Comment (AC1)

Dear anonymous referee 1

We send responses to referee1's comments for the submitted manuscript "Substantial nutrient consumption in the dark subsurface layer during a diatom bloom: the case study on Funka Bay, Hokkaido, Japan" by Umezawa et al. The title was revised according to the comment. The referee's comments were very helpful, and we have revised the manuscript taking all comments. The modification was highlighted in this response file. We attached an additional figure (spatial distributions of temperature, salinity, and density at surface 1 m) to explain a possibility of subduction.

Best regards

Corresponding author: Atsushi Ooki

**Reviewer's comment 1**

1)      Effects of the physical processes.

The authors denied the vertical mixing and horizontal mixing are the main reasons to decline the nutrient concentration at 30–50 m depth of this area.  However, the water density on 15th March is lighter than that of 4th March.  The authors did not show the horizontal distributions of density and nutrient concentration.  So the subduction processes cannot be denied.  When the authors have much data, the nutrient(nitrate)-density plot would help discussion on it.  I am concerned about the possibility that the low-nutrient water formed in the euphotic layer in the other area subducted at the observation station. For example, such a phenomenon occurs in an anticyclonic eddy.

**Respond to the comment 1**

We added a discussion about the subduction with new figures  as follows:

**Line 300 - 312**

4.2.4 Subduction of surface water into the subsurface layer

Fourth, we discuss a possibility if subduction of surface water caused the decrease in nutrient concentrations at the subsurface layer (30 – 50 m) of the observation station 30. At the medium depth (40 m) of the subsurface layer, temperature, salinity, and density were 3.5 – 3.6 °C, 33.64, and 26.7σ, respectively, on these dates. Suppose surface water in certain area of the bay subducted and it reached 40 m-depth at the observation station on 15 March, the subducted surface water should have the same temperature, salinity, and density. The average current speed at 40 m-depth between these dates was 3.3 cm s$^{-1}$ (unpublish data),

which was obtained from acoustic doppler current profiler (ADCP) set on the sea floor. The middle layer water at the station could have reached from anywhere of the bay within 11 days. We prepared spatial distributions of temperature, salinity, and density at the sea surface (1 m) on 4 March (Fig. 7a-c). From these spatial distributions, there was not any area that satisfied required temperature, salinity, and density to form subduction water into the subsurface layer (see an enlarged figure 7c). The low-density water (< 26.65σ) was spreading to the back of the bay since 4 March. We considered that the subsurface layer water at the station was not associated with subduction. Thus, we excluded a possibility of subduction as a reason for the nutrient decline.

[Figure]

Fig. 7 Horizontal distributions of temperature (a), salinity (b), and density (c) at the surface (1 m) on 4 March 2019. Lines at temperature range 3.5 - 3.6 °C, salinity 33.6, and density 26.7σ were drawn in the figure. The all lines were drawn in an enlarged figure of density. The location of observation station was maked with a cross.

**Reviewer's comment 2**

The other possible process is the nutrient diffusion process. I don't know the diffusion of this area, > several tens μmol N m−2 d−1 usually occurs in the ocean. The nutrient diffusion occurs with physical disturbance, but diapycnal nutrient flux must be considered. The observations were snapshots, and the authors may not observe the diffusion processes, but the authors must show the nutrient flux at 50 m and 30 m depths are balanced on 4th and 15th March based on the slope of the nitracline and pycnocline, and this process is not the major nutrient decline process.

**Respond to the comment 2**

According to the referee's comment, we calculated the diffusive transport of nutrients. We added a discussion about the diffusive transport of nitrate as follows:

**Line 286 - 299**

**4.2.3 Diffusive transport between the surface and the subsurface layers**

Third, we discuss an effect of diffusive transport of $NO_3^-$ on concentration change at the subsurface layer (30 – 50 m) on 15 March. There are not any diffusive coefficients ($K\rho$) in Funka Bay. We referred a range of $K\rho$ (= $10^{-6} – 10^{-5}$ m$^2$ s$^{-1}$) measured just below the mixed layer (~ 30 m) at the western subarctic Pacific in summer (Dobashi et al. 2021). Concentration gradients of $NO_3^-$ were –0.000221 µmol m$^{-4}$ (= $\Delta NO_3^-{}_{20m–30m}$ / 10 m), –0.000141 µmol m$^{-4}$ (= $\Delta NO_3^-{}_{30m–40m}$ / 10 m), –0.000115 µmol m$^{-4}$ (= $\Delta NO_3^-{}_{40m–50m}$ / 10 m), and –0.0000135 µmol m$^{-4}$ (= $\Delta NO_3^-{}_{50m–60m}$ / 10 m). The range of diffusive transport of $NO_3^-$ were calculated to be 0.00022 – 0.0022 µmol m s$^{-2}$ between 20 m and 30 m, which could result in concentration change of 0.021 ~ 0.21 µmol L$^{-1}$ at 30 m for 11 days. Concentration changes between 30 m and 40 m and between 40 m and 50 m were calculated to be 0.013 ~ 0.13 µmol L$^{-1}$ and 0.011 ~ 0.11 µmol L$^{-1}$, respectively. The total concentration change at 30 m, which includes transports from 20 m layer and 40 m layer, ranges from –0.20 µmol L$^{-1}$ (= –0.21 + 0.013) to +0.11 µmol L$^{-1}$ (= –0.021 + 0.13). Ranges of total concentration change at 40 m and 50 m were –0.12 ~ +0.096 µmol L$^{-1}$ and –0.11 ~ –0.024 µmol L$^{-1}$, respectively. The observed decreases were of 1.6 µmol L$^{-1}$ at 30 m, 2.0 µmol L$^{-1}$ at 40 m, and 2.4 µmol L$^{-1}$ at 50 m between the two dates. Thus, we concluded that diffusive transport of $NO_3^-$ had a minor effect on the concentration decreases at the subsurface layer.

**Reviewer's comment 3**

2)      Impact of the biogeochemistry and primary production

The authors concluded "This consumption could result in reduced new production in the subsurface layer after the bloom, when this layer would once again become part of the euphotic zone, if the diatoms sank to deeper layers."  However, I cannot agree with this without the evidence that the diatoms are not increased in this layer. Diatoms have some unique modes of nutrient uptake (Martin-Jezequel et al. 2000).  Is the observed nutrient uptake of diatoms in the dark condition not linked to the growth?  When the authors have the time-series chlorophyll a concentration data in the laboratory experiments, please show the data and discuss that they did not fix carbon. In the case of cyanobacteria, they grow up in the twilight zone (Sohrin et al. 2011). In addition, the dark condition in the laboratory may be different from the dark condition of the field.  Even though the PAR is less than 0.1% at the surface, it was not completely dark in the ocean.  Many exciting discussions may be possible: nitrate uptake (new production) may occur in the twilight zone but not contribute to the primary production/ new production is underestimated when the nitrate uptake is not measured in more dark layers.

**Respond to the comment 3**

We proposed alternative hypotheses (1 and 2) to deduce the influence of nutrient uptake by diatoms in dark subsurface layer in bloom.

1) If the diatom population that had consumed half of the nutrients in the dark subsurface layer sank to the deeper layer during the bloom, then the primary production in the subsurface layer after the bloom, at which time it would be part of the euphotic zone, would be reduced by half compared to the production in the case where there was no nutrient consumption during the dark period.

2) If the diatoms that had consumed nutrients in the dark subsurface layer remained in that layer after the bloom, they would rapidly grow under the returning light conditions when the euphotic zone deepened after the bloom.

As for the first hypothesis, we are sure that nutrient loss from euphotic layer leads decrease in primary production. In the case of Funka Bay, we have noted that the consumption of nutrients in the dark subsurface layer would have an impact outside the bay, because the subsurface water is exchanged with Oyashio water. We need to test if diatoms that have once taken nutrients in darkness can grow rapidly in light afterward and if the light intensity (1% PAR, 0.1% PAR, and complete darkness) has an impact on nutrient uptake. We would like to use Martin-Jezequel et al (2000) and Sohrin et al (2011) as references to plan the future research.

To clarify our viewpoints, we have revised as follows:

**Line 315 - 326**

We propose alternative hypotheses (1 and 2) to deduce the influence of nutrient uptake by diatoms in dark subsurface layer in bloom.

1) If the diatom population that had consumed half of the nutrients in the dark subsurface layer sank to the deeper layer during the bloom, then the primary production in the subsurface layer after the bloom, at which time it would be part of the euphotic zone, would be reduced by half compared to the production in the case where there was no nutrient consumption during the dark period.

2) If the diatoms that had consumed nutrients in the dark subsurface layer remained in that layer after the bloom, they have a potential to rapidly grow under the returning light conditions when the euphotic zone deepened after the bloom.

In the case of Funka Bay, we note that the consumption of nutrients in the dark subsurface layer would have an impact outside the bay, because the subsurface water is exchanged with Oyashio water.

**Reviewer's comment 4**

3)      The structure of the manuscript

It was just an opinion, but I am familiar with the manuscript which divides results and discussion. I believe the authors can divide them.  However, if the authors considered the present style is better, this is not mandatory.  This is the option, too, but the title of the manuscript should be a more appealing one.  The present title only attracts local interests.  For example, when the authors consider the observed phenomenon possibly occurs everywhere under diatom blooms, the title can be revised as "Significant nutrient consumption in the dark subsurface layer during a diatom bloom: the case study on Funka Bay, Hokkaido, Japan".

**Respond to the comment 4**

According to the comment, we divided results and discussions and revised the title.

**Reviewer's comment 5**

L14: Times of observations are necessary. Technically, the authors' observation is not time-series, because the observation was conducted randomly.

**Respond to the comment 5**

We revised the manuscript accordingly as follows:

**Line 349 - 350**

We conducted repetitive observations in Funka Bay, Hokkaido, Japan, 15 February, 4 and 15 March, and 14 April 2019.

**Reviewer's comment 6**

L21 "We believe that this is the first study to present observational evidence for the consumption of the main nutrients by diatoms in the dark subsurface layer during the spring bloom." In my opinion, this sentence is not important. Instead of this sentence, the authors should add why they considered the nutrient decline does not occur by physical processes.

**Respond to the comment 6**

We added a sentence about exclusions of physical processes as an explanation of nutrient reduction in the subsurface layer. We think the sentence "this is the first study to present ---" is important. **So, we described it in the abstract.**

**Line 19 - 20**

==We excluded possibilities of three physical process, water mixing, diffusive transport, and subduction, as reasons for the decrease in nutrients in the subsurface layer.==

**Reviewer's comment 7**

L26 "Si: NO3– ratio". Yes, this is not wrong and described in Harrison et al 2004, but $Si(OH)_4:NO_3$ or Si: N ratio is more appropriate. This is just opinion.

**Respond to the comment 7**

We corrected accordingly.

**Reviewer's comment 8**

L31–32: References are required.

**Respond to the comment 8**

We added a reference (Rosa et al., 2007).

**Reviewer's comment 9**

L34–35: "From time-series observations in the bay, it is possible to examine the temporal changes of biochemical parameters within the same identified water mass while the water is in the bay." I cannot understand this sentence clearly. What means "while the water is in the bay"?

**Respond to the comment 9**

We revised the sentence as follows:

**Line 35 - 37**

From repetitive observations in the bay, it is possible to collect the same identified water mass in different times while the water is in the bay and examine the temporal changes of biochemical parameters within the same identified water mass.

**Reviewer's comment 10**

L37–39: "A massive spring bloom dominated by diatom species occurs in March every year before the Oyashio water flows into the surface of the bay, and it lasts until late March or early April, when Oyashio water occupies the surface of the bay (Odate 1987; Maita and Odate 1988)." I cannot understand this sentence. Please clarify. Can the author divide it into two sentences?

**Respond to the comment 10**

We revised the sentence as follows:

**Line 39 - 41**

A massive spring bloom dominated by diatom species occurs in March every year (Odate 1987; Maita and Odate 1988). Oyashio water flows into the bay from the surface after the begging of the bloom. The bloom lasts until late March or early April. Oyashio water occupies the surface of the bay in April (Kudo and Matsunaga, 1999).

**Reviewer's comment 11**

L58–61: I could not find any meaning in these two sentences. I cannot see any discussion of VOIs in this manuscript. In addition, the reference is under consideration. So I considered these two sentences should be removed.

**Respond to the comment 11**

According to the comment, we removed the sentence.

**Reviewer's comment 12**

Method: L72–73: "Observations in Funka Bay have been reported elsewhere (Shimizu et al. 2017)." What did the authors want to describe? Some information on the observations conducted in 2019 was described in 2017? Describe the details or remove the sentence.

**Respond to the comment 12**

According to the comment, we removed the sentence.

**Reviewer's comment 13**

L78: How do the authors calculate the analytical precision? This is very low. Did the authors measure the nutrient concentration of not-frozen samples? If the results are frozen samples, the precisions are too good, in particular, silicate.

**Respond to the comment 13**

We added information about the determination of precision as follows.

**Line 77 - 78**

Analytical precision was 0.12% for $NO_3^-$, 0.21% for $NO_2^-$, 0.19% for $PO_4^{3-}$, 0.11% for $SiO_2$, and 0.34% for $NH_4^+$ as determined by repeated measurement (n = 7) of reference seawater for nutrient standards (KANSO, standard Lot BZ, Osaka, Japan).

**Reviewer's comment 14**

L82: Where Thalassiosira nordenskioeldiicome from? Algae collection?

**Respond to the comment 14**

We added a sentence accordingly:

**Line 83 - 84**

A diatom *Thalassiosira nordenskioeldii*, which predominates in the early phase of the spring bloom in Funka Bay (Ban et al 2000), was isolated from natural seawater collected in the western subarctic Pacific Ocean (Oyashio area) in May 2019.

**Reviewer's comment 15**

L95: "We set the initial concentrations of nutrients at 23 times those of the first dark incubation." Why did the authors set so high initial nutrient concentration? The environments are very different from the field observations.

**Respond to the comment 15**

The added information about the amount of added amount of nutrients as follows:

Line 231-233

The added amount of $NO_3^-$ per chl-a (0.022 = 31.1 µmol $L^{-1}$ / 1426 µg $L^{-1}$) into the nutrient depleted culture of the first experiment was 9% of the amount of $NO_3^-$ per chl-a (0.24 = 3.4 µmol $L^{-1}$ / 14 µg $L^{-1}$) in seawater at 30 m on 4 March.

Line 241-244

If the diatoms consumed nutrients much more quickly than we collected the culture sample on day 2, the daily consumption rates are underestimated.

We added enough amount of nutrients in the second incubation experiment. The added amount of $NO_3^-$ per chl-a (10.3 = 743.5 µmol $L^{-1}$ / 72.5 µg $L^{-1}$) was 43 times of the seawater.

**Reviewer's comment 16**

* Did not the authors conduct the microscopic observations? Is Thalassiosira nordenskioeldii the dominant species of the observations? This is very important. Because the other species is dominant, the authors' incubation experiments are meaningless.

**Respond to the comment 16**

We added information about microscopic analysis as follows:

Line 228-231

From microscopic image analysis, *Thalassiosira nordenskioeldii* occupied 14.2% of number of phytoplankton cells (n = 1209) collected by plankton net (mesh = 100 µm) on 15 March 2019. Other dominant species were *Chaetoceros* spp. and other *Thalassiosira* sp. We confirmed that *Thalassiosira nordenskioeldii* was one of the dominant species in the spring bloom 2019.

**Reviewer's comment 17**

L102–106: Please define the water masses at the materials and methods. When the authors defined in the materials and methods section, the results will be simpler.

**Respond to the comment 17**

We revised the manuscript accordingly.

**Reviewer's comment 18**

L106: "The revised classification result" This is unclear. Did the authors revise in this manuscript or revise in Ooki et al. 2019?

**Respond to the comment 18**

We used the definition of water mass classification proposed in Ooki e al. (2019). Ooki et al. (2019) made a minor change in the classification by Ohtani and Kido (1980).F

We removed the phrase "The revised classification result"

**Reviewer's comment 19**

L115–119: This is not a result.  Please define in the materials and methods section.

**Respond to the comment 19**

We revised the manuscript accordingly.

**Reviewer's comment 20**

L123: "the original data in supplementary information of Ooki et al., (submitted)." Is it right?  I can see the supplementary information of this manuscript.

**Respond to the comment 20**

We have used the same chl-a data in the two papers (this paper and unpublished paper). We removed the description "original data in supplementary information of Ooki et al., submitted".

**Reviewer's comment 21**

L124: "The data for chl-a are taken from a related article."  What does it mean?  I think it is acceptable to share the data with other manuscripts.

**Respond to the comment 21**

We removed the description "The data for chl-a are taken from a related article".

**Reviewer's comment 22**

L150–168: These paragraphs were "discussion". These discussions can be put after the results section because the results after this paragraph are not contained the results of the discussion. For me, this style is hard to follow.

**Respond to the comment 22**

According to the comment, we moved these paragraphs from "results" to "discussion".

**Reviewer's comment 23**

3.3.2 This section (results of incubation experiments) should be shown before 3.3.1.

**Respond to the comment 23**

We revised accordingly.

**Reviewer's comment 24**

L270: Villareal et al. reported Rhizosolenia, and not Thalassiosira. Do the authors have any evidence on the vertical migration of Thalassiosira? If not, this discussion is speculative. In addition, the authors' names are wrong: Wirtz and Lan Smith (2020) are correct.

**Respond to the comment 24**

We added some explanations about the vertical migration of Thalassiosira (Richardson and Cullen, 1995). To clarify our viewpoint, we revised as follows (please see the sentence marked with yellow):

Line 328-337
In relation to the second hypothesis, an interesting survival strategy for diatom, *Rhizosolenia*, which forms large aggregations (mats), has been proposed (Villareal et al. 1996; Richardson

et al. 1998; Villareal et al., 1999; Villareal et al., 2014). The survival strategy of *Rhizosolenia* is that they consume $NO_3^-$ in the dark subsurface layer, and then migrate to the surface euphotic layer where they have a growth advantage in oligotrophic subtropical open ocean areas.  For the coastal marine diatom, *Thalassiosira weissflogii,* was studied to examine changes in buoyancy and ratio of carbohydrate to protein which determine the cell density (Richardson and Cullen, 1995). They revealed that accumulation of carbohydrate as a result of nitrate depletion leads rises in cellular density and sinking speed and that accumulation of protein as a result of nitrate addition after the nitrate depletion leads a positive buoyancy. Several modelling studies reported contributions of primary production by vertically migrating phytoplankton to net primary production. For example, Witz and Lan Smith, (2020) estimated that vertically migrating phytoplankton contributes 7% of net primary production at the subarctic gyre of the western Pacific.

---

## Author Comment (AC2)

Dear anonymous referee1

We will send responses to referee's comments of "Substantial nutrient consumption in the dark subsurface layer during a diatom bloom: the case study on Funka Bay, Hokkaido, Japan" (revised title) by Umezawa et al. The referee's comments were very helpful, and we have revised the manuscript taking all comments. Corrections in the revised manuscript (and the responses to referee's comments) are highlighted.

Best regards,

Corresponding author: Atsushi Ooki

**Contents of response letter**
Response to referee 1 (comment 1 – 24)
Revised / added figures (Fig.2, 5, 6, 7, 8)

**Revised chapter structure**
Abstract
1. Introduction
2. Material and methods
   2.1 Shipboard observations
   2.2 Analytical procedures
   2.3 Incubation experiments to test for nutrient consumption by diatoms in darkness
   2.4 Water mass types and mixed-layer and euphotic-zone depths
   2.5 Spatial distributions of temperature, salinity, and density at the sea surface
3. Results
   3.1 Hydrographic features
   3.2 Biogeochemical parameters
      3.2.1 Chl-a
      3.2.2 Nitrate
      3.2.3 Phosphate and silicate
      3.2.4 Ammonium
   3.3 Limiting factor of primary production during the bloom
4. Discussions
   4.1 Nitrate consumption in the dark layer between 15 February and 4 March
   4.2 Nutrient consumption in the dark subsurface layer between 4 and 15 March
      4.2.1 Nutrient consumption by diatoms in darkness
      4.2.2 Water mixing as a possible explanation for nutrient reduction
      4.2.3 Diffusive transport between the surface and the subsurface layers
      4.2.4 Subduction of surface water into the subsurface layer
   4.3 The influence of nutrient consumption by diatoms in the dark subsurface layer
5. Conclusion

**Response to referee 1**

**Reviewer's comment 1**

1)      Effects of the physical processes.

The authors denied the vertical mixing and horizontal mixing are the main reasons to decline the nutrient concentration at 30–50 m depth of this area.  However, the water density on 15th March is lighter than that of 4th March.  The authors did not show the horizontal distributions of density and nutrient concentration.  So the subduction processes cannot be denied.  When the authors have much data, the nutrient(nitrate)-density plot would help discussion on it.  I am concerned about the possibility that the low-nutrient water formed in the euphotic layer in the other area subducted at the observation station. For example, such a phenomenon occurs in an anticyclonic eddy.

**Respond to the comment 1**

We added a discussion about the subduction with new figures  as follows:

Line 144-152

2.5 Spatial distributions of temperature, salinity, and density at the sea surface
Spatial distributions of temperature, salinity, and density at the sea surface (1 m) were obtained from an ocean reanalysis product provided by Meteorological Research Institute in Japan. This is produced with an operational system for monitoring and forecasting the status of coastal and open-ocean waters around Japan (the JPN system; Hirose et al., 2020). The JPN system includes a double-nested ocean model, the core of which is a Japanese coastal model with a horizontal resolution of 2 km. Three sub-models are interconnected using a nesting technique: a global model (horizontal resolution ~100km), a North Pacific model (horizontal resolution ~10km), and Japanese coastal model (horizontal resolution ~2km). A four-dimensional variational method is applied to the North Pacific model as the assimilation scheme. The process of tides and river runoff are taken into consideration in this JPN system. See the technical report for more detail (Hirose et al. 2020).

**Line 374 - 387**

**4.2.4 Subduction of surface water into the subsurface layer**

Fourth, we discuss a possibility if subduction of surface water caused the decrease in nutrient concentrations at the subsurface layer (30 – 50 m) of the observation station 30. At the medium depth (40 m) of the subsurface layer, temperature, salinity, and density were 3.5 – 3.6 °C, 33.64, and 26.7$\sigma$, respectively, on 4 and 15 March (Fig. 2). Suppose surface water in certain area of the bay subducted and it reached 40-m depth at the observation station on 15 March, the subducted water should have the same temperature, salinity, and density as it had been at the surface. The average current speed at 40-m depth between these dates was 3.3 cm s$^{-1}$ (unpublish data), which was obtained from acoustic doppler current profiler (ADCP) set on the sea floor at the station. The middle layer water at the station could have reached from anywhere of the bay within 11 days. We obtained the spatial distributions of temperature, salinity, and density at the sea surface (1 m) on 4 March using the ocean reanalysis product provided by Meteorological Research Institute in Japan (Fig. 8a-c).
From these spatial distributions, there was not any area that satisfied required temperature (3.5 – 3.6 °C), salinity (33.64), and density (26.7$\sigma$) to form subduction water at 40-m depth of the observation

station, see an enlarged map of Fig. 8c. We considered that the subsurface layer water at the station was not associated with subduction. Thus, we excluded a possibility of subduction as a reason for the nutrient decline.

**Reviewer's comment 2**

The other possible process is the nutrient diffusion process. I don't know the diffusion of this area, > several tens µmol N m$^{-2}$ d$^{-1}$ usually occurs in the ocean.  The nutrient diffusion occurs with physical disturbance, but diapycnal nutrient flux must be considered.  The observations were snapshots, and the authors may not observe the diffusion processes, but the authors must show the nutrient flux at 50 m and 30 m depths are balanced on 4th and 15th March based on the slope of the nitracline and pycnocline, and this process is not the major nutrient decline process.

**Respond to the comment 2**

According to the referee's comment, we calculated the diffusive transport of nutrients. We added a discussion about the diffusive transport of nitrate as follows:

**Line 360 - 373**

**4.2.3 Diffusive transport between the surface and the subsurface layers**

Third, we discuss an effect of diffusive transport of NO$_3^-$ on concentration decrease at the subsurface layer (30 – 50 m) on 15 March. There is not any previous study to have measured diffusive coefficients (K$\rho$) in Funka Bay. We referred a range of K$\rho$ (= 10$^{-6}$ – 10$^{-5}$ m$^2$ s$^{-1}$) measured just below the mixed layer (~ 30 m) at the western subarctic Pacific in summer (Dobashi et al. 2021). Concentration gradients of NO$_3^-$ were –0.000221 µmol m$^{-4}$ (= $\Delta$NO$_3^-$$_{20m-30m}$ / 10 m), –0.000141 µmol m$^{-4}$ (= $\Delta$NO$_3^-$$_{30m-40m}$ / 10 m), –0.000115 µmol m$^{-4}$ (= $\Delta$NO$_3^-$$_{40m-50m}$ / 10 m), and –0.0000135 µmol m$^{-4}$ (= $\Delta$NO$_3^-$$_{50m-60m}$ / 10 m).  The range of diffusive transport of NO$_3^-$ were calculated to be 0.00022 – 0.0022 µmol m s$^{-2}$ between 20 m and 30 m, which could result in concentration change of 0.021 ~ 0.21 µmol L$^{-1}$ at 30 m for 11 days. Concentration changes between 30 m and 40 m and between 40 m and 50 m were calculated to be 0.013 ~ 0.13 µmol L$^{-1}$ and 0.011 ~ 0.11 µmol L$^{-1}$, respectively. The sum of concentration changes at 30 m, which include transports from 20 m layer and 40 m layer, ranges from –0.20 µmol L$^{-1}$ (= –0.21 + 0.013) to +0.11 µmol L$^{-1}$ (= –0.021 + 0.13). Ranges of the sum of concentration changes at 40 m and 50 m were –0.12 ~ +0.096 µmol L$^{-1}$ and –0.11 ~ –0.024 µmol L$^{-1}$, respectively. The observed decreases were of 1.6 µmol L$^{-1}$ at 30 m, 2.0 µmol L$^{-1}$ at 40 m, and 2.4 µmol L$^{-1}$ at 50 m between these dates. Thus, we concluded that diffusive transport of NO$_3^-$ had a minor effect on the concentration decreases at the subsurface layer.

**Reviewer's comment 3**

2)      Impact of the biogeochemistry and primary production

The authors concluded "This consumption could result in reduced new production in the subsurface layer after the bloom, when this layer would once again become part of the euphotic zone, if the diatoms sank to deeper layers." However, I cannot agree with this without the evidence that the diatoms are not increased in this layer. Diatoms have some unique modes of nutrient uptake (Martin-Jezequel et al. 2000). Is the observed nutrient uptake of diatoms in the dark condition not linked to the growth? When the authors have the time-series chlorophyll a concentration data in the laboratory experiments, please show the data and discuss that they did not fix carbon. In the case of cyanobacteria, they grow up in the twilight zone (Sohrin et al. 2011). In addition, the dark condition in the laboratory may be different from the dark condition of the field. Even though the PAR is less than 0.1% at the surface, it was not completely dark in the ocean. Many exciting discussions may be possible: nitrate uptake (new production) may occur in the twilight zone but not contribute to the primary production/ new production is underestimated when the nitrate uptake is not measured in more dark layers.

**Respond to the comment 3**

We proposed alternative hypotheses (1 and 2) to deduce the influence of nutrient uptake by diatoms in dark subsurface layer in bloom.

1) If the diatom population that had consumed half of the nutrients in the dark subsurface layer sank to the deeper layer during the bloom, then the primary production in the subsurface layer after the bloom, at which time it would be part of the euphotic zone, would be reduced by half at maximum compared to the production in the case where there was no nutrient consumption during the dark period.

2) If the diatoms that had consumed nutrients in the dark subsurface layer remained in that layer after the bloom or migrated to the upper layer, they have a potential to rapidly grow under the returning light conditions when the euphotic zone deepened after the bloom.

We need to test if diatoms that have once taken nutrients in darkness can grow rapidly in light afterward and if the light intensity (1% PAR, 0.1% PAR, and complete darkness) has an impact on nutrient uptake as future research. We would like to use Martin-Jezequel et al (2000) and Sohrin et al (2011) as references to plan the future research.

To clarify our viewpoints, we have revised as follows:

**Line 393 - 422**

Nutrient uptake by diatoms in dark subsurface layer in bloom would have impacts on primary production and distribution of phytoplankton in bloom and post-bloom. We propose alternative hypotheses (1 and 2) to deduce the influence of nutrient uptake in the dark subsurface layer.

1) If the diatom population that had consumed half of the nutrients in the dark subsurface layer sank to the deeper layer during the bloom, then the primary production in the subsurface layer after the bloom, at which time it would be part of the euphotic zone, would be reduced by half at maximum compared to the production in the case where there was no nutrient consumption during the dark period.

2) If the diatoms that had consumed nutrients in the dark subsurface layer remained in that layer after the bloom or migrated to the upper layer, they have a potential to rapidly grow under the returning light conditions when the euphotic zone deepened after the bloom.

In the case of Funka Bay, we note that the consumption of nutrients in the dark subsurface layer would have an impact outside the bay, because the subsurface water was exchanged with Oyashio water.

In relation to the second hypothesis, an interesting survival strategy for diatom, *Rhizosolenia*, which forms large aggregations (mats), has been proposed (Villareal et al. 1996; Richardson et al. 1998; Villareal et al., 1999; Villareal et al., 2014). The survival strategy of *Rhizosolenia* is that they consume $NO_3^-$ in the dark subsurface layer, and then migrate to the euphotic zone where they have a growth advantage in oligotrophic subtropical open ocean areas. For the coastal marine diatom, *Thalassiosira weissflogii,* was studied to examine changes in buoyancy in relation to ratios of carbohydrate to protein which determine the cell density (Richardson and Cullen, 1995). They revealed that accumulation of carbohydrate as a result of nitrate depletion leads rises in cellular density and sinking speed and that accumulation of protein as a result of nitrate addition after the nitrate depletion leads a positive buoyancy. Several modelling studies have suggested contributions of primary production by vertically migrating phytoplankton to net primary production. For example, Witz and Lan Smith, (2020) estimated that vertically migrating phytoplankton contributes 7% of net primary production at the subarctic gyre of the western Pacific.

These previous studies have not yet found any evidence of decrease in $NO_3^-$ in the dark subsurface layer from observation. If the hypothesis of diatoms' migration strategy proposed by previous studies is true, the results of our study will provide evidence for the decrease in $NO_3^-$ in the dark subsurface layer associated with the diatoms' strategy.

**Reviewer's comment 4**

3)      The structure of the manuscript

It was just an opinion, but I am familiar with the manuscript which divides results and discussion. I believe the authors can divide them.  However, if the authors considered the present style is better, this is not mandatory.  This is the option, too, but the title of the manuscript should be a more appealing one.  The present title only attracts local interests.  For example, when the authors consider the observed phenomenon possibly occurs everywhere under diatom blooms, the title can be revised as "Significant nutrient consumption in the dark subsurface layer during a diatom bloom: the case study on Funka Bay, Hokkaido, Japan".

**Respond to the comment 4**

According to the comment, we divided results and discussions and revised the title.

**Reviewer's comment 5**

L14: Times of observations are necessary. Technically, the authors' observation is not time-series, because the observation was conducted randomly.

**Respond to the comment 5**

We revised the manuscript accordingly as follows:

**Line 349 - 350**

We conducted ==repetitive== observations in Funka Bay, Hokkaido, Japan, ==15 February, 4 and 15 March, and 14 April 2019.==

**Reviewer's comment 6**

L21 "We believe that this is the first study to present observational evidence for the consumption of the main nutrients by diatoms in the dark subsurface layer during the spring bloom." In my opinion, this sentence is not important. Instead of this sentence, the authors should add why they considered the nutrient decline does not occur by physical processes.

**Respond to the comment 6**

We added a sentence about exclusions of physical processes as an explanation of nutrient reduction in the subsurface layer. We removed the sentence "this is the first study to present ---" from **abstract.**

**Line 21 - 22**

==We excluded possibilities of three physical process, water mixing, diffusive transport, and subduction, as reasons for the decrease in nutrients in the subsurface layer.==

**Reviewer's comment 7**

L26 "Si: NO3– ratio". Yes, this is not wrong and described in Harrison et al 2004, but Si(OH)4:NO3 or Si: N ratio is more appropriate. This is just opinion.

**Respond to the comment 7**

We corrected accordingly.

**Reviewer's comment 8**

L31–32: References are required.

**Respond to the comment 8**

We added a reference (Rosa et al., 2007).

**Reviewer's comment 9**

L34–35: "From time-series observations in the bay, it is possible to examine the temporal changes of biochemical parameters within the same identified water mass while the water is in the bay." I cannot understand this sentence clearly. What means "while the water is in the bay"?

**Respond to the comment 9**

We revised the sentence as follows:

**Line 50 - 53**

==From repetitive observations in the bay, it is possible to collect seawater samples originated from the same water mass in different times when the water remains in the bay during the observation period and then examine the temporal changes of biogeochemical parameters within the same water mass.==

**Reviewer's comment 10**

L37–39: "A massive spring bloom dominated by diatom species occurs in March every year before the Oyashio water flows into the surface of the bay, and it lasts until late March or early April, when Oyashio water occupies the surface of the bay (Odate 1987; Maita and Odate 1988)." I cannot understand this sentence. Please clarify. Can the author divide it into two sentences?

**Respond to the comment 10**

We revised the sentence as follows:

**Line 55 - 58**

==In the Funka Bay, diatom bloom initiates in late winter, February, before Oyashio water flows into the bay (Kudo and Matsunaga, 1999). A massive spring bloom dominated by diatom species occurs in March every year (Odate 1987; Maita and Odate 1988) when Oyashio water flows into the bay. The==

bloom lasts until late March or early April when Oyashio water occupies the surface of the bay (Kudo and Matsunaga, 1999).

**Reviewer's comment 11**

L58–61: I could not find any meaning in these two sentences. I cannot see any discussion of VOIs in this manuscript. In addition, the reference is under consideration. So I considered these two sentences should be removed.

**Respond to the comment 11**

According to the comment, we revised as follows:

Line 50 - 56

From repetitive observations in the bay, it is possible to collect seawater samples originated from the same water mass in different times when the water remains in the bay during the observation period and then examine the temporal changes of biogeochemical parameters within the same water mass. For example, temporal changes in nutrients (Kudo and Matsunaga 1999; Kudo et al. 2000), dissolved iron (Hioki et al. 2015), volatile organic iodine (Shimizu et al. 2017) and isoprene (Ooki et al. 2019) have been examined in relation to primary production in the bay.

**Reviewer's comment 12**

Method: L72–73: "Observations in Funka Bay have been reported elsewhere (Shimizu et al. 2017)." What did the authors want to describe? Some information on the observations conducted in 2019 was described in 2017? Describe the details or remove the sentence.

**Respond to the comment 12**

According to the comment, we removed the sentence.

**Reviewer's comment 13**

L78: How do the authors calculate the analytical precision? This is very low. Did the authors measure the nutrient concentration of not-frozen samples? If the results are frozen samples, the precisions are too good, in particular, silicate.

**Respond to the comment 13**

We added information about the determination of precision as follows.

**Line 91 - 93**

Analytical precision was 0.12% for $NO_3^-$, 0.21% for $NO_2^-$, 0.19% for $PO_4^{3-}$, 0.11% for $SiO_2$, and 0.34% for $NH_4^+$ as determined by repeated measurement (n = 7) of reference seawater for nutrient standards (KANSO, standard Lot BZ, Osaka, Japan).

**Reviewer's comment 14**

L82: Where Thalassiosira nordenskioeldiicome from? Algae collection?

**Respond to the comment 14**

We added a sentence accordingly:

**Line 97 - 99**

We conducted dark incubation experiments four times using a diatom *Thalassiosira nordenskioeldii*, which predominates in the early phase of the spring bloom in Funka Bay (Ban et al 2000). *Thalassiosira nordenskioeldii* was isolated from natural seawater collected in the western subarctic Pacific Ocean in May 2019.

**Reviewer's comment 15**

L95: "We set the initial concentrations of nutrients at 23 times those of the first dark incubation." Why did the authors set so high initial nutrient concentration? The environments are very different from the field observations.

**Respond to the comment 15**

We added information about the amount of added amount of nutrients as follows:

Line 293-295

The added amount of $NO_3^-$ per chl-a (0.022 = 31.1 µmol $L^{-1}$ / 1426 µg $L^{-1}$) was 6% of the ratio of $NO_3^-$/chl-a (0.40 = 4.8 µmol $L^{-1}$ / 12 µg $L^{-1}$) in seawater at 40 m on 4 March.

Line 308-310

In the second experiment, we added excess amount of nutrients into the nutrient-depleted medium, in which cultured diatoms were in a decline phase of growth. The added amount of $NO_3^-$ per chl-a (10.3 = 743.5 µmol $L^{-1}$ / 72.5 µg $L^{-1}$) was 26 times of the seawater at 40 m on 4 March.

**Reviewer's comment 16**

* Did not the authors conduct the microscopic observations?  Is Thalassiosira nordenskioeldii the dominant species of the observations?  This is very important.  Because the other species is dominant, the authors' incubation experiments are meaningless.

**Respond to the comment 16**

We added information about microscopic analysis as follows:

Line 289-292

From microscopic image analysis, *Thalassiosira nordenskioeldii* occupied 14.2% of number of phytoplankton cells (n = 1209) collected by plankton net (mesh = 100 µm) on 15 March 2019. Other dominant species were *Chaetoceros* spp. and other *Thalassiosira* sp. We confirmed that *Thalassiosira nordenskioeldii* was one of the dominant species in the spring bloom 2019.

**Reviewer's comment 17**

L102–106: Please define the water masses at the materials and methods.  When the authors defined in the materials and methods section, the results will be simpler.

**Respond to the comment 17**

We revised the manuscript accordingly.

**Reviewer's comment 18**

L106: "The revised classification result"  This is unclear.  Did the authors revise in this manuscript or revise in Ooki et al. 2019?

**Respond to the comment 18**

We used the definition of water mass classification proposed in Ooki e al. (2019). Ooki et al. (2019) made a minor change in the classification by Ohtani and Kido (1980).F

We removed the phrase "The revised classification result"

**Reviewer's comment 19**

L115–119: This is not a result. Please define in the materials and methods section.

**Respond to the comment 19**

We revised the manuscript accordingly.

**Reviewer's comment 20**

L123: "the original data in supplementary information of Ooki et al., (submitted)." Is it right? I can see the supplementary information of this manuscript.

**Respond to the comment 20**

We have used the same chl-a data in the two papers (this paper and unpublished paper). We removed the description "original data in supplementary information of Ooki et al., submitted". And, we added chl-a data in supplement material.

**Reviewer's comment 21**

L124: "The data for chl-a are taken from a related article." What does it mean? I think it is acceptable to share the data with other manuscripts.

**Respond to the comment 21**

We removed the description "The data for chl-a are taken from a related article".

**Reviewer's comment 22**

L150–168: These paragraphs were "discussion". These discussions can be put after the results section because the results after this paragraph are not contained the results of the discussion. For me, this style is hard to follow.

**Respond to the comment 22**

According to the comment, we moved these paragraphs from "results" to "discussion".

**Reviewer's comment 23**

3.3.2 This section (results of incubation experiments) should be shown before 3.3.1.

**Respond to the comment 23**

We revised accordingly.

**Reviewer's comment 24**

L270: Villareal et al. reported Rhizosolenia, and not Thalassiosira. Do the authors have any evidence on the vertical migration of Thalassiosira? If not, this discussion is speculative. In addition, the authors' names are wrong: Wirtz and Lan Smith (2020) are correct.

**Respond to the comment 24**
We added some explanations about the vertical migration of Thalassiosira (Richardson and Cullen, 1995). To clarify our viewpoint, we revised as follows:

Line 408-418
In relation to the second hypothesis, an interesting survival strategy for diatom, *Rhizosolenia*, which forms large aggregations (mats), has been proposed (Villareal et al. 1996; Richardson et al. 1998; Villareal et al., 1999; Villareal et al., 2014). The survival strategy of *Rhizosolenia* is that they consume $NO_3^-$ in the dark subsurface layer, and then migrate to the euphotic zone where they have a growth advantage in oligotrophic subtropical open ocean areas. For the coastal marine diatom, *Thalassiosira weissflogii,* was studied to examine changes in buoyancy in relation to ratios of carbohydrate to protein which determine the cell density (Richardson and Cullen, 1995). They revealed that accumulation of carbohydrate as a result of nitrate depletion leads rises in cellular density and sinking speed and that accumulation of protein as a result of nitrate addition after the nitrate depletion leads a positive buoyancy. Several modelling studies have suggested contributions of primary production by vertically migrating phytoplankton to net primary production. For example, Witz and Lan Smith, (2020) estimated that vertically migrating phytoplankton contributes 7% of net primary production at the subarctic gyre of the western Pacific.

[Figure]

Fig. 2 Vertical profiles of temperature (a), salinity (b), and density (c) at station 30 in Funka Bay, Japan, on 15 February, 4 March, and 15 March.

[Figure]

Fig. 5 Vertical profiles of $NO_3^-$ (a), $PO_4^{3-}$ (b), $NH_4^+$ (c), $Si(OH)_4$ (d), Chl-a (e), and AOU (f) at station 30 in Funka Bay, Hokkaido, Japan, on 15 February, 4 and 15 March, and 14 April 2019.

[Figure]

Fig. 6 Temporal change of the nutrient concentrations in the dark incubation experiment using the diatom Thalassiosira nordenskioeldii for $NO_3^-$ (a), $PO_4^{2-}$ (b), and $Si(OH)_4$ (c) of the first experiment and $NO_3^-$ (d), $PO_4^{2-}$ (e), and $Si(OH)_4$ (f) of the second experiment. The diatom was pre-cultured for 17 or 42 days under light conditions before nutrients were added. Each incubation bottle (n=1) with nutrient addition was put in darkness on day 0.

[Figure]

Fig. 7 Temporal change of the nutrient and chl-a concentrations (mean ± 1stdev, n = 4) in the dark incubation experiment using the diatom Thalassiosira nordenskioeldii for $NO_3^-$ (a), $PO_4^{2-}$ (b), $Si(OH)_4$ (c), and chl-a (d) of the third experimet and for $NO_3^-$ (e), $PO_4^{2-}$ (f), $Si(OH)_4$ (g), and chl-a (h) of the fourth experiments. The diatom was cultured for 10 or 11 days under light conditions before nutrients were added. Incubation bottles (n=4) with nutrient addition was put in darkness on day 0.

[Figure]

Fig. 8 Horizontal distributions of temperature (a), salinity (b), and density (c) at the surface (1 m) of the Funka-Bay on 4 March 2019. Lines at temperature range 3.5 - 3.6 °C, salinity 33.6, and density 26.7σ were drawn in the figure. The all boudary lines were drawn in an enlarged figure of density. The location of observation station was maked with a cross. The ocean reanalysis product using an operational system for monitoring and forecasting the status of coastal and open-ocean waters around Japan (the JPN system) was provided by Meteorological Research Institute in Japan.

---

## Author Comment (AC3)

Dear anonymous referee2

We will send responses to referee's comments of "Substantial nutrient consumption in the dark subsurface layer during a diatom bloom: the case study on Funka Bay, Hokkaido, Japan" (revised title) by Umezawa et al. The referee's comments were very helpful, and we have revised the manuscript taking all comments. Corrections in the revised manuscript (and the responses to referee's comments) are highlighted.

Best regards,

Corresponding author: Atsushi Ooki

**Contents of response letter**
    Response to referee 2 (comment 1 – 56)
    Revised / added figures (Fig.2, 5, 6, 7, 8)

**Revised chapter structure**
    Abstract
    1. Introduction
    2. Material and methods
        2.1 Shipboard observations
        2.2 Analytical procedures
        2.3 Incubation experiments to test for nutrient consumption by diatoms in darkness
        2.4 Water mass types and mixed-layer and euphotic-zone depths
        2.5 Spatial distributions of temperature, salinity, and density at the sea surface
    3. Results
        3.1 Hydrographic features
        3.2 Biogeochemical parameters
            3.2.1 Chl-a
            3.2.2 Nitrate
            3.2.3 Phosphate and silicate
            3.2.4 Ammonium
        3.3 Limiting factor of primary production during the bloom
    4. Discussions
        4.1 Nitrate consumption in the dark layer between 15 February and 4 March
        4.2 Nutrient consumption in the dark subsurface layer between 4 and 15 March
            4.2.1 Nutrient consumption by diatoms in darkness
            4.2.2 Water mixing as a possible explanation for nutrient reduction
            4.2.3 Diffusive transport between the surface and the subsurface layers
            4.2.4 Subduction of surface water into the subsurface layer
        4.3 The influence of nutrient consumption by diatoms in the dark subsurface layer
    5. Conclusion

**Response to reviewer 2**

GENERAL COMMENTS
**Reviewer's comment 1**
*In general, I feel that the introduction is poorly structured and weak. A lot of background about the hydrology but also about the dynamic of the phytoplankton bloom and productivityin Funka Bay is missing. I invite the authors to develop their bibliography and look closer in the literature for studies that have already been conducted in Funka Bay and develop their introduction a little further. For example (and this is not exhaustive), Nakada et al. (2013) for water mass dynamics, Shinada et al. (1999) and Radiarta and Saitoh (2008) for phytoplankton bloom and productivity dynamics in the bay. Most importantly, the objectives of this work arenot clearly presented here. Why authors decided to focus on the processes affecting nutrient reduction in the dark subsurface layer during the bloom, what is the purpose of this study?
**Response to the comment 1**
According to the comment, we added a reference (Nakada et al 2013) and clarified the purpose of this study. We revised as follows:

Line 31-58
Dissolved iron and nitrate ($NO_3^-$) supplied from below the surface to the surface euphotic zone through winter vertical water mixing sustain spring phytoplankton bloom in the Oyashio region (Nishioka et al. 2011). Most previous studies about marine primary production have concerned with the nutrient consumption by phytoplankton in the euphotic zone because most phytoplankton species, except for dinoflagellates (e.g. Cullen and Horrigan 1981), are commonly assumed to be incapable of moving actively between the surface mixed layer and below the surface (subsurface layer). A few studies have focused on the vertical migration of a diatom, *Rhizosolenia*, to uptake nutrients in the subsurface layer and grow in the euphotic zone in the oligotrophic subtropical Pacific (Villareal et al. 1996; Richardson et al. 1998; Villareal et al., 1999; Villareal et al., 2014). As for the subarctic area, a modelling study that simulated a lot of chl-a profiles, taking into phytoplankton's migration behaviour, demonstrated that vertically migrating phytoplankton can pump up considerable amount of nutrient to the surface layer from the dark subsurface layer and contributes 7% of net primary production at the subarctic gyre of the western Pacific, Oyashio region (Witz and Lan Smith, 2020). These previous studies have not yet shown observational evidence of nutrient reduction associated with consumption by phytoplankton in the dark subsurface layer, however, nutrient reduction in the dark subsurface layer have been found in the Funka Bay, Hokkaido, Japan (Kudo and Matsunaga, 1999), which faces to the Oyashio-Kuroshio transitional area in the western North Pacific.
 Oyashio water reaches the area off the coast of Hokkaido, Japan, where the subtropical water derived from Kuroshio or the Tsugaru warm current waters are also found (Rosa et al., 2007). A small portion of Oyashio water enters Funka Bay in early spring. The bay water exchanges twice a year, with cold Oyashio water in early spring and Tsugaru warm water in early fall (Ohtani 1971). The Oyashio water, a cold and low salinity water, flows into the bay along the northern coast of the bay, forming an anticlockwise flow from late of March to middle of April (Nakada et al. 2013). From repetitive observations in the bay, it is possible to collect seawater samples originated from the same water mass in different times when the water remains in the bay during the observation period and then examine the temporal changes of biogeochemical parameters within the same water mass. For example, temporal changes in nutrients (Kudo and Matsunaga 1999; Kudo et al. 2000), dissolved iron (Hioki et al. 2015), volatile organic iodine (Shimizu et al. 2017) and isoprene (Ooki et al. 2019) have been examined in relation to primary production in the bay. In the Funka Bay, diatom bloom initiates in late winter, February, before Oyashio water flows into the bay (Kudo and Matsunaga, 1999). A massive spring bloom dominated by diatom species occurs in March every year (Odate 1987; Maita and Odate 1988) when Oyashio water flows into the bay. The bloom lasts until late March or early April when Oyashio water occupies the surface of the bay (Kudo and

Matsunaga, 1999).

Line 73-76
In this paper, we examine the temporal variation of nutrient concentrations in Funka Bay from the early phase of the diatom bloom (February) to post-bloom (April) through repetitive observations in 2019. And we focused on the processes affecting nutrient reduction in the dark subsurface layer during the bloom to show evidence of nutrient consumption by diatoms in darkness.

**Reviewer's comment 2**
*Two main comments for the Material and Method section:
I do not think that 4 sampling days conducted in a random way can be considered as a time series, perhaps a survey…
**Response to the comment 2**
According to the comment, we used "repetitive" instead of "time-series".

**Reviewer's comment 3**
When investigating biological processes, incubation experiments such as those conducted in this study MUST be replicated (triplicates are a must), otherwise any interpretation coming outfrom the experiment results is only speculative and conclusion cannot be ensured. This is very important!
**Response to the comment 2**
According to the comment, we carried out the third and fourth incubation experiments additionally (each quadruplicated, n=4). The results of the third and fourth experiments were close to the nutrient reduction found at 30–50-m depths in Funka Bay. However, the dark consumption rates of our experiments and previous studies had wide range values. Thus, we concluded that "dark consumption by diatoms had a potential to reduce nutrients by half in the dark subsurface layer of Funka bay".

We added sentences as follows:
Line 116-125
In the third and fourth experiments, diatom culture *Thalassiosira nordenskioeldii* had grown in modified f/2 medium ($NO_3^-$, 175 µmol $L^{-1}$; $PO_4^{3-}$, 6.5 µmol $L^{-1}$; $Si(OH)_4$, 18.8 µmol $L^{-1}$) for 10 and 11 days, respectively. The chl-*a* concentrations in the pre-culture mediums for the third and fourth experiments reached 145 and 198 µg $L^{-1}$, respectively. The concentrations of $NO_3^-$, $PO_4^{3-}$, and $Si(OH)_4$ in the pre-cultured mediums dropped as follows: $NO_3^-$, 9.27 µmol $L^{-1}$; $PO_4^{3-}$, 0.42 µmol $L^{-1}$; and $Si(OH)_4$, < 1 µmol $L^{-1}$ for the third experiment; $NO_3^-$, 0.66 µmol $L^{-1}$; $PO_4^{3-}$, 0.42 µmol $L^{-1}$; and $Si(OH)_4$, < 1 µmol $L^{-1}$ for the fourth experiment. We added nutrients (stock f/2 medium) into the pre-culture mediums, after which concentrations (day 0) were as follows: $NO_3^-$, 186 µmol $L^{-1}$; $PO_4^{3-}$, 7.7 µmol $L^{-1}$; and $Si(OH)_4$, 16.1 µmol $L^{-1}$ for the third experiment; $NO_3^-$, 213 µmol $L^{-1}$; $PO_4^{3-}$, 9.1 µmol $L^{-1}$; and $Si(OH)_4$, 21.2 µmol $L^{-1}$ for the fourth experiment. Each pre-culture medium (160 mL) was divided into 4 cell-cultivation flasks and put in darkness at 6 °C. On days 1, 2, and 3, 8 mL and 1 mL of incubation medium (n = 4) were filtered to measure nutrient and chl-*a* concentrations, respectively.
Line 316-322
In the third and fourth experiment, the added amount of $NO_3^-$ per chl-a (0.77 – 0.96) was 1.9 – 2.4 times of the seawater at 40 m on 4 March. The results of the third and fourth experiments (Fig. 7a-h) demonstrated that the consumption rates, which were calculated from the concentration difference of nutrient between day 0 and day 3 and the initial chl-a concentrations (day 0), were 0.034 – 0.043 µmol (µg chl-*a*)$^{-1}$ d$^{-1}$ for $NO_3^-$, 0.0059 – 0.0086 µmol (µg chl-*a*)$^{-1}$ d$^{-1}$ for $PO_4^{3-}$, and 0.034 – 0.035 µmol (µg chl-*a*)$^{-1}$ d$^{-1}$ for $Si(OH)_4$. The estimated $\Delta NO_3^-$ (−4.3 ~ −5.4 µmol $L^{-1}$), $\Delta PO_4^{3-}$ (−0.75 ~ −1.1 µmol $L^{-1}$), and $\Delta Si(OH)_4$ (−4.3 ~ −4.3 µmol $L^{-1}$) were close to the actual decreases between the two

dates: $\Delta NO_3^-$, $-2.0$ µmol $L^{-1}$; $\Delta PO_4^{3-}$, $-0.12$ µmol $L^{-1}$; $\Delta Si(OH)_4$, 3.7 µmol $L^{-1}$.
Line 335-336
Although the dark consumption rates had wide ranges, we concluded that dark consumption by diatoms had a potential to reduce nutrients by half in the dark subsurface layer of Funka Bay.

**Reviewer's comment 4**

*Section 3.2 of the Discussion is really long compared to other sections and contain a lot of redundancies. I suggest to re-organize this section chronologically instead of per nutrient, for example 3.2.1 pre-bloom, 3.2.2 bloom and 3.2.3 post-bloom situation. This might help reducingthe length of this section (especially for chl*a* and nitrate) and remove most of the redundancies.

**Response to the comment 4**

According to the reviewers', we have largely reconstructed the manuscript . We divided the chapter "results and discussion" into separate chapters.

**Reviewer's comment 5**

Moreover, this will allow authors to discuss the evolution of the nutrient ratios which will help confirming or infirming their hypothesis and which has not been explored in the discussion?

**Response to the comment 5**

According to the comment, we added a chapter as follows:

Line 247-257

3.3 Limiting factor of primary production during the bloom

On 15 February before the occurrence of massive diatom bloom, the average concentrations of $NO_3^-$, $PO_4^{3-}$ and $Si(OH)_4$ at the surface $0 - 10$ m were 9.1, 0.86 and 19.8 µmol $L^{-1}$, respectively. On 4 March at the peak of the bloom, the average concentrations of $NO_3^-$, $PO_4^{3-}$ and $Si(OH)_4$ at the surface $0 - 10$ m were 0.34, 0.43 and 5.6 µmol $L^{-1}$, respectively. Uptake ratios of N:P and Si:N at the surface between 15 February and 4 March were 20.5 $\{= (9.1 - 0.34) / (0.86 - 0.43)\}$ and 1.62 $\{= (19.9 - 5.6) / (9.1 - 0.34)\}$, respectively. Similar uptake ratios during diatom bloom in Funka Bay have been reported to be N:P = $15.6 - 23.6$ and Si:N = $1.9 - 2.7$ (Kudo and Matsunaga 1999). From the uptake ratio of N:P, $NO_3^-$ in the surface water could have been depleted since 4 March. On 15 March at the decline phase of the bloom, the average concentrations of $NO_3^-$, $PO_4^{3-}$ and $Si(OH)_4$ at the surface $0 - 10$ m were 0.54, 0.37 and 4.5 µmol $L^{-1}$, respectively. Since sufficient amount of $Si(OH)_4$ remained in the surface water on 15 March, we considered that the N-depletion in the surface water limited primary production after the peak of the bloom.

Line 334-335

In the dark subsurface layer of Funka Bay, N-depleted diatoms sunk from the surface after the peak of bloom could have enhanced $NO_3^-$ consumption in darkness.

**Reviewer's comment 6**

*The conclusion is poorly reflecting the main observations and hypothesis raised by the authors to explain the nutrient drawdown observed in the dark subsurface water of Funka Bay. Indeed, Authors cannot reject the influence of vertical mixing as they did L284-288, they haven't discussed the possibility of nutrient diffusion (active or passive), and the consumption rate of diatoms measured during the incubation experiment are not enough to fully explain the observed nutrient drawdown. The conclusion MUST be revised to truly reflect the main results of this study.

**Response to the comment 6**

We added discussions about possibilities of diffusive transport of nitrate and entrainment of

subducted water from the surface. And we revised the explanation about the results of incubation experiments.

Line 360 – 387

[revised manuscript text omitted]

**Reviewer's comment 7**
ABSTRACT
The abstract should use non-technical terms. For example, while "mixed-layer" is a common term used in oceanography and it is ok to use it in the abstract, Dark Zone Depth needs to be defined. Another option could be use "below the euphotic zone" instead of "dark zone depth".The main conclusion of the abstract needs to be revised since the nutrient consumption by diatom in the dark layer is likely not the only possible explanation to the observed nutrient drawdown in the subsurface waters of Funka Bay.

**Response to the comment 7**

According to the comment, we added a definition "dark-zone depths (0.1% of surface PAR depth) " in the abstract. We defined the euphotic-zone depth and the dark-zone depth as 1% and 0.1%, respectively, of surface PAR depth. Thus, we used the term, "dark-zone depth".

We added explanations that the physical processes were excluded from reasons of the nutrient reduction.

Line 21-22

We excluded possibilities of three physical process, water mixing, diffusive transport, and subduction, as reasons for the decrease in nutrients in the subsurface layer.

**Reviewer's comment 8**

L14 and later in the manuscript: February 15th to April 14th of 2019. Please be careful on the calendar notation along the manuscript.

**Response to the comment 8**

Professional scientific editors (ELSS, Inc., Tsukuba, Japan) recommended to use "15 February to 14 April". If the BG editor requires proof reading by professional editors for the revised manuscript, we will offer proof reading to ELSS again.

**Reviewer's comment 9**

L16: "On both date"? During the whole time series from Feb 15th to Apr 14th? or during the bloom from March 4th to March 15th? By the way, winter water was above the euphotic zone (so above the dark-zone depths) on March 4th in fig.2.

**Response to the comment 9**

We have revised as "on 4 and 15 March".

Yes, winter water was found above the euphotic zone depth on 4 March. Deeper winter water remained below the dark-zone depths from 4 March to 15 March.

**Reviewer's comment 10**

L17: Same comment as above, please specify which dates. On fig.4, while $NO_3^-$ concentration decreases by approximately half, it is absolutely not the case for $Si(OH)_4$ and $PO_4^{3-}$ (e.g. at 50m concentrations dropped from 0.7 to 0.6 $\mu mol\ L^{-1}$ and from 15 to 13 $\mu mol\ L^{-1}$ for $PO_4^{3-}$ and $Si(OH)_4$ respectively)

**Response to the comment 10**

According to the comment 9, we specified the two dates (4 and 15 March) at the prior sentence, thus "these dates" is specified as "4 and 15 March".

**Reviewer's comment 11**

L20-23: Authors present a list of several publications that have already studied dark nutrient consumption rates and ratios (see L250-259)

**Response to the comment 11**

According to the comment, we removed the sentence from abstract.

**Reviewer's comment 12**

L25-26: Surface euphotic zone sounds redundant.

**Response to the comment 12**

According to the comment, we removed "surface" form "surface euphotic" throughout the manuscript.

**Reviewer's comment 13**

L28: Use either "Si:N ratio" or "Si(OH)$_4$:NO$_3^-$ ratio"

**Response to the comment 13**

According to the comment, we used "Si:N ratio".

**Reviewer's comment 14**

L38: It is actually believed that the intrusion of the Oyashio water in the Bay triggers the diatombloom by strengthening water column stratification.

**Response to the comment 14**

We recognize that diatom growth begins in late winter before the Oyashio water flows into the bay and massive diatom bloom in the bay is triggered by intrusion of the Oyashio water.

According to the comment, we revised as follows:

Line 55-57

 In the Funka Bay, diatom bloom initiates in late winter, February, before Oyashio water flows into the bay (Kudo and Matsunaga, 1999). A massive spring bloom dominated by diatom species occurs in March every year (Odate 1987; Maita and Odate 1988) when Oyashio water flows into the bay.

**Reviewer's comment 15**

L56-58: This is unfortunate since nutrient limitation can change from time to time. This manuscript doesn't have to focus on nutrient limitation in the surface water, however, it would be interesting to have a paragraph (even a short one) about what controls the dynamic of the bloom in the euphotic zone during this study. This is important since nutrient drawdown occurring in the surface layer will likely affect the subsurface biogeochemistry by creating gradient (and potentially diffusive fluxes) of nutrients, or by triggering adaptive response from the biology (such as vertical migration, resting spore formation etc.)

**Response to the comment 15**

According to the comment, we added a chapter about nutrient ratio and nutrient limitation as follows:

Line 247-257

**3.3 Limiting factor of primary production during the bloom**

On 15 February before the occurrence of massive diatom bloom, the average concentrations of NO$_3^-$, PO$_4^{3-}$ and Si(OH)$_4$ at the surface 0 – 10 m were 9.1, 0.86 and 19.8 µmol L$^{-1}$, respectively. On 4 March at the peak of the bloom, the average concentrations of NO$_3^-$, PO$_4^{3-}$ and Si(OH)$_4$ at the surface 0 – 10 m were 0.34, 0.43 and 5.6 µmol L$^{-1}$, respectively. Uptake ratios of N:P and Si:N at the surface between 15 February and 4 March were 20.5 {= (9.1 – 0.34) / (0.86 – 0.43)} and 1.62 {= (19.9 – 5.6) / (9.1 – 0.34)}, respectively. Similar uptake ratios during diatom bloom in Funka Bay have been reported to be N:P = 15.6 – 23.6 and Si:N = 1.9 – 2.7 (Kudo and Matsunaga 1999). From the uptake ratio of N:P, NO$_3^-$ in the surface water could have been depleted since 4 March. On 15 March at the decline phase of the bloom, the average concentrations of NO$_3^-$, PO$_4^{3-}$ and Si(OH)$_4$ at the surface 0 – 10 m were 0.54, 0.37 and 4.5 µmol L$^{-1}$, respectively. Since sufficient amount of Si(OH)$_4$ remained in the surface

water on 15 March, we considered that the N-depletion in the surface water limited primary production after the peak of the bloom.

**Reviewer's comment 16**

L58-61: VOI are not presented nor discussed in this manuscript. Please remove these sentencesor add a figure, present and discuss the results of VOI measurements, and highlight their contribution in answering the objectives of the present study (why VOIs are measured?).

**Response to the comment 16**

According to the comment, the reference about VOI was removed from there.

**Reviewer's comment 17**

L72-73: What does this sentence mean? What kind of observations? Are they relevant for this paper? Please remove this sentence or develop.

**Response to the comment 17**

According to the comment, we removed the sentence.

**Reviewer's comment 18**

L77: $SiO_2$ is not a nutrient. This is the formula for particulate silica, silicic acid (the nutrient) is $Si(OH)_4$, please make sure to double-check the notations throughout the manuscript.

**Response to the comment 18**

According to the comment, we changed "$SiO_2$" to "$Si(OH)_4$" throughout the manuscript.

**Reviewer's comment 19**

L78: Analytical precision: this is almost too good to be true! I am curious to see how this has been calculated?

**Response to the comment 19**

We measured reference seawater for nutrient standards (KANSO). The precisions of repetitive analysis are pretty good.

According to the comment, we added an explanation about the precision as follows:

Line 91-93

Analytical precision was 0.12% for $NO_3^-$, 0.21% for $NO_2^-$, 0.19% for $PO_4^{3-}$, 0.11% for $Si(OH)_4$, and 0.34% for $NH_4^+$ as determined by repetitive measurement (n = 7) of reference seawater for nutrient standards (KANSO, standard Lot BZ, Osaka, Japan).

**Reviewer's comment 20**

L83: please use axenic instead of sterile. F/2 medium has approximately 106 µmol.L$^{-1}$ $Si(OH)_4$ 882 µmol.L$^{-1}$ NO $^-$, and 36 µmol.L$^{-1}$ PO $^{3-}$ (see Guillard and Ryther, 1962; Guillard, 1975 or Andersen et al (2005), Algal Culturing Techniques). It is more likely a modified (not particularlySi-rich) f/2 medium then, although I agree it is still plenty of nutrients for diatoms.

**Response to the comment 20**

According to the comment, we used "axenic" instead of "sterile" and added a word "modified".

**Reviewer's comment 21**

L86: This is really unfortunate since checking for contamination is crutial regarding the objectiveof the incubation experiment. Indeed, contamination by a population of other micro-algae or bacteria can affect the consumption (or recycling) of $NO^-$ and $PO^{3-}_3$ (although it is probably less the case for $Si(OH)_4$). This have to be mentioned and discussed in the discussion.

**Response to the comment 21**

According to the comment, we added an explanation about the bacterial contamination as follows:
Line 297-299

==Since we did not check the bacterial contamination after the experiment, bacterial consumption and/or recycling of nutrients in the culture might have an effect on the results. We assumed that the bacterial activity had a less effect on nutrient changes in the high-density diatom culture.==

**Reviewer's comment 22**

L93-97: What was the purpose of this second dark incubation? and why adding much morenutrient compare to the first one? Please explain.

**Response to the comment 22**

Silicic acid was almost depleted on day 2 of the first experiment. If the diatoms consumed $Si(OH)_4$ much more quickly than we collected the culture sample on day 2, the daily consumption rates are underestimated. So, we added much more nutrients in the second experiment.

According to the comment, we added explanations as follows:

Line 306-309

==Silicic acid was almost depleted on day 2. If the diatoms exhausted $Si(OH)_4$ more quickly than we collected the culture sample on day 2, the daily consumption rates are underestimated.==

==In the second experiment, we added excess amount of nutrients into the nutrient-depleted medium, in which cultured diatoms were in a decline phase of growth.==

**Reviewer's comment 23**

3.1. Hydrographic features

Are the water masses defined based only on salinity? Or did the authors use a T-S diagram to define the distribution (depth range) of the different water masses during the study? If so, it would have been useful to have a figure with a T-S diagram that illustrate the position of watermass end-members (such as in Shimizu et al. 2017) in supplement material at least.

**Response to the comment 23**

According to the comment, we added an illustration of water mass definition as supplement material.

**Reviewer's comment 24**

L111: "biogeochemical"

3.2  Biogeochemical parameters

My guess is that authors meant "biogeochemical" instead of "biochemical" since biogeochemical refers to processes associated nutrient cycles and nutrient distribution whilebiochemical is more related to intra-cellular processes.

**Response to the comment 24**

According to the comment, we used "biogeochemical" instead of "biochemical".

**Reviewer's comment 25**

L115-118: Why authors chose this definition of the mixed layer, please explain AND cite appropriate reference. Same comment for euphotic zone and dark depth zone, appropriatereferencing is missing.

**Response to the comment 25**

According to the comment, we added references and explanations why we adapted the threshold 0.125 kg m$^{-3}$ as follows:

Line 135-143

The surface mixed layer was defined as the layer in which density differences ($\Delta\sigma$) were within 0.125 kg m$^{-3}$ relative to the density at 5-m depth. The threshold $\Delta\sigma = 0.125$ kg m$^{-3}$ is often used for monthly mean of mixed layer in oceanic climate studies (Spall 1991), while the threshold $\Delta\sigma = 0.01$ kg m$^{-3}$ is used for snap-shot observations (Thomson and Fine 2003). We used the maximum threshold $\Delta\sigma = 0.125$ kg m$^{-3}$ to ensure that the subsurface layer water had not mixed with the surface layer during intervals (11 days to a month) between our observations. The euphotic-zone depth was defined as the depth at which photosynthetically active radiation (PAR) was 1% of the surface PAR, where amount of photosynthesis is equal to respiration (Marra 2004). We defined the dark-zone depth at which PAR was 0.1% of the surface PAR, where amount of photosynthesis is approximately tenth part of the 1% PAR depth taking into account the light intensity only as a limiting factor of photosynthesis.

**Reviewer's comment 26**

L118: in general terms like "much less" should be avoided in papers, please be more specific and give a threshold or a range.

Also, mixed layer, euphotic zone and dark depth zone can be defined in the previous paragraph,since they can be considered as hydrographic features as well.

**Response to the comment 26**

According to the comment, we revised the sentence. Please see the response to the comment 25.

**Reviewer's comment 27**

L120-124: Authors cannot discuss in detail data that are not available either in another ALREADY published work, in a database, in supplement or in the main text of the manuscript.Authors cannot refer to an unpublished work when it concerns data availability, especially when these data are used to make some of the figures. This is not acceptable! Moreover, I don't understand this sentence since the chl*a* profile in fig.3 looks pretty low and homogenous on Feb 15 and it seems that the chl*a* data ARE presented in the supplement…

**Response to the comment 27**

According to the comment, we added chl-a data in supplementary information of this paper.

**Reviewer's comment 28**

L125-126: 27 to 30 µg L$^{-1}$ of chl*a* sounds very high to me. It looks like it's between 1.5 and 2 times higher than what is usually found in the bay (see L126-128). Is it a particularly exceptional year? Why is chl*a* that high?

**Response to the comment 28**

Yes, chl-a concentrations of 27 to 30 µg L$^{-1}$ are very high. We will examine the mechanism of such a high chl-a event as future research.

**Reviewer's comment 29**

L131: Sinking particles and suspended particles are different things. What makes authors conclude that particles can be sinking vs. in suspension? Please develop.

**Response to the comment 29**

We can not make a distinction between sinking particle and suspended particle. We used "sinking" in the manuscript.

**Reviewer's comment 30**

L140: A reference for the annual maximum is missing.

**Response to the comment 30**

The annual maximum level (~20 µmol $L^{-1}$) of nitrate was found in the bottom water in summer (e.g. Kudo et al. 2007).

According to the comment, we removed a phrase "the annual maximum".

**Reviewer's comment 31**

L150-167: Since there is a section specifically discussing the nutrient consumption in the dark subsurface layer, I would suggest to remove those hypotheses here to avoid redundancy with section 3.3. and to add this material to the discussion in section 3.3. This section 3.2 could thusbe renamed "nutrient dynamic in the euphotic zone and dark subsurface layer" or something similar.

**Response to the comment 31**

We have largely reconstructed the manuscript. We divided the chapter "results and discussion" into separate chapters.

**Reviewer's comment 32**

L168-175: I do not see why the first explanation of vertical mixing suggested for March 4th $NO_3^-$ decrease at depth cannot be still valid in March 15th. Indeed, authors based their explanation mostly on the AOU profile. However, AOU profile in the 30-50m layer in March 14th looks prettymuch the same as in March 4th (see fig.3).

**Response to the comment 32**

The reason for the same AOU values in the subsurface layer between 4 and 15 March is that net $O_2$ production had not occurred there.

According to the comment, we added AOU profiles (Fig. 5f) to show that the values stayed unchanged between the two dates. And we changed a phrase "are slightly decrease" to "remained almost the same values".

**Reviewer's comment 33**

L180-186: Note that the signal of regeneration is also clearly seen from the strongly positivevalues of AOU at depth in April 14th.

**Response to the comment 33**

According to the comment, we added a sentence about AOU increase in the bottom water as follows:

Line 234-235

==Obvious increase of AOU in the deep water (80 – 95-m depth) was found from 15 March (average 20.9 µmol $L^{-1}$) to 14 April (average 56.0 µmol $L^{-1}$), see Fig. 4b and Fig. 5f.==

**Reviewer's comment 34**

L185: It is important to be more specific here. For example, the sentence can be "The very high $PO_4^{3-}$, $Si(OH)_4$ and $NH_4^+$ concentrations measured at depth (1.9, 26.1, and 3.2 µmol L$^{-1}$, respectively) clearly indicate that nutrient regeneration already occurred at the bottom of the water column in April 14$^{th}$".

**Response to the comment 34**

According to the comment, we added explanations about rises in bottom water concentrations in detail as follows:

Line 215-217

In the deep water (80–95 m), the $NO_3^-$ concentrations slightly increased with time since 15 March: 5.38 µmol L$^{-1}$ on 4 March, 5.26 µmol L$^{-1}$ on 15 March, and 6.60 µmol L$^{-1}$ on 14 April.

Line 231-238

In contrast to the subsurface layer, the average concentrations of $PO_4^{3-}$ and $Si(OH)_4$ in the deep layer (80 – 95-m depth) increased with time; 0.78 µmol L$^{-1}$ and 15.3 µmol L$^{-1}$ on 4 March, 0.89 µmol L$^{-1}$ and 22.3 µmol L$^{-1}$ on 15 March, and 1.57 µmol L$^{-1}$ and 25.1 µmol L$^{-1}$ on 14 April, respectively. Obvious increase of AOU in the deep water (80 – 95-m depth) was found from 15 March (average 20.9 µmol L$^{-1}$) to 14 April (average 56.0 µmol L$^{-1}$), see Fig. 4b and Fig. 5f. Because the obvious increase of $PO_4^{3-}$ coincided with the rise in AOU, it likely resulted from remineralization following the decomposition of organic matter suspended in the bottom water or settled on the seafloor. The increase of $Si(OH)_4$ in the bottom water is likely resulted from dissolution of biogenic silica settled on the seafloor.

Line 242-246

Because the $NH_4^+$ concentrations were at their lowest during winter with total column average of 0.25 µmol L$^{-1}$ on 15 February, the signal from remineralization could be clearly detected on 15 March with average of 0.54 µmol L$^{-1}$ at the subsurface water (30 – 50 m). The deep water $NH_4^+$ concentrations obviously increased with time since 4 March: 0.31 µmol L$^{-1}$ on 4 March, 0.95 µmol L$^{-1}$ on 15 March, and 3.05 µmol L$^{-1}$ on 14 April.

**Reviewer's comment 35**

3.3   Nutrient consumption in the dark subsurface layer

I suggest to add a subsection between subsection 3.3.1 and 3.3.2 to discuss a third hypothesis for nutrient drawdown in the dark subsurface layer, the diffusive flux of nutrient, that have been neglected so far in the manuscript but I think could be of significant importance.

**Response to the comment 35**

We added a discussion about the diffusive transport of nitrate. Please see the response to the comment 6.

**Reviewer's comment 36**

L211: It is currently impossible to see the density difference between 5 and 30m in fig. 4 since both density profiles are incomplete. Moreover, it is more about the shape or curvature of the density profile rather than the magnitude of the density difference. Indeed, a smooth and progressive gradient of density will not act as a strong barrier against exchange compared to a sharp change in density (even though this latter is smaller).

**Response to the comment 36**

According to the comment, we added CTD profile (Fig.2) to show all plot and used the density gradient instead of the magnitude of density difference.

Line 341-342

Because the density (σ) gradient between 20-m and 30-m depths, $(\sigma_{30m} - \sigma_{20m}) / (30\ m - 20\ m)$, 0.0033 (kg m$^{-3}$ m$^{-1}$) on 4 March substantially increased to 0.021 (kg m$^{-3}$ m$^{-1}$) on 15 March,

**Reviewer's comment 37**

L212-214: Please show these data in a figure

**Response to the comment 37**

The data can obtain from the JMA website. We added a URL of download site as follows:

Line 345

https://www.data.jma.go.jp/risk/obsdl/index.php

**Reviewer's comment 38**

L215: Although this might be true, I do not agree with authors since we do not have access to the observations that support this conclusion (there is no figure illustrating the wind speed andthe density profiles on fig.4 are incomplete).

**Response to the comment 38**

According to the comment, we revised the figure (CTD profiles in Fig. 2) to show all plots.

**Reviewer's comment 39**

L218-228: This could be another subsection focused on discussing horizontal advection

**Response to the comment 39**

We added a discussion about a possibility of subduction water (subducted water horizontally moved to the observation station). Please see the response to the comment 6.

**Reviewer's comment 40**

L222-224: A mixing ratio of 8:2 between Funka-Bay and Oyashio waters is a curious way to present the mixing between these two water masses. Won't it be clearer to say that to generate the observed decrease in salinity, a mixing with 25% of Oyashio water (or 1:4 Oyashiowater and Funka Bay water) is necessary. By the way, by using salinities of 33.0, 33.47 and 33.58 for Osahio water, Funka Bay water, and the mix between the two of them, respectively; my estimate is 20% (1:5) of Oyashio water needed to produce the observed decrease in salinity. L227-228: This is not because the vertical mixing cannot explain all the nutrient drawdown that this hypothesis should be excluded, it is more likely not the main driver of the nutrient drawdown. It doesn't mean that it is not happening, it is more likely a combination of different processes. I suggest to rephrase this conclusion.

**Response to the comment 40**

According to the comment, we revised the sentence as follows:

Line 353-355

A mixing between 20% of Oyashio water and 80% of Funka Bay water at 30 m would change the salinity at 30-m depth from 33.58 (on 4 March) to 33.47 (on 15 March).

**Reviewer's comment 41**

L231-232: Same comment as above. The nutrient reduction in the subsurface layer is more likely driven by multiple processes, consumption of nutrient by diatom in the dark might be oneof them (not the only one).

**Response to the comment 41**

We agree with the comment 41 that the nutrient reduction in the subsurface layer is driven by multiple processes. We estimated that the diffusive transport occupied approximately 10% of the reduction and concluded that the consumption by diatom had the most important effect on the reduction. Please see the response to the comment 6.

**Reviewer's comment 42**

L237 and after: Please explain the μg chl$a^{-1}$ d$^{-1}$ unit used for consumption rates. Chl$a$ concentrations during the incubation experiment are not presented in fig.5 nor elsewhere in the manuscript. Also, in the main text $NO^-$, $PO_3^{3-}$ as well as Si(OH) consumption are discussedbut only $NO^-$ is show in fig.5. The evolution of ALL nutrient concentration MUST be presented somewhere if authors are discussing them in the manuscript.

**Response to the comment 42**

According to the comment, we added an explanation of the consumption rate and the values of chl-a concentrations in the figure. We used the chl-a concentrations on day 0 for the calculations of nutrient consumption rate per unit chl-a.

And, we added figures of the dark incubation results for $PO_4$ and $Si(OH)_4$ and added a sentence explaining chl-a changes during the dark incubations.

Line 299-301

The daily consumption rates per unit chl-a amount calculated from the concentration difference of nutrients between day 0 and day 2 and the initial concentration of chl-a (1426 μg L$^{-1}$) of the dark incubation

Line 322-323

The chl-a concentrations were increased in darkness from 145 μg L$^{-1}$ (day 0) to 250 μg L$^{-1}$ (day 3) for the third experiment and from 198 μg L$^{-1}$ (day 0) to 294 μg L$^{-1}$ (day 3) for the fourth experiment.

Revised figures: Fig. 6 and Fig. 7 (attached below)

**Reviewer's comment 43**

L238-342: Although calculation of consumption rates cannot be verified since nutrient concentration during the incubation are not presented, it seems that the consumption rate of diatoms estimated from the incubation experiment are very low and cannot fully explain the nutrient drawdown observed in the 30-50m layer between march 4[th] and March 14[th].

**Response to the comment 43**

According to the comments, we additionally conducted the third and fourth experiments. We explained that "although the dark consumption rates had wide ranges, we concluded that dark consumption by diatoms had a potential to reduce nutrients by half in the dark subsurface layer of Funka Bay" as follows:

Line 317-322

The results of the third and fourth experiments (Fig. 7a-h) demonstrated that the consumption rates, which were calculated from the concentration difference of nutrient between day 0 and day 3 and the initial chl-a concentrations (day 0), were 0.034 – 0.043 μmol (μg chl-$a$)$^{-1}$ d$^{-1}$ for $NO_3^-$, 0.0059 – 0.0086 μmol (μg chl-$a$)$^{-1}$ d$^{-1}$ for $PO_4^{3-}$, and 0.034 – 0.035 μmol (μg chl-$a$)$^{-1}$ d$^{-1}$ for $Si(OH)_4$. The estimated $\Delta NO_3^-$ (–4.3 ~ –5.4 μmol L$^{-1}$), $\Delta PO_4^{3-}$ (–0.75 ~ –1.1 μmol L$^{-1}$), and $\Delta Si(OH)_4$ (–4.3 ~ –4.3 μmol L$^{-1}$) were close to the actual decreases between the two dates: $\Delta NO_3^-$, –2.0 μmol L$^{-1}$; $\Delta PO_4^{3-}$, –0.12 μmol L$^{-1}$; $\Delta Si(OH)_4$, 3.7 μmol L$^{-1}$.

Line 335-336

Although the dark consumption rates had wide ranges, we concluded that dark consumption by diatoms had a potential to reduce nutrients by half in the dark subsurface layer of Funka Bay.

**Reviewer's comment 44**

L243-245: Are these consumption rates calculated between day 0 and day 2 such as in the first incubation experiment? I don't think rates from the second incubation experiment can be compared to rates in the first experiment since too many parameters were different between the two experiments (e.g. concentrations were much higher in the second experiment, since diatom uptake usually follows a Michaelis-Menten function, this can change a lot the dynamic of the diatom uptake!). Does this mean that, because the nutrient concentrations in the field were much lower compare to those set up during both experiments, we can assume that the nutrient consumption rates were probably much lower in the water column and thus would probably contribute to a very small amount of the observed nutrient drawdown?

**Response to the comment 44**

We revised the time interval for the consumption rate calculation of the second experiment to between day 0 and day 2. We added explanations about the amount of added nitrate per unit chlorophyll. The observed nitrate amount per unit chlorophyll in the bay were within the range of added nitrate amount per unit chlorophyll in cultures of dark incubations.

Line 293-295
The added amount of $NO_3^-$ per chl-a (0.022 = 31.1 µmol $L^{-1}$ / 1426 µg $L^{-1}$) was 6% of the ratio of $NO_3^-$/chl-a (0.40 = 4.8 µmol $L^{-1}$ / 12 µg $L^{-1}$) in seawater at 40 m on 4 March.
Line 309-310
The added amount of $NO_3^-$ per chl-a (10.3 = 743.5 µmol $L^{-1}$ / 72.5 µg $L^{-1}$) was 26 times of the seawater at 40 m on 4 March
Line 316-317
In the third and fourth experiment, the added amount of $NO_3^-$ per chl-a (0.77 – 0.96) was 1.9 – 2.4 times of the seawater at 40 m on 4 March.

**Reviewer's comment 45**

L247-249: I do not understand this sentence. Why would the consumption rate be largely dependent on the chl*a* content in the diatom cells if, as suggested by authors L165, this nutrientconsumption occurs "without photosynthetic growth"? Please explain.

**Response to the comment 45**

Although we did not count cell number in the seawater sample collected in the bay and the sample of incubation experiment, capacity of nutrient consumption in darkness might depend on the cell density and nutrition conditions for each cell. We will examine property of nutrient consumption in darkness associated with chl-a, cell number, nutrition condition of diatom cell as future research.

**Reviewer's comment 46**

L248-249: I totally agree with this statement!

**Response to the comment 46**

The main purpose of this study is to show the observational results of nutrient reduction in the dark subsurface layer associated with consumption by diatom, excluding possibilities of physical processes. Investigations of dark consumption properties associated with nutrition condition of diatom cell are remained for future research.

**Reviewer's comment 47**

L251: I do not understand this sentence, please rephrase.

**Response to the comment 47**

We revised the sentences as follows:

Line 324-326

==Cochlan et al. (1991) carried out onboard incubations with a diatom dominating natural seawater setting dark periods of 2–4 hours after light periods. They have reported dark consumption rates for $NO_3^-$ of 0.09–0.14 μmol (μg chl-$a$)$^{-1}$ d$^{-1}$, which are close to the results from our second incubation.==

**Reviewer's comment 48**

L268-273: All diatoms do not migrate. Those vertical migrations have been observed for mats of Rhizosolenia. Do authors have evidence for such a behavior in Thalassiosira? This needs to be discussed here.

**Response to the comment 48**

According to the comment, we added a reference about the changes in buoyancy of Thalassiosira (Richardson and Cullen, 1995).

Line 411-415

==For the coastal marine diatom, *Thalassiosira weissflogii,* was studied to examine changes in buoyancy in relation to ratios of carbohydrate to protein which determine the cell density (Richardson and Cullen, 1995). They revealed that accumulation of carbohydrate as a result of nitrate depletion leads rises in cellular density and sinking speed and that accumulation of protein as a result of nitrate addition after the nitrate depletion leads a positive buoyancy.==

**Reviewer's comment 49**

L275: What are those assumptions? They need to be presented and discussed here.

**Response to the comment 49**

According to the comment, we revised the sentence as follows:

Line 420-422

==These previous studies have not yet found any evidence of decrease in $NO_3^-$ in the dark subsurface layer from observation. If the hypothesis of diatoms' migration strategy proposed by previous studies is true, the results of our study will provide evidence for the decrease in $NO_3^-$ in the dark subsurface layer associated with the diatoms' strategy.==

**Reviewer's comment 50**

L279-280: Once again, I'm not convinced that 4 sampling days can be considered as a time series.

**Response to the comment 50**

We used "repetitive" instead of "time-series" in the revised manuscript.

**Reviewer's comment 51**

L291-292: I do not agree with this. The nutrient consumption rates estimated from the incubation experiment are too low to explain the nutrient drawdown observed in the 30-50m layer in Funka Bay.

**Response to the comment 51**

We carried out additional incubation experiments. The results were close to the observed nutrient reduction rate per chlorophyll. Please see the response to the comment 43.

**Reviewer's comment 52**

Figure 1: There is only one station/sampling site in this study. Describe what O (Oyashio water)and T

(Tsugaru water) are in the figure caption or legend.

**Response to the comment 52**

According to the comment, we described "T: Subtropical Tsugaru current" and "O: Subpolar Oyashio current" in the figure legend.

**Reviewer's comment 53**

Figure 2: Define the euphotic zone and MLD in the captions instead of in the legend. Be consistent in the formatting of the legend (e.g. WO: Transitional water instead of TransitionalWater: WO).

**Response to the comment 52**

According to the comment, we described the definition of the euphotic-zone and mixed-layer depths in the caption and corrected the legends, "WO: Transitional Water".

**Reviewer's comment 54**

Figure 3: MLD is not at the same depth on Apr 14[th] in fig.2 (40m) and fig.3 (15m), same on March 4[th] fig.2 (~5m) vs. fig.3 (~10m). Please remove the red rectangle that is supposed to show the concentration change in 4-15 March. It actually doesn't help to read the figure. Most importantly this figure involves a lot of interpolations made from a total of four stations. The time coverage is probably not enough and could generate bias in the figure that could lead to misinterpretation. For example, on panel a. it seems that the bloom started soon after the sampling in Feb15, and peaked on March 4[th], but we know that diatoms can consume nutrients pretty quickly in the mixed layer and the bloom could have started anytime between Feb 15[th] and March 4[th], same for the termination of the bloom. I would suggest to present these data as profiles which will be more accurate instead of interpolated time-sections.

**Response to the comment 54**

We think that the rectangle showing the depth-temporal range (30-50m, 4-15 March) helps to read the figure. According to the comment, we added figures of vertical profile of chl-a, nutrients and AOU on (Fig. 5) to avoid the misinterpretation. We corrected the MLD on Apr 14th in the figure and revised a sentence as follows:

Line 184-185

==**15 March** Chl-*a* concentrations had decreased at all depths by 15 March, however, there were still high levels (0–10 m, 11.0–16.2 µg L$^{-1}$) within the euphotic surface mixed layer (0 – 18 m) and in the deeper dark layer (20–95 m, 2.3–7.8 µg L$^{-1}$).==

Fig.5

Please see an attached figure below.

**Reviewer's comment 55**

Figure 4: The scale on the x axis in the temperature, salinity and density panels are not appropriate since the profile corresponding to March 15[th] in all panels is cut at the surface, same for the March 4[th] density profile. It is hard to discuss incomplete profiles. Right now, wecannot conclude that stratification is stronger on March 15[th] as stated L211.

The nutrient concentration difference in panel d, e and f are pretty obvious indeed, no need to add extra arrows. Moreover, the left-pointing arrow on panel c doesn't mean anything since density increases in the mixed layer from March 4[th] to March 15[th]. I would be interesting to add the standard deviations for the nutrient concentration measurements.

**Response to the comment 55**

According to the comment, we revised the CTD profiles (Fig.2) to show all and remove the arrows.

Although we determined the analytical precision of nutrient measurements by repetitive analysis of standard seawater (KANSO), we did not measure the seawater sample more than three times a sample. Thus, we could not show the standard deviation on the figure.

Fig.2
Please see an attached figure below.

**Reviewer's comment 56**
Figure 5: Overall, I like this figure. It is clear, although the scale of the x axis on panel b does notmake sense (two squares correspond to 4 days (0 to 4) then to two days (4 to 6), then back to 4days (6 to 10). I also still don't understand how the authors have defined their set-up conditions(e.g. why adding 30μmol. $L^{-1}$ $NO_3^-$ in the first experiment and around 750μmol $L^{-1}$ in the second one? Why this threshold of 1429μg $L^{-1}$ chl$a$ in the first experiment but 72.5μg $L^{-1}$ in the second one? Although I don't remember is the culture experiment has been replicated (Working with biology, I think it is crucial to run that kind of experiments in triplicate, or at least duplicate them) but this figure needs error bars.

**Response to the comment 56**
According to the comment, we revised the figure. The initial concentrations of chl-a were the results of pre-cultivation for growing diatom till nutrients were once exhausted in the light condition. According to the comment, we carried out additional experiments (n = 4). We revised the figures of the first and second experiments and added figures of the additional third and fourth experiments.

Fig. 6 and Fig.7
Please see attached figures below.

[Figure]

Fig. 2 Vertical profiles of temperature (a), salinity (b), and density (c) at station 30 in Funka Bay, Japan, on 15 February, 4 March, and 15 March.

[Figure]

Fig. 5 Vertical profiles of NO$_3^-$ (a), PO$_4^{3-}$ (b), NH$_4^+$ (c), Si(OH)$_4$ (d), Chl-a (e), and AOU (f) at station 30 in Funka Bay, Hokkaido, Japan, on 15 February, 4 and 15 March, and 14 April 2019.

[Figure]

Fig. 6 Temporal change of the nutrient concentrations in the dark incubation experiment using the diatom Thalassiosira nordenskioeldii for $NO_3^-$ (a), $PO_4^{2-}$ (b), and $Si(OH)_4$ (c) of the first experiment and $NO_3^-$ (d), $PO_4^{2-}$ (e), and $Si(OH)_4$ (f) of the second experiment. The diatom was pre-cultured for 17 or 42 days under light conditions before nutrients were added. Each incubation bottle (n=1) with nutrient addition was put in darkness on day 0.

[Figure]

Fig. 7 Temporal change of the nutrient and chl-a concentrations (mean ± 1stdev, n = 4) in the dark incubation experiment using the diatom Thalassiosira nordenskioeldii for $NO_3^-$ (a), $PO_4^{3-}$ (b), $Si(OH)_4$ (c), and chl-a (d) of the third experimet and for $NO_3^-$ (e), $PO_4^{3-}$ (f), $Si(OH)_4$ (g), and chl-a (h) of the fourth experiments. The diatom was cultured for 10 or 11 days under light conditions before nutrients were added. Incubation bottles (n=4) with nutrient addition was put in darkness on day 0.

[Figure]

Fig. 8 Horizontal distributions of temperature (a), salinity (b), and density (c) at the surface (1 m) of the Funka-Bay on 4 March 2019. Lines at temperature range 3.5 - 3.6 °C, salinity 33.6, and density 26.7σ were drawn in the figure. The all boudary lines were drawn in an enlarged figure of density. The location of observation station was maked with a cross. The ocean reanalysis product using an operational system for monitoring and forecasting the status of coastal and open-ocean waters around Japan (the JPN system) was provided by Meteorological Research Institute in Japan.

---

## Author Response (AR2)

**Response to review Report#1**

**Reviewer's comment #1-1)**

The authors described "diatoms that had been growing in the surface waters and then sank to the subsurface layer" (L25–26). However, any evidence did not show the growth in the surface layer. In addition, the authors did not reject the effect of vertical migration of diatoms. Therefore, I cannot clearly understand what the authors want to say here. I feel that this sentence is not essential. The authors described that pre-incubation is necessary for nutrient consumption of diatom in the dark (is it right? Please explain clearly in the results section), and the fact Please clarify.

**Response to the comment #1-1)**

According to the comment, we removed the description "" and revised the sentences to explain the fact as follows.

**Abstract, Line 19-25:**

Incubation experiments using the diatom *Thalassiosira nordenskioeldii* showed that this diatom could consume added nutrients in the dark at substantial rates after the pre-culturing to deplete nutrient. Incubation experiment using natural seawater collected in growing phase of bloom on 8 March 2022 also showed that nutrient-depleted phytoplankton could consume added nutrients in the dark. We excluded possibilities of three physical process, water mixing, diffusive transport, and subduction, as the main reasons for the decrease in nutrients in the subsurface layer. We conclude that the nutrient reduction in the subsurface layer (30–50 m) between 4 and 15 March 2019 could be explained by dark consumption by diatoms at that layer.

**Conclusions, Line 484-489:**

(1) From the dark incubation experiments, we confirmed that the diatom *Thalassiosira nordenskioeldii*, which is one of the dominant diatom species in the bloom of Funka Bay, could consume added nutrients in the dark at substantial rates after preculturing to deplete nutrients and that phytoplankton in nutrient-depleted natural seawater collected in the bay before the peak of diatom bloom on 8 March 2022 could also consumed added nutrients in

the dark. Although the consumption rates varied over a wide range, we concluded that dark consumption of nutrient by diatom at the dark subsurface layer had a potential to reduce nutrient by half in the dark subsurface layer (30–50 m).

**Reviewer's comment #1-2)**

2) The structure of the manuscript must be improved again. I could not find the results of "incubation experiments." I confused the descriptions of the "without pre-culturing" experiment. I could not find the explanations of the "without pre-culturing" experiment in the materials and methods sections, so I cannot understand what this indicate.
The results section is hard to follow. The description is fragmentary. In particular, at 3.2 Biogeochemical parameters.

**Response to the comment #1-2)**

According to the comment, we have revised the structure of the manuscript.
1) We described the results of incubation experiments in the Result chapter.
2) We changed the description from "without pre-culturing" to "without nutrient addition"
3) We removed meaningless descriptions from the incubation results.

The result of continuous dark incubation using natural seawater without pre-culturing, in which medium seawater had originally high concentration of $NO_3^-$ (4.6 – 7.4 $\mu$ mol $L^{-1}$) on 8 March 2022, was inconsistent with the result of the third *Thalassiosira* experiments, in which medium had the same level of $NO_3^-$ (9.27 $\mu$ mol $L^{-1}$) at the end of pre-culturing before nutrients were added. That is, nutrients were not consumed in continuous dark by the natural seawater experiment but substantially consumed in the dark by the third *Thalassiosira* experiments. In the third *Thalassiosira* experiments, $NO_3^-$ in the pre-cultured medium was rapidly consuming on day 10 of the pre-culturing with approximately 8.61 $\mu$ mol $L^{-1}$ per day, which was calculated from the concentration difference of $NO_3^-$ in pre-cultured medium on day 10 of the third experiment and day 11 of the fourth experiment: (9.27 – 0.66) $\mu$ mol $L^{-1}$ / (11 – 10) day. We considered that the rapid consumption of nutrients had been maintained in the dark period of the third *Thalassiosira* experiment.

4) We removed meaningless descriptions that had made fragmentary and hard to follow from the chapter 3.2 "Biogeochemical parameters".

**3.2.2 Nitrate**

......... We hypothesized that the diatoms that had settled from the surface to the subsurface layer consumed $NO_3^-$ in the dark. This possibility will be discussed in section 4.2. The $NO_3^-$ concentrations in the deeper layer (60–95 m) had not changed since 4 March.

**14 April** By 14 April, the euphotic-zone depth had deepened to 47 m. The $NO_3^-$ concentrations had decreased to below the detection limit ($<0.05$ $\mu$ mol $L^{-1}$) in the upper euphotic zone (0–30 m) and decreased to 1.4 $\mu$ mol $L^{-1}$ in the lower euphotic zone (40 m).  In the deep water (80–95 m), the $NO_3^-$ concentrations slightly increased from 5.26 $\mu$ mol $L^{-1}$ on 15 March to 6.60 $\mu$ mol $L^{-1}$ on 14 April. ~~There is a time lag for regeneration of $NO_3^-$ in bottom water after organic matter decomposition because the regeneration of $NO_3^-$ follows the remineralization of $NH_4^+$ from organic matter and its oxidation (nitrification). A time lag of 1–2 months for $NO_3^-$ regeneration after $NH_4^+$ regeneration has been observed every year in Funka Bay (Kudo et al. 2007). Thus, the signal from $NO_3^-$ regeneration could not be seen during the spring bloom, and a slight signal was detected on 14 April. In contrast, signals of the regeneration of $PO_4^{3-}$ and $NH_4^+$ from organic matter in bottom water were obvious on 15 March and 14 April, after the decline phase of the bloom, as discussed in the next two sections.~~

**Reviewer's comment #1-3)**

3) To reject the effect of the physical processes, the authors use the ocean circulation model. Please check temperature and salinity in the model is the same as in the observations (should be added as supplement information). Is the model localized for Funka Bay? If the T-S is the same as in the observation, I feel the authors cannot reject the possibility of physical processes without the gridded observations. Of course, the authors can be the effect of diatom consumption in the dark condition may be the primary cause based on the quantitative estimation of the incubation experiments.

**Response to the comment #1-3)**

According to the comment, we added modelled vertical profiles of temperature, salinity, and density on 4 and 15 March 2019 around the station 30 in Funka Bay in supplementary figure (Fig. s3). The model results were not good agreement with the observational results. We also explained about it.

**Line 432-437:** Note that the modelled vertical profiles of temperature, salinity and density were not in good agreement with the observed profiles (Fig. s3). In the model result, the influence of low salinity Oyashio water, which flowed in the surface of Funka Bay, was estimated to be stronger than the observational result. If the influence of Oyashio water had been as strong as the model result, the subsurface water (30 – 50 m) might have changed between 4 and 15 March 2019. We considered that the subsurface water had stayed between both dates because of weak Oyashio water inflow.

**Reviewer's comment #1-4)**

4) Please check the usages of parentheses and tilde (~). Tilda means similarity and does not mean "range."

**Response to the comment #1-4)**

According to the comment, we revised the manuscript not to use '~' as range.

**Line 412 – 417:** The range of diffusive transport of $NO_3^-$ were calculated to be 0.00022 – 0.0022 $\mu$ mol m s$^{-2}$ between 20 m and 30 m, which could result in concentration change of 0.021–0.21 $\mu$ mol L$^{-1}$ at 30 m for 11 days. Concentration changes between 30 m and 40 m and between 40 m and 50 m were calculated to be 0.013–0.13 $\mu$ mol L$^{-1}$ and 0.011 – 0.11 $\mu$ mol L$^{-1}$, respectively. The sum of concentration changes at 30 m, which include transports from 20 m layer and 40 m layer, ranges from –0.20 $\mu$ mol L$^{-1}$ (= –0.21 + 0.013) to +0.11 $\mu$ mol L$^{-1}$ (= –0.021 + 0.13). Ranges of the sum of concentration changes at 40 m and 50 m were from –0.12 to +0.096 $\mu$ mol L$^{-1}$ and from –0.11 to –0.024 $\mu$ mol L$^{-1}$, respectively.

**Response to review Report#2**

**Reviewer's comment #2-1)**

The dynamics of primary production and associated nutrient consumption in the ocean surface layer have focused on the processes in the euphotic layer, where sufficient light is available for photosynthesis. In this study, the authors discuss various aspects on the apparent nutrient utilization in the twilight zone, where light is not sufficient for photosynthesis.

I have serious concern on the logics that nutrient consumption occurs in the field by defining the twilight layer as the dark layer.

The first thing to consider is that in oceanography, the lower limit of the euphotic layer is defined as the layer that attenuates to 1% of the surface PAR. On the other hand, the surface PAR varies widely in time and space with the changes with the cloud cover and solar incidence angle. The surface PAR under clear skies may exceed 2000 µE m-2 sec-1, and the calculated PAR at the bottom of the euphotic layer is 20 µE m-2 sec-1. On cloudy days, the surface PAR drops below 200 µE m-2 sec-1, and the PAR at the bottom of the euphotic layer is 2 µE m-2 sec-1. On the other hand, the compensating light intensity at which photosynthesis and respiration balance in the photosynthesis-light curve should have a certain value ranging from 1.2 µE m-2 sec-1 to 30 µE m-2 sec-1, depending on the literature, calculated from 24-hour values (Marra, 2004; Regaudie-de-Gioux & Duarte, 2010). It seems possible that the light condition exceeds the compensation light intensity at depths below the euphotic layer during the daytime under clear skies in the field, whereas the compensation light intensity of T. nordenskioeldii is not known. Furthermore, the evaluation of compensation light intensity for net production should be done on a daily (24-hour) basis, not on an instantaneous basis, as in the oxygen-based net production study by Gran (1912). Compensation light intensity is also expressed in terms of 24-hour integrated photon flux. This is because the positive production during the daytime is balanced by the negative production during the nighttime, when respiration exceeds production. In other words, photosynthesis may take place during the daytime even below the euphotic layer and the associated nutrient uptake may occur although nutrient uptake does not always synchronize photosynthesis. Thus, it is not appropriate to consider that nutrient uptake occurs in the absence of photosynthesis below the euphotic layer. In addition, the sense of the balance of carbon dioxide and oxygen in photosynthesis and respiration can not apply to the uptake of nutrients in the metabolic cycle of phytoplankton. In other words, in respiration, phytoplankton does not produce nutrients. The author should revise the present manuscript considering this point.

**Response to the comment #2-1)**

According to the comment, we clearly defined the dark-layer depth, which does not mean aphotic layer, as follows.

**Line 164-166**: We defined the dark-layer depth at which PAR was 0.1% of the surface PAR, where amount of photosynthesis is approximately tenth part of the 1% PAR depth taking into account the light intensity only as a limiting factor of photosynthesis.

And we added vertical profile of ratio of PAR relative to the surface PAR and temporal variation of the surface PAR obtained from the Muroran meteorological observatory to show the light environment during the bloom. Please see the added figure (Fig. 2d) and supplementary figure (Fig. s2). We added explanations about light environment. According to the referee's comment, we used compensation light intensity expressed by 24-hour average (Regaudie-De-Gioux and Duarte, 2010) to compare the light intensity at the dark-layer depth during the bloom in Funka Bay. We assumed that photosynthesis below the dark-layer depth (less than 0.1% of surface PAR) made no difference in biochemical parameters such as nutrients.

**Line 190-198:** As for light environment, euphotic-zone depths (or compensation depth), which are defined as the depth where PAR was 1% of the surface PAR, were 40 m on 15 February, 11 m on 4 March, and 17 m on 15 March. Dark-layer depths, which we defined as the depth where PAR was 0.1% of surface PAR, were 60 m on 15 February, 17 m on 4 March, and 30 m on 15 March (Fig. 2d). The daily average of surface PAR during the period between 4 March and 15 March including day and night was 19.3 mol photon m$^{-2}$ d$^{-1}$ (= 224 $\mu$ mol photon m$^{-2}$ s$^{-1}$), which was estimated from the global solar radiation at the Muroran meteorological observatory (Fig. s2). The daily average of PAR at the dark-layer depth was estimated to be 0.0193 mol photon m$^{-2}$ d$^{-1}$, which was only 1.8% of global average of compensation irradiance (1.1 mol photon m$^{-2}$ d$^{-1}$) for metabolic balance, photosynthesis = respiration (Regaudie-De-Gioux and Duarte, 2010) Below the dark-layer depths, we assumed that photosynthesis made no difference in biochemical parameters described in latter sections.

We toned down the expression about photosynthesis at the dark layer, as follow.
**Line 355-357**: The latter reduction could not have been affected by photosynthetic consumption by diatoms because there was almost no light available for photosynthesis. Here we discuss the possible reasons for the nutrient reductions between 4 and 15 March.

According to the comment " In other words, in respiration, phytoplankton does not produce nutrients", we removed the following description, because nutrients could be consumed when AOU did not change at compensation depth.

**Reviewer's comment #2-2)**

L309: In assessing dominant phytoplankton, the use of plankton net (100 μm mesh) may cause a failure in collecting smaller phytoplankton even diatom species. It seems OK for just collecting T. nordenskioeldii for culture experiment.

**Response to the comment #2-2)**

We also think that the use of plankton net (100 μm mesh) is OK just for collecting aggregates of T. nordenskioeldii for culture experiment.

**Reviewer's comment #2-3)**

The authors should cite Kudo et al. (2015) in ECSS. They conducted annual primary production measurement in Funka Bay by the in situ 24-hour mooring incubation.

**Response to the comment #2-3)**

According to the comment, we cited the latest paper which reported annual primary production in Funka Bay (Kudo et al., 2015), as follows.

**Line 63-64**: One-third of annual primary production occurs during the spring bloom (Kudo and Matsunaga 1999; Kudo et al. 2015).

---

## Author Response (AR3)

**Response to editor's correction**

We corrected the manuscript according to the editor's correction.

>Line 465: Several modelling⋯.REF NEEDED HERE

We have found a modelling study that estimated contribution of vertically migrating phytoplankton to primary production (Witz and Lan Smith, 2020). Several modelling studies have reported effects and characteristics of vertically migrating phytoplankton on their growth and vertical distribution (not primary production). Therefore, we corrected as follows.

Line 465-467: A modelling study estimated that vertically migrating phytoplankton contributes 7% of net primary production at the subarctic gyre of the western Pacific (Witz and Lan Smith, 2020).

We found a mistake in the unit of diffusive transport and corrected as follows.

Line 412: diffusive transport of $NO_3^-$ were calculated to be 0.00022 – 0.0022 $\mu$ mol $m^{-2}$ $s^{-1}$